# Maximum Entropy Model Correction in Reinforcement Learning

**Amin Rakhsha**[1,2]**, Mete Kemertas**[1,2]**, Mohammad Ghavamzadeh**[3]**, Amir-massoud Farahmand**[1,2]
[1]Department of Computer Science, University of Toronto, [2]Vector Institute, [3]Amazon
`{aminr,kemertas,farahmand}@cs.toronto.edu, ghavamza@amazon.com`

## Abstract

We propose and theoretically analyze an approach for planning with an approximate model in reinforcement learning that can reduce the adverse impact of model error. If the model is accurate enough, it accelerates the convergence to the true value function too. One of its key components is the MaxEnt Model Correction (MoCo) procedure that corrects the model's next-state distributions based on a Maximum Entropy density estimation formulation. Based on MaxEnt MoCo, we introduce the Model Correcting Value Iteration (MoCoVI) algorithm, and its sampled-based variant MoCoDyna. We show that MoCoVI and MoCoDyna's convergence can be much faster than the conventional model-free algorithms. Unlike traditional model-based algorithms, MoCoVI and MoCoDyna effectively utilize an approximate model and still converge to the correct value function.

## 1 Introduction

Reinforcement learning (RL) algorithms can be divided into model-free and model-based algorithms based on how they use samples from the environment with dynamics $\mathcal{P}$. Model-free algorithms directly use samples from $\mathcal{P}$ to approximately apply the Bellman operator on value functions. At its core, the *next-state expectations* $\mathbb{E}_{X' \sim \mathcal{P}(\cdot|x,a)}[\phi(X')]$ are estimated for a function $\phi$, such as the value function, at all state-action pairs $(x, a)$. Model-based reinforcement learning (MBRL) algorithms, on the other hand, use samples from the environment to train a world model $\hat{\mathcal{P}}$ to approximate $\mathcal{P}$. The world model $\hat{\mathcal{P}}$ can be considered an approximate but cheap substitute of the true dynamics $\mathcal{P}$, and is used instead of $\mathcal{P}$ to solve the task.

The world model $\hat{\mathcal{P}}$ often cannot be learned perfectly, and some inaccuracies between $\mathcal{P}$ and $\hat{\mathcal{P}}$ is inevitable. This error in the model can catastrophically hinder the performance of an MBRL agent, especially in complex environments that learning an accurate model is challenging (Talvitie, 2017; Jafferjee et al., 2020; Abbas et al., 2020). In some of these challenging environments, estimating the next-state expectations accurately might be much easier than learning a model. Motivated by this scenario, we aim to bridge the gap between model-based and model-free algorithms and ask: *Can we improve MBRL algorithms by using both the next-state expectations and the approximate model $\hat{\mathcal{P}}$?*

In this paper, we consider a discounted MDP with the true dynamics $\mathcal{P}$, and we suppose that we have access to an approximate model $\hat{\mathcal{P}} \approx \mathcal{P}$. At this level of abstraction, we do not care about how $\hat{\mathcal{P}}$ is obtained – it may be learned using a conventional Maximum Likelihood Estimate (MLE) or it might be a low-fidelity and fast simulator of the true dynamics $\mathcal{P}$. We further assume that for any function $\phi$ of states, we can obtain the next-state expectations $\mathbb{E}_{X' \sim \mathcal{P}(\cdot|x,a)}[\phi(X')]$ for all states $x$ and actions $a$. We consider this procedure costly compared to ones involving $\hat{\mathcal{P}}$ which will be considered free.

We propose the MaxEnt Model Correction (MaxEnt MoCo) algorithm, which can reduce the impact of model error on MBRL agents regardless of their planning algorithm. MaxEnt MoCo first estimates $\mathbb{E}_{X' \sim \mathcal{P}(\cdot|x,a)}[\phi_i(X')]$ for all $(x, a)$ and a set of *measurement* functions $\phi_i$. The main idea is that whenever the planning algorithm normally uses $\hat{\mathcal{P}}(\cdot|x,a)$ for some state-action $(x, a)$, a corrected distribution $\bar{p}$ is calculated and used instead. The distribution $\bar{p}$ is obtained by minimally modifying $\hat{\mathcal{P}}(\cdot|x,a)$ so that the next-state expectations $\mathbb{E}_{X' \sim \bar{p}}[\phi_i(X')]$ based on $\bar{p}$ are (more) consistent with the estimated $\mathbb{E}_{X' \sim \mathcal{P}(\cdot|x,a)}[\phi_i(X')]$. This procedure is known as Maximum Entropy density estimation (Dudík et al., 2007) – hence the name MaxEnt MoCo. We show that if the true value function can

be well-approximated by a linear combination of the measurement functions $\phi_i$, the value function estimated by MaxEnt MoCo can be significantly more accurate than the normally computed one using $\hat{\mathcal{P}}$.

We also introduce Model Correcting Value Iteration (MoCoVI) (Section 4) and its sample-based variant MoCoDyna (Section 5), which iteratively update the set of measurement functions $\phi_i$. These algorithms select their past value functions as the measurement functions, and execute MaxEnt MoCo to get a new, more accurate value function. This choice of measurement functions proves to be effective. We show that if the model is accurate enough, MoCoVI and MoCoDyna can converge to the true value function, and the convergence can be much faster than a model-free algorithm that doesn't have access to a model. In this paper, we study the theoretical underpinnings of maximum entropy model correction in RL. We provide theoretical analysis that applies to both finite and continuous MDPs, and to the approximate versions of the algorithms with function approximation.

## 2   BACKGROUND

In this work, we consider a discounted Markov Decision Process (MDP) defined as $M = (\mathcal{X}, \mathcal{A}, \mathcal{R}, \mathcal{P}, \gamma)$ (Szepesvári, 2010). We use commonly used definitions and notations, summarized in Appendix B. We briefly mention that we denote the value of a policy $\pi$ by $V^\pi$ and the optimal value function by $V^*$. Whenever we need to be explicit about the dependence of the value functions to reward kernel $\mathcal{R}$ and the transition kernel $\mathcal{P}$, we use $V^\pi = V^\pi(\mathcal{R}, \mathcal{P})$ and $V^* = V^*(\mathcal{R}, \mathcal{P})$. For any function $\phi \colon \mathcal{X} \to \mathbb{R}$, we define $\mathcal{P}\phi \colon \mathcal{X} \times \mathcal{A} \to \mathbb{R}$ as $(\mathcal{P}\phi)(x, a) \triangleq \int \mathcal{P}(\mathrm{d}x'|x, a)\phi(x')$ for all $(x, a) \in \mathcal{X} \times \mathcal{A}$. We refer to the problem of finding $V^{\pi_{\mathrm{PE}}}$ for a specific policy $\pi_{\mathrm{PE}}$ as the *Policy Evaluation (PE)* problem, and to the problem of finding an optimal policy as the *Control* problem. In this paper, we assume an approximate model $\hat{\mathcal{P}} \approx \mathcal{P}$ is given. We define $\hat{V}^\pi$ and $\hat{\pi}^*$ in the approximate MDP $\hat{M} = (\mathcal{X}, \mathcal{A}, \mathcal{R}, \hat{\mathcal{P}}, \gamma)$ similar to their counterparts in the true MDP $M$. We assume the PE and control problems can be solved in $\hat{M}$ as it is a standard part of MBRL algorithms.

### 2.1   IMPACT OF MODEL ERROR

In MBRL, the agent relies on the approximate model $\hat{\mathcal{P}}$ to solve the PE and Control problems (Sutton, 1990). A purely MBRL agent learns value functions and policies only using $\hat{\mathcal{P}}$, which means it effectively solves the approximate MDP $\hat{M} = (\mathcal{X}, \mathcal{A}, \mathcal{R}, \hat{\mathcal{P}}, \gamma)$ instead of the true MDP $M$. The advantage of this approach is that it only requires access to the cost-efficient $\hat{\mathcal{P}}$, hence avoiding costly access to $\mathcal{P}$ (e.g., via real-world interaction). However, the model error can dramatically degrade the agent's performance (Talvitie, 2017; Jafferjee et al., 2020; Abbas et al., 2020). The extent of the performance loss has been theoretically analyzed in prior work (Ávila Pires and Szepesvári, 2016; Talvitie, 2017; Farahmand et al., 2017; Farahmand, 2018). To characterize model errors and their impact mathematically, we define the following error measure for each state-action pair $(x, a)$:

$$\epsilon_{\mathrm{Model}}(x, a) = \sqrt{D_{\mathrm{KL}}(\,\mathcal{P}(\cdot|x, a)\,\|\,\hat{\mathcal{P}}(\cdot|x, a)\,)}. \tag{2.1}$$

We note that the choice of KL divergence for quantifying the model error is a natural one. Indeed, in conventional model learning (see e.g., Janner et al. 2019), a common choice of optimization objective is the maximum likelihood estimation (MLE) loss, which minimizes the empirical estimate of the KL-divergence of the approximate next-state distribution to the ground-truth. The following lemma provides performance guarantees for an MBRL agent as a function of $\epsilon_{\mathrm{Model}}$. Similar bounds have appeared in recent work (Ávila Pires and Szepesvári, 2016; Farahmand, 2018; Rakhsha et al., 2022).

**Lemma 1.** *Suppose that $\mathcal{P}$ is the true environment dynamics, $\hat{\mathcal{P}}$ is an approximation of $\mathcal{P}$, and $\|\epsilon_{\mathrm{Model}}\|_\infty = \sup_{x, a \in \mathcal{X} \times \mathcal{A}} \epsilon_{\mathrm{Model}}(x, a)$ is the worst-case error between them. Let $c_1 = \gamma\sqrt{2}/(1 - \gamma)$. We have $\|V^{\pi_{\mathrm{PE}}} - \hat{V}^{\pi_{\mathrm{PE}}}\|_\infty \leq \frac{\gamma}{1-\gamma}\|(\mathcal{P}^{\pi_{\mathrm{PE}}} - \hat{\mathcal{P}}^{\pi_{\mathrm{PE}}})V^{\pi_{\mathrm{PE}}}\|_\infty \leq c_1\|\epsilon_{\mathrm{Model}}\|_\infty \cdot \|V^{\pi_{\mathrm{PE}}}\|_\infty$ and $\|V^* - V^{\hat{\pi}^*}\|_\infty \leq \frac{2c_1\|\epsilon_{\mathrm{Model}}\|_\infty}{1-c_1\|\epsilon_{\mathrm{Model}}\|_\infty}\|V^*\|_\infty$.*

Note that the model error impacts the PE solution through the term $(\mathcal{P}^{\pi_{\mathrm{PE}}} - \hat{\mathcal{P}}^{\pi_{\mathrm{PE}}})V^{\pi_{\mathrm{PE}}}$. A similar observation can be made for the Control problem. This dependence has been used in designing value-aware losses for model learning (Farahmand et al., 2017; Farahmand, 2018; Voelcker et al., 2022; Abachi et al., 2022) and proves to be useful in our work as well.

## 2.2 MAXIMUM ENTROPY DENSITY ESTIMATION

Consider a random variable $Z$ defined over a domain $\mathcal{Z}$ with unknown distribution $p \in \mathcal{M}(\mathcal{Z})$, and a set of measurement functions $\phi_i : \mathcal{Z} \to \mathbb{R}$ for $i = 1, 2, \ldots, d$. Suppose that the expected values $\bar{\phi}_i = \mathbb{E}_p[\phi_i(Z)]$ of these functions under $p$ are observed. Our goal is to find a distribution $q$ such that $\mathbb{E}_q[\phi_i(Z)]$ matches $\bar{\phi}_i$ for all $i$. For example, if $\mathcal{Z} = \mathbb{R}$, $\phi_1(z) = z$, and $\phi_2(z) = z^2$, we are interested in finding a $q$ such that its first and second moments are the same as $p$'s.

In general, there are many densities that satisfy these constraints. Maximum entropy (MaxEnt) principle prescribes picking the most *uncertain* distribution as measured via (relative) entropy that is consistent with these observations (Jaynes, 1957). MaxEnt chooses $q^* = \mathrm{argmax}_{\mathbb{E}_q[\phi_i(Z)]=\bar{\phi}_i} H(q)$, where $H(q)$ is the entropy of $q$, or equivalently, it minimizes the KL divergence (relative entropy) between $q$ and the uniform distribution (or Lebesgue measure) $u$, i.e., $q^* = \mathrm{argmin}_{\mathbb{E}_q[\phi_i(Z)]=\bar{\phi}_i} D_{\mathrm{KL}}(\, q \parallel u \,)$.

In some applications, prior knowledge about the distribution $q$ is available. The MaxEnt principle can then be generalized to select the distribution with the minimum KL divergence to a prior $\hat{p}$:

$$q^* = \underset{\mathbb{E}_q[\phi_i(Z)]=\bar{\phi}_i}{\mathrm{argmin}} D_{\mathrm{KL}}(\, q \parallel \hat{p} \,). \tag{2.2}$$

This is called the Principle of minimum discrimination information or the Principle of Minimum Cross-Entropy (Kullback, 1959; Shore and Johnson, 1980; Kapur and Kesavan, 1992), and can be viewed as minimally *correcting* the prior $\hat{p}$ to satisfy the constraints given by observations $\bar{\phi}_i$. In line with prior work, we call density estimation under this framework *MaxEnt density estimation* whether or not the prior is taken to be the uniform distribution (Dudík et al., 2004; Dudík et al., 2007).

While the choice of KL divergence is justified in various ways (e.g., the axiomatic approach of Shore and Johnson 1980), the use of other divergences has also been studied in the literature (Altun and Smola, 2006; Botev and Kroese, 2011). Although we focus on KL divergence in this work, in principle, our algorithms can also operate with other divergences provided that solving the analogous optimization problem of the form (2.2) is computationally feasible.

Problem (2.2) and its variants have been studied in the literature; the solution is a member of the family of Gibbs distributions:

$$q_\lambda(A) = \int_{z \in A} \hat{p}(\mathrm{d}z) \cdot \exp\left(\sum_{i=1}^d \lambda_i \phi_i(z) - \Lambda_\lambda\right), \tag{2.3}$$

where $A \subseteq \mathcal{Z}$, $\lambda \in \mathbb{R}^d$, and $\Lambda_\lambda$ is the log normalizer, i.e., $\Lambda_\lambda = \log \int \hat{p}(\mathrm{d}z) \cdot \exp\left(\sum_{i=1}^d \lambda_i \phi_i(z)\right)$. The dual problem for finding the optimal $\lambda$ takes the form

$$\lambda^* = \underset{\lambda \in \mathbb{R}^d}{\mathrm{argmin}} \log \int \hat{p}(\mathrm{d}z) \exp\left(\sum_{i=1}^d \lambda_i \phi_i(z)\right) - \sum_{i=1}^d \lambda_i \bar{\phi}_i \,. \tag{2.4}$$

Iterative scaling (Darroch and Ratcliff, 1972; Della Pietra et al., 1997), gradient descent, Newton, and quasi-Newton methods (see Malouf, 2002) have been suggested for solving this problem. After finding $\lambda^*$, if $\mathrm{Var}[\exp(\sum_i \lambda_i \phi_i(\hat{Z}))]$ for $\hat{Z} \sim \hat{p}$ is small, e.g. when $\hat{p}$ has low stochasticity, $\Lambda_\lambda^*$ can be estimated with samples from $\hat{p}$. Then, one can sample from $q^*$ by sampling from $Z_0 \sim \hat{p}$ and assign the importance sampling weight $\exp\left(\sum_{i=1}^d \lambda_i^* \phi_i(Z_0) - \Lambda_{\lambda^*}\right)$. In general algorithms such Markov Chain Monte Carlo can be used for sampling (Brooks et al., 2011). When the observations $\bar{\phi}_i$ are empirical averages, Maximum entropy density estimation is equivalent to maximum likelihood estimation that uses the family of Gibbs distributions of the form (2.3) (Della Pietra et al., 1997).

## 3 MAXIMUM ENTROPY MODEL CORRECTION

As discussed in Section 2.2, MaxEnt density estimation allows us to correct an initial estimated distribution of a random variable using the expected values of some functions of it. In this section, we introduce the MaxEnt Model Correction (MaxEnt MoCo) algorithm, which applies this tool to correct the next-state distributions in the approximate model $\hat{\mathcal{P}}$ towards the true distributions in $\mathcal{P}$.

We assume that for any function $\phi\colon \mathcal{X} \to \mathbb{R}$, we can obtain (an approximation of) $\mathcal{P}\phi$. This operation is at the core of many RL algorithms. For instance, each iteration $k$ of Value Iteration (VI) involves obtaining $\mathcal{P}V_k$ for value function $V_k$. This procedure can be approximated when samples from $\mathcal{P}$ are available with techniques such as stochastic approximation (as in TD Learning) or regression (as in fitted value iteration). Due to its dependence on the true dynamics $\mathcal{P}$, we consider this procedure costly and refer to it as a *query*. On the other hand, we will ignore the cost of any other calculation that does not involve $\mathcal{P}$, such as calculations and planning with $\hat{\mathcal{P}}$. In Section 3.1, we consider the exact setting where similar to the conventional VI, we can obtain $\mathcal{P}\phi$ exactly for any function $\phi\colon \mathcal{X} \to \mathbb{R}$. Then in Section 3.2, we consider the case that some error exists in the obtained $\mathcal{P}\phi$, which resembles the setting considered for approximate VI.

## 3.1 EXACT FORM

In this section, we assume that for any function $\phi\colon \mathcal{X} \to \mathbb{R}$, we can obtain $\mathcal{P}\phi$ exactly. We show that in this case, MaxEnt density estimation can be used to achieve planning algorithms with strictly better performance guarantees than Lemma 1. To see the effectiveness of MaxEnt density estimation to improve planning, consider the idealized case where the true value function $V^{\pi_{\mathrm{PE}}}$ for the PE problem is known to us. Consequently, we can obtain $\mathcal{P}V^{\pi_{\mathrm{PE}}}$ by querying the true dynamics $\mathcal{P}$. Assume that we could perform MaxEnt density estimation (2.2) for every state $x$ and action $a$. We minimally change $\hat{\mathcal{P}}(\cdot|x,a)$ to a new distribution $\bar{\mathcal{P}}(\cdot|x,a)$ such that $\mathbb{E}_{X'\sim\bar{\mathcal{P}}(\cdot|x,a)}[V^{\pi_{\mathrm{PE}}}(X')] = (\mathcal{P}V^{\pi_{\mathrm{PE}}})(x,a)$.

We then use any arbitrary planning algorithm using the new dynamics $\bar{\mathcal{P}}$ instead of $\hat{\mathcal{P}}$, which means we solve MDP $\bar{M} = (\mathcal{X}, \mathcal{A}, \mathcal{R}, \bar{\mathcal{P}})$ instead of $\hat{M}$. Due to the constraint in finding $\bar{\mathcal{P}}$, we have $\bar{\mathcal{P}}V^{\pi_{\mathrm{PE}}} = \mathcal{P}V^{\pi_{\mathrm{PE}}}$, therefore $r^{\pi_{\mathrm{PE}}} + \gamma\bar{\mathcal{P}}^{\pi_{\mathrm{PE}}}V^{\pi_{\mathrm{PE}}} = r^{\pi_{\mathrm{PE}}} + \gamma\mathcal{P}^{\pi_{\mathrm{PE}}}V^{\pi_{\mathrm{PE}}} = V^{\pi_{\mathrm{PE}}}$. In other words, $V^{\pi_{\mathrm{PE}}}$ satisfies the Bellman equation in $\bar{M}$. This means that MaxEnt MoCo completely eliminates the impact of the model error on the agent, and we obtain the true value function $V^{\pi_{\mathrm{PE}}}$. The same argument can be made for the Control problem when we know $V^*$ and correction is performed via constraints given by $\mathcal{P}V^*$. The true optimal value function $V^*$ satisfies the Bellman optimality equation in $\bar{M}$, and it can consequently be shown that the optimal value function $\bar{V}^*$ and policy $\bar{\pi}^*$ in $\bar{M}$ match $V^*$ and $\pi^*$.

In practice, the true value functions $V^{\pi_{\mathrm{PE}}}$ or $V^*$ are unknown – we are trying to find them after all. In this case, we do the correction procedure with a set of *measurement functions* $\phi_1, \ldots, \phi_d$ with $\phi_i\colon \mathcal{X} \to \mathbb{R}$. The set of measurement functions can be chosen arbitrarily. As shall be clear later, we prefer to choose them such that their span can approximate the true value function $V^{\pi_{\mathrm{PE}}}$ or $V^*$ well. In this section and Section 3.2, we focus on the properties of model error correction for any given set of functions. In Sections 4 and 5, we will introduce techniques for finding a good set of such functions.

Now, we introduce the MaxEnt MoCo algorithm. In large or continuous MDPs, it is not feasible to perform MaxEnt density estimation for all $x, a$. Instead, we take a lazy computation approach and calculate $\bar{\mathcal{P}}(\cdot|x,a)$ only when needed. The dynamics $\bar{\mathcal{P}}\colon \mathcal{X} \times \mathcal{A} \to \mathcal{M}(\mathcal{X})$ is never constructed as a function of states and actions by the agent, and it is defined only for the purpose of analysis. First, we obtain $\mathcal{P}\phi_i$ for $i = 1, 2, \ldots, d$ through $d$ queries to the true dynamics $\mathcal{P}$. Then, we execute any planning algorithm that can normally be used in MBRL to solve the approximate MDP $\hat{M}$. The only modification is that whenever the planning algorithm uses the distribution $\hat{\mathcal{P}}(\cdot|x,a)$ for some state $x$ and action $a$, e.g. when simulating rollouts from $(x,a)$, we find a corrected distribution $\bar{\mathcal{P}}(\cdot|x,a)$ using MaxEnt density estimation and pass it to the planning algorithm instead of $\hat{\mathcal{P}}(\cdot|x,a)$ that would normally be used. The new distribution $\bar{\mathcal{P}}(\cdot|x,a)$ is given by

$$\bar{\mathcal{P}}(\cdot|x,a) \triangleq \underset{q\in\mathcal{M}(\mathcal{X})}{\arg\min} \quad D_{\mathrm{KL}}( q \parallel \hat{\mathcal{P}}(\cdot|x,a) ), \tag{P1}$$
$$\text{such that} \quad \mathbb{E}_{X'\sim q}[\phi_i(X')] = (\mathcal{P}\phi_i)(x,a) \qquad (i = 1, 2, \ldots, d).$$

As discussed in Section 3, the optimization problem (P1) can be solved through the respective convex dual problem as in (2.4). Also note that the dual problem only has $d$ parameters, which is usually small,[1] and solving it only involves $\hat{\mathcal{P}}$ that is considered cheap.

---

[1]For a reference, in our experiments $d \leq 3$. Even if $d$ is large, specialized algorithms have been developed to efficiently solve the optimization problem (Dudík et al., 2007).

We now analyze the performance of MaxEnt MoCo in PE. Let $\bar{V}^{\pi_{\mathrm{PE}}}$ be the value function of $\pi_{\mathrm{PE}}$ in MDP $\bar{M} = (\mathcal{X}, \mathcal{A}, \mathcal{R}, \bar{\mathcal{P}}, \gamma)$. We will show that the error of MaxEnt MoCo depends on how well $V^{\pi_{\mathrm{PE}}}$ can be approximated with a linear combination of the measurement functions. To see this, first note that the constraints in (P1) mean that $(\bar{\mathcal{P}}^{\pi_{\mathrm{PE}}} - \mathcal{P}^{\pi_{\mathrm{PE}}})\phi_i = 0$. Thus, for any $w \in \mathbb{R}^d$ we can write the upper bound on $\|V^{\pi_{\mathrm{PE}}} - \bar{V}^{\pi_{\mathrm{PE}}}\|_\infty$ that is given in Lemma 1 as

$$\frac{\gamma}{1-\gamma} \left\| (\mathcal{P}^{\pi_{\mathrm{PE}}} - \bar{\mathcal{P}}^{\pi_{\mathrm{PE}}}) V^{\pi_{\mathrm{PE}}} \right\|_\infty = \frac{\gamma}{1-\gamma} \left\| (\mathcal{P}^{\pi_{\mathrm{PE}}} - \bar{\mathcal{P}}^{\pi_{\mathrm{PE}}})(V^{\pi_{\mathrm{PE}}} - \sum_{i=1}^d w_i \phi_i) \right\|_\infty \tag{3.1}$$

$$\leq \frac{\sqrt{2}\gamma}{1-\gamma} \sup_{x,a} \sqrt{D_{\mathrm{KL}}(\, \mathcal{P}(\cdot|x,a) \,\|\, \bar{\mathcal{P}}(\cdot|x,a) \,)} \left\| V^{\pi_{\mathrm{PE}}} - \sum_{i=1}^d w_i \phi_i \right\|_\infty,$$

where the last inequality is proved similar to the proof of the second inequality in Lemma 1. Now, from the general Pythagoras theorem for KL-divergence (see Thm. 11.6.1 of Cover and Thomas 2006), for any $(x, a)$, we have

$$D_{\mathrm{KL}}(\, \mathcal{P}(\cdot|x,a) \,\|\, \bar{\mathcal{P}}(\cdot|x,a) \,) \leq D_{\mathrm{KL}}(\, \mathcal{P}(\cdot|x,a) \,\|\, \hat{\mathcal{P}}(\cdot|x,a) \,). \tag{3.2}$$

This inequality is of independent interest as it shows that MaxEnt MoCo is reducing the MLE loss of the model. It is worth mentioning that since $\bar{\mathcal{P}}$ is not constructed by the agent, this improved MLE loss can go beyond what is possible with the agent's model class. A feature that is valuable in complex environments that are hard to model. Inequalities (3.2) and (3.1) lead to an upper bound on $\|V^{\pi_{\mathrm{PE}}} - \bar{V}^{\pi_{\mathrm{PE}}}\|_\infty$. We have the following proposition:

**Proposition 1.** *Suppose that $\mathcal{P}$ is the true environment dynamics, $\hat{\mathcal{P}}$ is an approximation of $\mathcal{P}$, and $\epsilon_{\mathrm{Model}}$ is defined as in (2.1). Let $c_1 = \gamma\sqrt{2}/(1-\gamma)$ as in Lemma 1. Then,*

$$\left\| V^{\pi_{PE}} - \bar{V}^{\pi_{PE}} \right\|_\infty \leq c_1 \|\epsilon_{\mathrm{Model}}\|_\infty \inf_{w \in \mathbb{R}^d} \left\| V^{\pi_{PE}} - \sum_{i=1}^d w_i \phi_i \right\|_\infty,$$

$$\left\| V^* - V^{\bar{\pi}^*} \right\|_\infty \leq \frac{2c_1 \|\epsilon_{\mathrm{Model}}\|_\infty}{1 - c_1 \|\epsilon_{\mathrm{Model}}\|_\infty} \inf_{w \in \mathbb{R}^d} \left\| V^* - \sum_{i=1}^d w_i \phi_i \right\|_\infty.$$

The significance of this result becomes apparent upon comparison with Lemma 1. Whenever the value function can be represented sufficiently well within the span of the measurement functions $\{\phi_i\}$ used for correcting $\hat{\mathcal{P}}$, the error between the value function $\bar{V}^{\pi_{\mathrm{PE}}}$ of the modified dynamics $\bar{\mathcal{P}}$ compared to the true value function $V^{\pi_{\mathrm{PE}}}$ is significantly smaller than the error of the value function $\hat{V}^{\pi_{\mathrm{PE}}}$ obtained from $\hat{\mathcal{P}}$ — compare $\inf_{w \in \mathbb{R}^d} \|V^{\pi_{\mathrm{PE}}} - \sum_{i=1}^d w_i \phi_i\|_\infty$ with $\|V^{\pi_{\mathrm{PE}}}\|_\infty$.

## 3.2 Approximate Form

In the previous section, we assumed that the agent can obtain $\mathcal{P}\phi_i$ exactly. This is an unrealistic assumption when we only have access to samples from $\mathcal{P}$ such as in the RL setting. Estimating $\mathcal{P}\phi_i$ from samples is a regression problem and has error. We assume that we have access to the approximations $\psi_i \colon \mathcal{X} \times \mathcal{A} \to \mathbb{R}$ of $\mathcal{P}\phi_i$ such that $\psi_i \approx \mathcal{P}\phi_i$ with the error quantified by $\epsilon_{\mathrm{Query}}$. Specifically, for any $(x, a)$, we have $\epsilon_{\mathrm{Query}}(x,a) = \|\psi(x,a) - (\mathcal{P}\phi)(x,a)\|_2$ where $\phi \colon \mathcal{X} \to \mathbb{R}^d$ and $\psi \colon \mathcal{X} \times \mathcal{A} \to \mathbb{R}^d$ are the $d$-dimensional vectors formed by $\phi_i$ and $\psi_i$ functions.

When the observations are noisy, MaxEnt density estimation is prone to overfitting (Dudík et al., 2007). Many techniques have been introduced to alleviate this issue including regularization (Chen and Rosenfeld, 2000a; Lebanon and Lafferty, 2001), introduction of a prior (Goodman, 2004), and constraint relaxation (Kazama and Tsujii, 2003; Dudík et al., 2004). In this work, we use $\ell_2^2$ regularization (Lau, 1994; Chen and Rosenfeld, 2000b; Lebanon and Lafferty, 2001; Zhang, 2004; Dudík et al., 2007) and leave the study of the other approaches to future work.

The regularization is done by adding $\frac{1}{4}\beta^2 \|\lambda\|_2^2$ to the objective of the dual problem (2.4). This pushes the dual parameters to remain small. The hyperparameter $\beta$ controls the amount of regularization. Smaller $\beta$ leads a solution closer to the original one. Notice that with extreme regularization when $\beta \to \infty$, we get $\lambda = 0$, which makes the solution of MaxEnt density estimation the same as the

initial density estimate $\hat{p}$. The regularization of the dual problem has an intuitive interpretation in the primal problem. With the regularization, the primal problem (P1) is transformed to

$$\bar{\mathcal{P}}(\cdot|x,a) \triangleq \underset{q}{\operatorname{argmin}} \; D_{\mathrm{KL}}(\, q \,\|\, \hat{\mathcal{P}}(\cdot|x,a)\,) + \frac{1}{\beta^2}\sum_{i=1}^{d}\Big(\mathbb{E}_{X'\sim q}[\phi_i(X')] - \psi_i(x,a)\Big)^2. \tag{P2}$$

We now have introduced a new hyperparameter $\beta$ to MaxEnt MoCo. As $\beta \to 0$, the solution converges to that of the constrained problem (P1), because intuitively, $\beta$ controls how much we *trust* the noisy observations $\psi_i$. Smaller values of $\beta$ means that we care about being consistent with the queries more than staying close to $\hat{\mathcal{P}}$, and larger values of $\beta$ shows the opposite preference. It turns out the impact of the choice of $\beta$ is aligned with this intuition. As $\|\epsilon_{\mathrm{Model}}\|_\infty$ increases or $\|\epsilon_{\mathrm{Query}}\|_\infty$ decreases, we should rely on the queries more and choose a smaller $\beta$. We provide the analysis for a general choice of $\beta$ in the supplementary material, and here focus on when $\beta = \|\epsilon_{\mathrm{Query}}\|_\infty/\|\epsilon_{\mathrm{Model}}\|_\infty$.

**Theorem 1.** *Let $c_1 = \gamma\sqrt{2}/(1-\gamma)$, $c_2 = 3\gamma\sqrt{d}/(1-\gamma)$, and $\beta = \|\epsilon_{\mathrm{Query}}\|_\infty/\|\epsilon_{\mathrm{Model}}\|_\infty$. For any $w_{\max} \geq 0$, we have*

$$\left\|V^{\pi_{PE}} - \bar{V}^{\pi_{PE}}\right\|_\infty \leq 3c_1\|\epsilon_{\mathrm{Model}}\|_\infty \inf_{\|w\|_\infty \leq w_{\max}}\left\|V^{\pi_{PE}} - \sum_{i=1}^{d}w_i\phi_i\right\|_\infty + c_2\|\epsilon_{\mathrm{Query}}\|_\infty \cdot w_{\max},$$

$$\left\|V^* - V^{\bar{\pi}^*}\right\|_\infty \leq \frac{6c_1\|\epsilon_{\mathrm{Model}}\|_\infty}{1 - 3c_1\|\epsilon_{\mathrm{Model}}\|_\infty} \inf_{\|w\|_\infty \leq w_{\max}}\left\|V^* - \sum_{i=1}^{d}w_i\phi_i\right\|_\infty + \frac{2c_2\|\epsilon_{\mathrm{Query}}\|_\infty}{1 - 3c_1\|\epsilon_{\mathrm{Model}}\|_\infty} \cdot w_{\max}.$$

The above theorem shows that the error in the queries contribute an additive term to the final bounds compared to the exact query setting analyzed in Proposition 1. This term scales with $w_{\max}$, which can be chosen arbitrarily to minimize the upper bound. Larger values of $w_{\max}$ allow a better approximation of $V^{\pi_{PE}}$ and $V^*$ in the infimum terms, but amplify the query error $\epsilon_{\mathrm{Query}}$. Thus, if $V^{\pi_{PE}}$ (or $V^*$) can be approximated by some weighted sum of the measurement functions using smaller weights, $w_{\max}$ can be chosen to be smaller. Unlike the exact case discussed in Proposition 1, the choice of measurement functions is important beyond the subspace generated by their span. Therefore, transformations of the measurement functions such as centralization, normalization, or orthogonalization might improve the effectiveness of MaxEnt Model Correction.

One limitation of the results of Theorem 1 is that they depend on the $\ell_\infty$ norm of $\epsilon_{\mathrm{Model}}$ and $\epsilon_{\mathrm{Query}}$. However, if the functions $\hat{\mathcal{P}}$ and $\psi_i$ are estimated with function approximation, their error is generally controlled in some weighted $\ell_p$ norm. Thus, error analysis of RL algorithms in weighted $\ell_p$ norm is essential and has been the subject of many studies (Munos, 2003; 2007; Farahmand et al., 2010; Scherrer et al., 2015). We do provide this analysis for MaxEnt MoCo, but to keep the main body of the paper short and simple, we defer them to the supplementary material.

## 4 Model Correcting Value Iteration

In the previous section, we introduced MaxEnt model correction for a given set of measurement functions $\phi_1, \ldots, \phi_d$. We saw that a good set of functions is one that for some $w \in \mathbb{R}^d$, the true value function $V^{\pi_{PE}}$ or $V^*$ is well approximated by $\sum_i w_i\phi_i$. In this section, we introduce the Model Correcting Value Iteration (MoCoVI) algorithm that iteratively finds increasingly better measurement functions. We show that if the model is accurate enough, MoCoVI can utilize the approximate model to converge to the true value function despite the model error, and do so with a better convergence rate than the conventional VI. Since MoCoVI calls the MaxEnt MoCo procedure iteratively, we introduce a notation for it. If $\bar{\mathcal{P}}$ is the corrected dynamics based on the set of measurement functions $\Phi$ and their query results $\Psi$, and $\bar{V}^{\pi_{PE}}, \bar{V}^*, \bar{\pi}^*$ are the respective $V^{\pi_{PE}}, V^*, \pi^*$ in $\bar{M} = (\mathcal{X}, \mathcal{A}, \mathcal{R}, \bar{\mathcal{P}})$, we define $\mathrm{MoCo}_\beta^{\pi_{PE}}(\mathcal{R}, \hat{\mathcal{P}}, \Phi, \Psi) \triangleq \bar{V}^{\pi_{PE}}$ and $\mathrm{MoCo}_\beta^*(\mathcal{R}, \hat{\mathcal{P}}, \Phi, \Psi) \triangleq (\bar{V}^*, \bar{\pi}^*)$ to be the solution of PE and Control problems obtained with MaxEnt MoCo.

To start with, consider the PE problem and assume that we can make exact queries to $\mathcal{P}$. We set $\phi_1, \ldots, \phi_d \colon \mathcal{X} \to \mathbb{R}$ to be an arbitrary initial set of measurement functions, with query results $\psi_i = \mathcal{P}\phi_i$ for $1 \leq i \leq d$. We perform the MaxEnt MoCo procedure using $\phi_{1:d}$ and $\psi_{1:d}$ to obtain

$V_0 = \mathrm{MoCo}_\beta^{\pi_{\mathrm{PE}}}(\mathcal{R}, \hat{\mathcal{P}}, \phi_{1:d}, \psi_{1:d})$. In the next iteration, we set $\phi_{d+1} = V_0$.[2] Then, we query $\mathcal{P}$ at $\phi_{d+1}$ to obtain $\psi_{d+1} = \mathcal{P}\phi_{d+1}$. By executing MaxEnt MoCo with the last $d$ queries, we arrive at $V_1 = \mathrm{MoCo}_\beta^{\pi_{\mathrm{PE}}}(\mathcal{R}, \hat{\mathcal{P}}, \phi_{2:d+1}, \psi_{2:d+1})$. We can use Proposition 1 to bound the error of $V_1$.

$$\|V^{\pi_{\mathrm{PE}}} - V_1\|_\infty \leq \frac{\gamma\sqrt{2}}{1-\gamma} \cdot \|\epsilon_{\mathrm{Model}}\|_\infty \cdot \frac{\inf_{w \in \mathbb{R}^d}\left\|V^{\pi_{\mathrm{PE}}} - \sum_{i=1}^d w_i\phi_{1+i}\right\|_\infty}{\|V^{\pi_{\mathrm{PE}}} - V_0\|_\infty} \cdot \|V^{\pi_{\mathrm{PE}}} - V_0\|_\infty$$

As $\sum_{i=1}^d w_i\phi_{1+i}$ is equal to $V_0$ with the choice of $w_{1:d-1} = 0$ and $w_d = 1$, the fraction above is less than or equal to 1. Generally, the fraction gets smaller with larger $d$ and better measurement function, leading to a more accurate $V_1$. If the model is accurate enough, the new value function $V_1$ is a more accurate approximation of $V^{\pi_{\mathrm{PE}}}$ than the initial $V_0$. By repeating this procedure, we may converge to the true value function $V^{\pi_{\mathrm{PE}}}$.

We now introduce MoCoVI based on the above idea. We start with an initial set of measurement functions $\phi_1, \ldots, \phi_d$ and their query results $\psi_1, \ldots, \psi_d$ such that $\psi_i \approx \mathcal{P}\phi_i$ for $1 \leq i \leq d$. At each iteration $k \geq 0$, we execute MaxEnt MoCo with $\phi_{k+1:k+d}$ and $\psi_{k+1:k+d}$ to obtain $V_k$ (and $\pi_k$). In the end, we set $\phi_{k+d+1} = V_k$ and query $\mathcal{P}$ to get the new query result. That is, for any $k \geq 0$

$$\begin{cases} V_k = \mathrm{MoCo}_\beta^{\pi_{\mathrm{PE}}}(\mathcal{R}, \hat{\mathcal{P}}, \phi_{k+1:k+d}, \psi_{k+1:k+d}) \quad \text{or} \quad V_k, \pi_k = \mathrm{MoCo}_\beta^*(\mathcal{R}, \hat{\mathcal{P}}, \phi_{k+1:k+d}, \psi_{k+1:k+d}), \\ \phi_{k+d+1} = V_k \ , \ \psi_{k+d+1} \approx \mathcal{P}\phi_{k+d+1}. \end{cases}$$

The choice of value functions can be motivated from two viewpoints. First, it has been suggested that features learned to represent the past value function may be useful to represent the true value functions as well (Dabney et al., 2021). This suggests that the true value function may be approximated with the span of the past value functions. A property shown to be useful in Theorem 2. Second, this choice means that the corrected transition dynamics $\bar{\mathcal{P}}$ at iteration $k$ will satisfy $\bar{\mathcal{P}}V_{k-i} \approx \mathcal{P}V_{k-i}$ for $i = 1, 2, \ldots, d$. This property has been recognized to be valuable for the dynamics that is used for planning in MBRL, and implemented in value-aware model learning losses (Farahmand et al., 2017; Farahmand, 2018; Abachi et al., 2020; Voelcker et al., 2022; Abachi et al., 2022). However, practical implementations of these losses has been shown to be challenging (Voelcker et al., 2022; Lovatto et al., 2020). In comparison, MoCoVI works with any model learning approach and creates this property through MaxEnt density estimation. The next theorem provides convergence result of MoCoVI in supremum norm based on the analysis in Theorem 1.

**Theorem 2.** *Let $K \geq 1$. Assume $\epsilon_{\mathrm{Query}}^\infty(x, a) = \sqrt{d} \cdot \sup_{i \geq 0}|(\mathcal{P}\phi_i)(x, a) - \psi_i(x, a)|$ and $\beta = \|\epsilon_{\mathrm{Query}}\|_\infty / \|\epsilon_{\mathrm{Model}}\|_\infty$. Let $c_1, c_2$ be as in Theorem 1 and $w_{\max} \geq 1$. Define $V^{\mathrm{target}} = V^{\pi_{\mathrm{PE}}}$ for PE and $V^{\mathrm{target}} = V^*$ for Control. Finally, let*

$$\gamma' = 3c_1\|\epsilon_{\mathrm{Model}}\|_\infty \cdot \max_{1 \leq k \leq K} \frac{\inf_{\|w\|_\infty \leq w_{\max}}\|V^{\mathrm{target}} - \sum_{i=1}^d w_i\phi_{k+i}\|_\infty}{\|V^{\mathrm{target}} - V_{k-1}\|_\infty}.$$

*We have*

$$\|V^{\pi_{\mathrm{PE}}} - V_K\|_\infty \leq \gamma'^K\|V^{\pi_{\mathrm{PE}}} - V_0\|_\infty + \frac{1 - \gamma'^K}{1 - \gamma'}c_2\|\epsilon_{\mathrm{Query}}^\infty\|_\infty w_{\max},$$

$$\|V^* - V^{\pi_K}\|_\infty \leq \frac{2\gamma'^K}{1 - 3c_1\|\epsilon_{\mathrm{Model}}\|_\infty} \cdot \|V^* - V_0\|_\infty + \frac{1 - \gamma'^K}{1 - \gamma'}\frac{2c_2\|\epsilon_{\mathrm{Query}}^\infty\|_\infty}{1 - 3c_1\|\epsilon_{\mathrm{Model}}\|_\infty}w_{\max}.$$

This result should be compared with the convergence analysis of approximate VI. Notice that both MoCoVI and VI query $\mathcal{P}$ once per iteration, which makes this comparison fair. According to Munos (2007), $\|V^* - V^{\pi_K}\|_\infty$ for VI is bounded by $\frac{2\gamma^K}{1-\gamma}\|V^* - V_0\|_\infty + \frac{2\gamma(1-\gamma^{K-1})}{(1-\gamma)^2}\|\epsilon_{\mathrm{Query}}\|_\infty$. Here we considered the error in applying the Bellman operator equal to the query error. In VI, the initial error $\|V^* - V_0\|_\infty$ decreases with the rate $\mathcal{O}(\gamma^K)$. In comparison, for MoCoVI, the initial error decreases with the rate $\mathcal{O}(\gamma'^k)$. While the convergence rate of VI is tied to the fixed parameter $\gamma$ and become undesirable if $\gamma$ is close to 1, the rate of MoCoVI improves with more accurate models. Consequently, the convergence rate of MoCoVI can be much faster than VI if the model is accurate enough.

---

[2]According to the discussion after Theorem 1, it might be beneficial to set $\phi_{d+1}$ to some linear transformations of $V_0$ in presence of query error. For the sake of simplicity of the results, we don't consider such operations.

---

**Algorithm 1** MoCoDyna($T, d, c, \beta, K$)

---

1: Initialize $\phi_1, \ldots, \phi_{d+c}, \psi_1, \ldots, \psi_{d+c}$, and $\hat{\mathcal{P}}, \hat{r}$.
2: **for** $t = 1, 2, \ldots, T$ **do**
3:      Sample $X_t, A_t, R_t, X_t'$ from the environment.
4:      $\hat{r}, \hat{\mathcal{P}} \leftarrow \text{Update}(\hat{r}, \hat{\mathcal{P}}, X_t, A_t, R_t, X_t')$
5:      $\psi_{1:d+c} \leftarrow \text{Update}(\psi_{1:d+c}, X_t, A_t, X_t')$
6:      $V_t \leftarrow \text{MoCo}_\beta^{\pi_{\text{PE}}}(\hat{r}, \hat{\mathcal{P}}, \phi_{1:d}, \psi_{1:d})$    or    $V_t, \pi_t \leftarrow \text{MoCo}_\beta^*(\hat{r}, \hat{\mathcal{P}}, \phi_{1:d}, \psi_{1:d})$,
7:      **if** $t \mod K = 0$ **then**
8:          Pop $\phi_1, \psi_1$
9:          $\phi_{d+c} \leftarrow \text{MeasurementCreation}(V_t, \phi_{1:d+c-1})$     ,     $\psi_{d+c}(x, a) \leftarrow 0$

---

A closely comparable algorithm to MoCoVI is OS-VI (Rakhsha et al., 2022). OS-VI also does solve a new MDP at each iteration, but instead of changing the transition dynamics, changes the reward function. The convergence rate of OS-VI, when stated in terms of our $\epsilon_{\text{Model}}$ using Pinsker's inequality, is $c_1 \|\epsilon_{\text{Model}}\|_\infty$. In comparison, $\gamma'$ can become much smaller if the past value functions can approximate the true value function well or if $d$ is increased. Moreover, OS-VI can diverge if the model is too inaccurate, but even if $\gamma' > 1$, the bound given in Theorem 1 still holds for $V_k$ for all $k$, which means MoCoVI does not diverge.

## 5 MODEL CORRECTING DYNA

We extend MoCoVI to the sample-based setting where only samples from the true dynamics $\mathcal{P}$ are available. The key challenge is that we can no longer obtain $\psi_k$ from $\phi_k$ by a single query. Instead, we should form an estimate of $\mathcal{P}\phi_k$ using the samples. In general, this is a regression task that is studied in supervised learning. In algorithms that a replay buffer of transitions $(X_i, A_i, R_i, X_i')_{i=1}^N$ is stored, the regression can be done with $(X_i, A_i)$ as the input and $\phi_k(X_i')$ as the target. In this paper, we present a version of the algorithm based on stochastic approximation, but we emphasize that the algorithm can be extended to use function approximation without any fundamental barriers.

An overview of MoCoDyna for finite MDPs is given in Algorithm 1. For some integer $c \geq 0$, we keep $d + c$ measurement functions $\phi_1, \ldots, \phi_{d+c}$. As explained later, this set of functions is updated similar to MoCoVI: the oldest function is regularly substituted with the current value function. A set of approximate query results $\psi_1, \ldots, \psi_{d+c}$ for the measurement functions is also maintained. That is, we will have $\psi_i \approx \mathcal{P}\phi_i$ for each $i$ via stochastic approximation. At each step, we get a sample $(X_t, A_t, R_t, X_t')$ from the environment. We update $\psi_i(X_t, A_t)$ for $i = 1, \ldots, d + c$ by $\psi_i(X_t, A_t) \leftarrow \psi_i(X_t, A_t) + \frac{1}{N_i(X_t, A_t)}(\phi_i(X_t') - \psi_i(X_t, A_t))$. Here, $N_i(X_t, A_t)$ is the number of times $(X_t, A_t)$ has been visited since the function $\phi_i$ has been added to the set of measurement functions. At every step, the agent also updates its approximate model $\hat{r}, \hat{\mathcal{P}}$ with $(X_t, A_t, R_t, X_t')$.

At each iteration, MoCoDyna runs the MaxEnt MoCo procedure to obtain the new value function and policy. That is, the agent uses an arbitrary planning algorithm to solve the PE or control problem with rewards $\hat{r}$ and the dynamics obtained by correcting $\hat{\mathcal{P}}$. The correction only uses the $d$ oldest measurement functions among the $d + c$ functions. The reason is that for a measurement function $\phi$ that has been added to the set recently, the agent has not had enough samples to form an accurate approximation of $\mathcal{P}\phi_i$. Finally, every $K$ steps, the agent updates its set of measurement functions. The oldest function $\phi_1$ is removed along with $\psi_1$. The new measurement function $\phi_{d+c}$ is chosen such that $V_t$ belongs to span of $\phi_{1:d+c}$. In the simplest form, we can set $\phi_{d+c} = V_t$, but as discussed after Theorem 1 some linear transformations might be beneficial. We allow this transformation by defining $\phi_{d+c} \leftarrow \text{MeasurementCreation}(V_t, \phi_{1:d+c-1})$.

## 6 NUMERICAL EXPERIMENTS

We empirically show the effectiveness of MoCoVI and MoCoDyna to utilize an approximate model. We consider the $6 \times 6$ grid world environment with four actions introduced by Rakhsha et al. (2022), with $\gamma = 0.9$. We defer the details of the environment to the supplementary material. As shown in Theorem 2, the convergence rate of MoCoVI depends on the model error and $d$. We introduce error

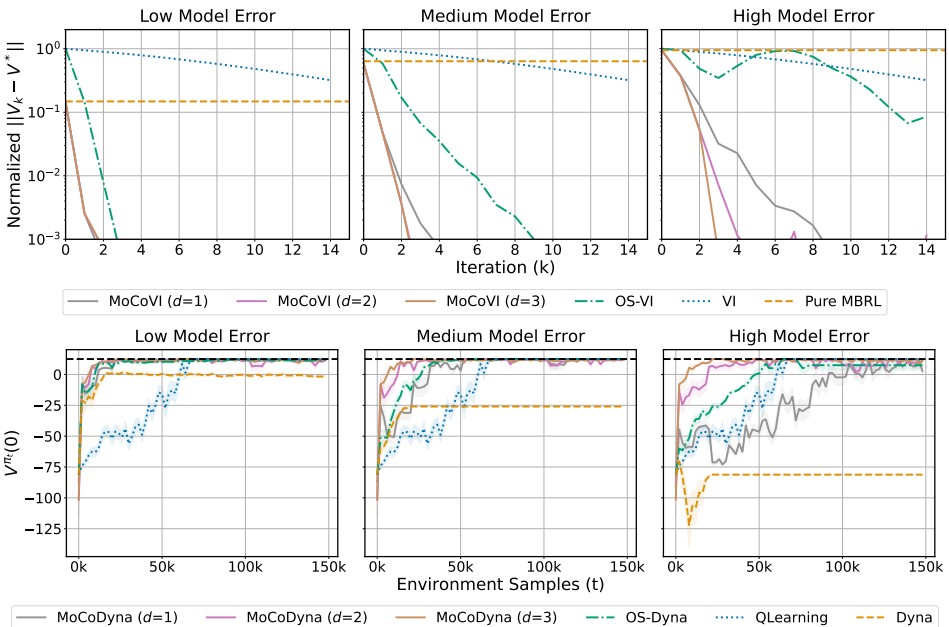

Figure 1: Comparison of **(top)** MoCoVI with VI, pure MBRL and OS-VI, and **(bottom)** MoCoDyna with QLearning, Dyna, and OS-Dyna. *(Left)* low ($\lambda = 0.1$), *(Middle)* medium ($\lambda = 0.5$), and *(Right)* high ($\lambda = 1$) model errors. Each curve is average of 20 runs. Shaded areas show the standard error.

to $\hat{\mathcal{P}}$ by smoothing the true dynamics $\mathcal{P}$ as suggested by Rakhsha et al. (2022): for $\lambda \in [0, 1]$, the smoothed dynamics $\mathcal{P}^{(\lambda)}$ is $\mathcal{P}^{(\lambda)}(\cdot|x, a) \triangleq (1 - \lambda) \cdot \mathcal{P}(\cdot|x, a) + \lambda \cdot U(\{x'|\mathcal{P}(x'|x, a) > 0\})$, where $U(S)$ is the uniform distribution over set $S$. The parameter $\lambda$ controls the model error, from no error with $\lambda = 0$ to a large error with $\lambda = 1$ (uniform transition probability over possible next-states).

Fig. 1 first compares MoCoVI with OS-VI (Rakhsha et al., 2022), VI, and the value function obtained based on the model. We set $\hat{\mathcal{P}} = \mathcal{P}^{(\lambda)}$ for $\lambda = 0.1, 0.5$ and 1. The plot shows normalized error of $V_k$ against $V^*$, that is, $\|V_k - V^*\|_1/\|V^*\|_1$. MoCoVI can converge to the true value function in a few iterations even with extreme model errors. The robustness, as expected, is improved with larger values of $d$. In comparison, OS-VI and VI show a much slower rate than MoCoVI and the value function obtained from $\hat{\mathcal{P}}$ suffers from the model error. Fig. 1 then shows the results in the RL setting. We compare MoCoDyna with OS-Dyna (Rakhsha et al., 2022), QLearning, and Dyna. At each step, the algorithms are given a sample $(X_t, A_t, R_t, X'_t)$ where $X_t, A_t$ are chosen uniformly in random. We use $\hat{\mathcal{P}} = \mathcal{P}^{(\lambda)}_{\text{MLE}}$ where $\mathcal{P}_{\text{MLE}}$ is the MLE estimate of dynamics at the moment. For OS-Dyna and QLearning which have a learning rate, for some $\alpha, N > 0$, we use the constant learning $\alpha$ for $t \leq N$ and $\alpha/(t - N)$ for $t > N$ to allow both fast initial convergence and stability. The results show a similar pattern as for MoCoVI. MoCoDyna can successfully solve the task with any model error. In fact, MoCoDyna significantly outperforms other algorithms. In comparison, QLearning and OS-Dyna show a slower rate of convergence, and Dyna cannot solve the task due to the model error.

# 7 CONCLUSION

In this work, we set out to bridge model-based and model-free approaches in RL by devising a cost-efficient approach to alleviate model errors. We develop the MaxEnt model correction framework, which adopts MaxEnt density estimation to reduce model errors given a small number of queries to the true dynamics. A thorough theoretical analysis indicates that our framework can significantly accelerate the convergence rate of policy evaluation and control algorithms, and ensure convergence to the true value functions despite model errors if said errors are sufficiently small. We also develop a sample-based variant, MoCoDyna, which extends the Dyna framework. Lastly, we confirm the practical relevance of our theoretical findings by benchmarking MoCo-based planning algorithms against their naive counterparts, and showing superior performance both in terms of convergence rate and expected returns. Future work should investigate deep RL applications of the MoCo framework.

ACKNOWLEDGMENTS

We would like to thank the members of the Adaptive Agents Lab, especially Claas Voelcker, who provided feedback on a draft of this paper. AMF acknowledges the funding from the Canada CIFAR AI Chairs program, as well as the support of the Natural Sciences and Engineering Research Council of Canada (NSERC) through the Discovery Grant program (2021-03701). MK acknowledges the support of NSERC via the Canada Graduate Scholarship - Doctoral program (CGSD3-568998-2022). Resources used in preparing this research were provided, in part, by the Province of Ontario, the Government of Canada through CIFAR, and companies sponsoring the Vector Institute.

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

# A    LIST OF APPENDICES

# B    BACKGROUND ON MARKOV DECISION PROCESSES

In this work, we consider a discounted Markov Decision Process (MDP) defined as $M = (\mathcal{X}, \mathcal{A}, \mathcal{R}, \mathcal{P}, \gamma)$ (Bertsekas and Tsitsiklis, 1996; Szepesvári, 2010; Sutton and Barto, 2019). Here, $\mathcal{X}$ is the state space, $\mathcal{A}$ is the action space, $\mathcal{R} \colon \mathcal{X} \times \mathcal{A} \to \mathcal{M}(\mathbb{R})$ is the reward kernel, $\mathcal{P} \colon \mathcal{X} \times \mathcal{A} \to \mathcal{M}(\mathcal{X})$ is the transition kernel, and $0 \leq \gamma < 1$ is the discount factor.[3] We define $r \colon \mathcal{X} \times \mathcal{A} \to \mathbb{R}$ to be the expected reward and assume it is known to the agent. A policy $\pi \colon \mathcal{X} \to \mathcal{M}(\mathcal{A})$ is a mapping from states to distributions over actions. We denote the expected rewards and transitions of a policy $\pi$ by $r^\pi \colon \mathcal{X} \to \mathbb{R}$ and $\mathcal{P}^\pi \colon \mathcal{X} \to \mathcal{M}(\mathcal{X})$, respectively. For any function $\phi \colon \mathcal{X} \to \mathbb{R}$, we define $\mathcal{P}\phi \colon \mathcal{X} \times \mathcal{A} \to \mathbb{R}$ as

$$(\mathcal{P}\phi)(x, a) \triangleq \int \mathcal{P}(\mathrm{d}x'|x, a)\phi(x') \qquad (\forall x, a).$$

The value function $V^\pi = V^\pi(\mathcal{R}, \mathcal{P})$ of a policy $\pi$ is defined as

$$V^\pi(x) \triangleq \mathbb{E}\left[\sum_{t=0}^{\infty} \gamma^t R_t | X_0 = x\right],$$

where actions are taken according to $\pi$, and $X_t$ and $R_t$ are the state and reward at step $t$. The value function of $\pi$ satisfies the Bellman equation: For all $x \in \mathcal{X}$, we have

$$V^\pi(x) = r^\pi(x) + \gamma \int \mathcal{P}^\pi(\mathrm{d}x'|x)V^\pi(x'), \tag{B.1}$$

or in short, $V^\pi = r^\pi + \gamma \mathcal{P}^\pi V^\pi$. The optimal value function $V^* = V^*(\mathcal{R}, \mathcal{P})$ is defined such that $V^*(x) = \max_\pi V^\pi(x)$ for all states $x \in \mathcal{X}$. Similarly, $V^*$ satisfies the Bellman optimality equation:

$$V^*(x) = \max_{a \in \mathcal{A}} \left\{ r(x, a) + \gamma \int \mathcal{P}(\mathrm{d}x'|x, a)V^*(x') \right\}. \tag{B.2}$$

We denote an optimal policy by $\pi^* = \pi^*(\mathcal{P}, \mathcal{R})$, for which we have $V^* = V^{\pi^*}$. We refer to the problem of finding $V^{\pi_{\mathrm{PE}}}$ for a specific policy $\pi_{\mathrm{PE}}$ as the *Policy Evaluation (PE)* problem, and to the problem of finding an optimal policy as the *Control* problem.

The *greedy* policy at state $x \in \mathcal{X}$ is

$$\pi_g(x; V) \leftarrow \operatorname*{argmax}_{a \in \mathcal{A}} \left\{ r(x, a) + \gamma \int \mathcal{P}(\mathrm{d}y|x, a)V(y) \right\}. \tag{B.3}$$

In this paper, we assume an approximate model $\hat{\mathcal{P}} \approx \mathcal{P}$ is given. We define $\hat{V}^\pi$ and $\hat{\pi}^*$ in the approximate MDP $\hat{M} = (\mathcal{X}, \mathcal{A}, \mathcal{R}, \hat{\mathcal{P}}, \gamma)$ similar to their counterparts in the true MDP $M$.

---

[3]For a domain $S$, we denote the space of all distributions over $S$ by $\mathcal{M}(S)$.

## C  PROOFS FOR SECTION 2.1

In this section, we provide the proof of Lemma 1. Before that, we first show two useful lemmas.

**Lemma 2.** *For two transition dynamics $\mathcal{P}_1, \mathcal{P}_2$ and any policy $\pi$ we have*

$$\|\mathcal{P}_1^\pi(\cdot|x) - \mathcal{P}_2^\pi(\cdot|x)\|_1 \leq \sqrt{2} \int \pi(\mathrm{d}a|x)\sqrt{D_{\mathrm{KL}}(\,\mathcal{P}_1(\cdot|x,a)\parallel\mathcal{P}_2(\cdot|x,a)\,)}$$

**Proof.** We have

$$
\begin{aligned}
\|\mathcal{P}_1^\pi(\cdot|x) - \mathcal{P}_2^\pi(\cdot|x)\|_1 &= \int_y |\mathcal{P}_1^\pi(\mathrm{d}y|x) - \mathcal{P}_2^\pi(\mathrm{d}y|x)| \\
&= \int_y \left| \int_a \pi(\mathrm{d}a, x)(\mathcal{P}_1(\mathrm{d}y|x,a) - \mathcal{P}_2(\mathrm{d}y|x,a)) \right| \\
&\leq \int_y \int_a \pi(\mathrm{d}a, x)|\mathcal{P}_1(\mathrm{d}y|x,a) - \mathcal{P}_2(\mathrm{d}y|x,a)| \\
&= \int_a \pi(\mathrm{d}a, x) \int_y |\mathcal{P}_1(\mathrm{d}y|x,a) - \mathcal{P}_2(\mathrm{d}y|x,a)| \\
&= \int_a \pi(\mathrm{d}a, x)\|\mathcal{P}_1(\cdot|x,a) - \mathcal{P}_2(\cdot|x,a)\|_1 \qquad\qquad\text{(C.1)}\\
&\leq \int_a \pi(\mathrm{d}a, x)\sqrt{2D_{\mathrm{KL}}(\,\mathcal{P}_1(\cdot|x,a)\parallel\mathcal{P}_2(\cdot|x,a)\,)}
\end{aligned}
$$

where we used the Pinsker's inequality. $\qquad\square$

**Lemma 3.** *For two transition dynamics $\mathcal{P}_1, \mathcal{P}_2$ and any policy $\pi$, Define*

$$G_{\mathcal{P}_1,\mathcal{P}_2}^\pi \triangleq (\mathbf{I} - \gamma\mathcal{P}_2^\pi)^{-1}(\gamma\mathcal{P}_1^\pi - \mathcal{P}_2^\pi)$$

*We have*

$$\left\|G_{\mathcal{P}_1^\pi,\mathcal{P}_2^\pi}^\pi\right\|_\infty \leq \frac{\gamma\sqrt{2}}{1-\gamma}\sup_{x,a}\sqrt{D_{\mathrm{KL}}(\,\mathcal{P}_1(\cdot|x,a)\parallel\mathcal{P}_2(\cdot|x,a)\,)}$$

**Proof.** When clear from context, we write $G^\pi$ instead of $G_{\mathcal{P}_1,\mathcal{P}_2}^\pi$. We have

$$
\begin{aligned}
\|G^\pi\|_\infty &\leq \gamma\left\|(\mathbf{I} - \gamma\mathcal{P}_2^\pi)^{-1}\right\|_\infty\|\mathcal{P}_1^\pi - \mathcal{P}_2^\pi\|_\infty \\
&\leq \frac{\gamma}{1-\gamma}\sup_x\|\mathcal{P}_1^\pi(\cdot|x) - \mathcal{P}_2^\pi(\cdot|x)\|_1
\end{aligned}
$$

where we used $\left\|(\mathbf{I} - A)^{-1} = 1/(1 - \|A\|_\infty)\right\|_\infty$ for $\|A\|_\infty \leq 1$ and the fact that $\|\mathcal{P}_2^\pi\|_\infty = 1$. Due to Lemma 2 we have for any $x$

$$
\begin{aligned}
\|\mathcal{P}_1^\pi(\cdot|x) - \mathcal{P}_2^\pi(\cdot|x)\|_1 &\leq \sqrt{2}\int \pi(\mathrm{d}a|x)\sqrt{D_{\mathrm{KL}}(\,\mathcal{P}_1(\cdot|x,a)\parallel\mathcal{P}_2(\cdot|x,a)\,)} \\
&\leq \sup_{x,a}\sqrt{2D_{\mathrm{KL}}(\,\mathcal{P}_1(\cdot|x,a)\parallel\mathcal{P}_2(\cdot|x,a)\,)}
\end{aligned}
$$

substituting this in the bound for $\|G^\pi\|_\infty$ gives the result. $\qquad\square$

We now give the proof of Lemma 1.

**Proof of Lemma 1 for PE**

**Proof.** Since $\hat{V}^\pi = (\mathbf{I} - \gamma\hat{\mathcal{P}}^\pi)^{-1}r^\pi$ and $r^\pi = (\mathbf{I} - \gamma\mathcal{P}^\pi)V^\pi$ we have

$$
\begin{aligned}
V^\pi - \hat{V}^\pi &= (\mathbf{I} - \gamma\hat{\mathcal{P}}^\pi)^{-1}(\mathbf{I} - \gamma\hat{\mathcal{P}}^\pi)V^\pi - (\mathbf{I} - \gamma\hat{\mathcal{P}}^\pi)^{-1}r^\pi \\
&= (\mathbf{I} - \gamma\hat{\mathcal{P}}^\pi)^{-1}[(\mathbf{I} - \gamma\hat{\mathcal{P}}^\pi)V^\pi - (\mathbf{I} - \gamma\mathcal{P}^\pi)V^\pi] \\
&= (\mathbf{I} - \gamma\hat{\mathcal{P}}^\pi)^{-1}(\gamma\mathcal{P}^\pi - \gamma\hat{\mathcal{P}}^\pi)V^\pi \\
&= G_{\mathcal{P},\hat{\mathcal{P}}}^\pi V^\pi \qquad\qquad\qquad\qquad\qquad\qquad\text{(C.2)}
\end{aligned}
$$

Thus,

$$
\begin{aligned}
\left\|V^{\pi} - \hat{V}^{\pi}\right\|_{\infty} &\leq \gamma \left\|(\mathbf{I} - \gamma\hat{\mathcal{P}}^{\pi})^{-1}\right\|_{\infty} \left\|(\mathcal{P}^{\pi} - \hat{\mathcal{P}}^{\pi})V^{\pi}\right\|_{\infty} \\
&\leq \frac{\gamma}{1-\gamma} \left\|(\mathcal{P}^{\pi} - \hat{\mathcal{P}}^{\pi})V^{\pi}\right\|_{\infty} \\
&\leq \frac{\gamma}{1-\gamma} \left\|(\mathcal{P}^{\pi} - \hat{\mathcal{P}}^{\pi})\right\|_{\infty} \|V^{\pi}\|_{\infty} \\
&\leq \frac{\gamma\sqrt{2}}{1-\gamma} \|\epsilon_{\text{Model}}\|_{\infty} \|V^{\pi}\|_{\infty}
\end{aligned}
$$

where we followed the proof of Lemma 3 for the last inequality. $\qquad \square$

**Proof of Lemma 1 for Control**

**Proof.** Define $r_0 = r + (\gamma\mathcal{P} - \gamma\hat{\mathcal{P}})V^*$ similar to Rakhsha et al. (2022, Lemma 3)

$$
V^{\pi^*}(r_0, \hat{\mathcal{P}}) = V^*(r, \mathcal{P}).
$$

Assume $f \preccurlyeq g$ mean $f(x) \geq g(x)$ for any $x$. We can write

$$
\begin{aligned}
0 &\preccurlyeq V^{\pi^*}(r, \mathcal{P}) - V^{\hat{\pi}^*}(r, \mathcal{P}) \\
&= V^{\pi^*}(r_0, \hat{\mathcal{P}}) - V^{\hat{\pi}^*}(r, \mathcal{P}) \\
&= V^{\pi^*}(r_0, \hat{\mathcal{P}}) - V^{\pi^*}(r, \hat{\mathcal{P}}) + V^{\pi^*}(r, \hat{\mathcal{P}}) - V^{\hat{\pi}^*}(r, \mathcal{P}) \\
&\preccurlyeq V^{\pi^*}(r_0, \hat{\mathcal{P}}) - V^{\pi^*}(r, \hat{\mathcal{P}}) + V^{\hat{\pi}^*}(r, \hat{\mathcal{P}}) - V^{\hat{\pi}^*}(r, \mathcal{P}) \\
&= (\mathbf{I} - \gamma\hat{\mathcal{P}}^{\pi^*})^{-1}(r_0^{\pi^*} - r^{\pi^*}) + V^{\hat{\pi}^*}(r, \hat{\mathcal{P}}) - V^{\hat{\pi}^*}(r, \mathcal{P}) \\
&= (\mathbf{I} - \gamma\hat{\mathcal{P}}^{\pi^*})^{-1}(\gamma\mathcal{P}^{\pi^*} - \gamma\hat{\mathcal{P}}^{\pi^*})V^* + V^{\hat{\pi}^*}(r, \hat{\mathcal{P}}) - V^{\hat{\pi}^*}(r, \mathcal{P}) \\
&= G_{\mathcal{P},\hat{\mathcal{P}}}^{\pi^*}V^* + V^{\hat{\pi}^*}(r, \hat{\mathcal{P}}) - V^{\hat{\pi}^*}(r, \mathcal{P}) \\
&= G_{\mathcal{P},\hat{\mathcal{P}}}^{\pi^*}V^* - G_{\mathcal{P},\hat{\mathcal{P}}}^{\hat{\pi}^*}V^{\hat{\pi}^*}(r, \mathcal{P}) \\
&= G_{\mathcal{P},\hat{\mathcal{P}}}^{\pi^*}V^* - G_{\mathcal{P},\hat{\mathcal{P}}}^{\hat{\pi}^*}V^* + G_{\mathcal{P},\hat{\mathcal{P}}}^{\hat{\pi}^*}(V^* - V^{\hat{\pi}^*}(r, \mathcal{P})) \\
&\preccurlyeq \left|G_{\mathcal{P},\hat{\mathcal{P}}}^{\pi^*}V^*\right| + \left|G_{\mathcal{P},\hat{\mathcal{P}}}^{\hat{\pi}^*}V^*\right| + \left|G_{\mathcal{P},\hat{\mathcal{P}}}^{\hat{\pi}^*}(V^* - V^{\hat{\pi}^*}(r, \mathcal{P}))\right|
\end{aligned}
$$

where we used (C.2). Comparing the first line with the least, we obtain

$$
\begin{aligned}
\left\|V^{\pi^*} - V^{\hat{\pi}^*}\right\|_{\infty} &\leq \left\|G_{\mathcal{P},\hat{\mathcal{P}}}^{\pi^*}V^*\right\|_{\infty} + \left\|G_{\mathcal{P},\hat{\mathcal{P}}}^{\hat{\pi}^*}V^*\right\|_{\infty} + \left\|G_{\mathcal{P},\hat{\mathcal{P}}}^{\hat{\pi}^*}(V^* - V^{\hat{\pi}^*})\right\|_{\infty} && \text{(C.3)} \\
&\leq 2c_1\|\epsilon_{\text{Model}}\|_{\infty}\|V^*\|_{\infty} + +c_1\|\epsilon_{\text{Model}}\|_{\infty}\left\|V^* - V^{\hat{\pi}^*}\right\|_{\infty}
\end{aligned}
$$

where we used Lemma 3. Rearranging the terms give the result. $\qquad \square$

## D   TECHNICAL DETAILS OF MAXIMUM ENTROPY DENSITY ESTIMATION

In this section we present the technical details of maximum entropy density estimation. This involves the duality methods for solving the optimization algorithms and some useful lemmas regarding their solutions. These problems are well studied in the literature. We do not make any assumptions on whether the environment states space, which will be the domain of the distributions in this section, is finite or continuous, and we make arguments for general measures. Due to this, our assumptions as well as our constraints may differ from the original papers in the literature. In those cases, we prove the results ourselves.

### D.1 MAXIMUM ENTROPY DENSITY ESTIMATION WITH EQUALITY CONSTRAINTS

Assume $Z$ is a random variable over domain $\mathcal{Z}$ with an unknown distribution $p \in \mathcal{M}(\mathcal{Z})$. For a set of functions $\phi_1, \ldots, \phi_d \colon \mathcal{Z} \to \mathbb{R}$ the expected values $\bar{\phi}_i = \mathbb{E}_{Z \sim p}[\phi_i(Z)]$ are given. We will use $\boldsymbol{\phi} \colon \mathcal{Z} \to \mathbb{R}^d$ and $\bar{\boldsymbol{\phi}} \in \mathbb{R}^d$ to refer to the respective vector forms. We also have access to an approximate distribution $\hat{p} \in \mathcal{M}(\mathcal{Z})$ such that $\hat{p} \approx p$. The maximum entropy density estimation gives a new approximation $q^*$ that is the solution of the following optimization problem

$$\min_{q \in \mathcal{M}(\mathcal{Z})} \quad D_{\mathrm{KL}}(\, q \parallel \hat{p} \,), \tag{D.1}$$

$$\text{s.t. } \mathbb{E}_{Z \sim q}[\phi_i(Z)] = \bar{\phi}_i \qquad (1 \le i \le d).$$

The KL-divergence in (D.1) is finite only if $q$ is absolutely continuous w.r.t. $\hat{p}$. In that case, $q$ can be specified with its density w.r.t $\hat{p}$ defined as $f \triangleq \frac{\mathrm{d}q}{\mathrm{d}\hat{p}} \colon \mathcal{Z} \to \mathbb{R}$ where $\frac{\mathrm{d}q}{\mathrm{d}\hat{p}}$ is the Radon-Nikodym derivative. Let $L_1(\mathcal{Z}, \hat{p})$ be the $L_1$ space over $\mathcal{Z}$ with measure $\hat{p}$. We have $f \in L_1(\mathcal{Z}, \hat{p})$.

Also for any $f \in L_1(\mathcal{Z}, \hat{p})$ such that $f \ge 0$ and $\int f(z)\hat{p}(\mathrm{d}z) = 1$ we can recover a distribution $q \in \mathcal{M}(\mathcal{Z})$ as

$$q(A) = \int_A f(z)\hat{p}(\mathrm{d}z). \tag{D.2}$$

Consequently, (D.1) can be written in terms of $f$. We have

$$\begin{aligned}
D_{\mathrm{KL}}(\, q \parallel \hat{p} \,) &= \int q(\mathrm{d}z) \log \frac{q(\mathrm{d}z)}{p(\mathrm{d}z)} \\
&= \int \frac{q(\mathrm{d}z)}{p(\mathrm{d}z)} \log \frac{q(\mathrm{d}z)}{p(\mathrm{d}z)} \cdot \hat{p}(\mathrm{d}z) \\
&= \int f(z) \log f(z) \cdot \hat{p}(\mathrm{d}z)
\end{aligned}$$

We can write (D.1) as

$$\min_f \quad \int f(z) \log f(z) \, \hat{p}(\mathrm{d}z), \tag{D.3}$$

$$\text{s.t. } \int f(z)\phi_i(z) \, \hat{p}(\mathrm{d}z) = \bar{\phi}_i \qquad (1 \le i \le d),$$

$$\int f(z) \, \hat{p}(\mathrm{d}z) = 1,$$

$$f \in L_1(\mathcal{Z}, \hat{p}).$$

The constraint $f \ge 0$ is implicit in the domain of the KL objective. The Lagrangian with dual parameters $\lambda, \Lambda'$ is

$$L(f, \lambda, \Lambda') = \int \left[ f(z) \log f(z) - \sum_{i=1}^d \lambda_i f(z)\phi_i(z) + \Lambda' f(z) \right] \hat{p}(\mathrm{d}z) + \sum_i \lambda_i \bar{\phi}_i - \Lambda'. \tag{D.4}$$

Then, we can obtain the dual objective $D_{\bar{\phi}}(\lambda, \Lambda') \triangleq \inf_f L(f, \lambda, \Lambda')$ from this result by Decarreau et al. (1992, Proposition 2.4).

**Lemma 4** ((Decarreau et al., 1992)). *For any fixed $\lambda \in \mathbb{R}^d, \Lambda' \in \mathbb{R}$, the Lagrangian $L$ in (D.4) has a unique minimizer $f_{\lambda, \Lambda'}$ defined as*

$$f_{\lambda, \Lambda'} \triangleq \exp\left( \sum_{i=1}^d \lambda_i \phi_i(z) - \Lambda' - 1 \right).$$

*The dual objective is given by*

$$D_{\bar{\phi}}(\lambda, \Lambda') = -\int \exp\left( \sum_{i=1}^d \lambda_i \phi_i(z) - \Lambda' - 1 \right)\hat{p}(\mathrm{d}z) + \sum_{i=1}^d \lambda_i \bar{\phi}_i - \Lambda'.$$

*It is concave, continuously differentiable, and its partial derivatives are*

$$\frac{\partial D_{\bar{\phi}}(\lambda, \Lambda')}{\partial \lambda_i} = \bar{\phi}_i - \int f_{\lambda, \Lambda'}(z)\phi_i(z)\hat{p}(\mathrm{d}z) \qquad , \qquad \frac{\partial D_{\bar{\phi}}(\lambda, \Lambda')}{\partial \Lambda'} = 1 - \int f_{\lambda, \Lambda'}(z)\hat{p}(\mathrm{d}z).$$

Thus, for the dual objective $D_{\bar{\phi}}(\lambda, \Lambda')$ we have $D_{\bar{\phi}}(\lambda, \Lambda') = L(f_{\lambda,\Lambda'}, \lambda, \Lambda')$. We arrive at the following dual problem

$$\max_{\lambda \in \mathbb{R}^d, \Lambda' \in \mathbb{R}} D_{\bar{\phi}}(\lambda, \Lambda') = \left[ -\int \exp\left( \sum_{i=1}^d \lambda_i \phi_i(z) - \Lambda' - 1 \right) \hat{p}(\mathrm{d}z) + \sum_{i=1}^d \lambda_i \bar{\phi}_i - \Lambda' \right], \quad \text{(D.5)}$$

The following result by Borwein and Lewis (1991, Corollary 2.6 and Theorem 4.8) shows the duality of the problems.

**Theorem 3** ((Borwein and Lewis, 1991)). *Assume $\phi_i \in L_\infty(\mathcal{Z}, \hat{p})$ for $i = 1, \ldots, n$. Under certain constraint qualification constraints, the value of (D.3) and (D.5) is equal with dual attainment. Furthermore, let $\lambda^*, \Lambda'^*$ be dual optimal. The primal optimal solution is $f_{\lambda^*,\Lambda'^*}$.*

We do not discuss the technical details of the constraint qualification constraints and refer the readers to (Altun and Smola, 2006; Borwein and Lewis, 1991; Decarreau et al., 1992) for a complete discussion.

For any $\lambda \in \mathbb{R}^d$, the optimal value of $\Lambda'$ can be computed. Due to Lemma 4, we have

$$\frac{\partial D_{\bar{\phi}}(\lambda, \Lambda')}{\partial \Lambda'} = \frac{\partial D_{\beta,\bar{\phi}}(\lambda, \Lambda')}{\partial \Lambda'} = 1 - \int f_{\lambda,\Lambda'}(z) \hat{p}(\mathrm{d}z) \quad \text{(D.6)}$$

solving for the optimal $\Lambda'$ gives the following value

$$\Lambda'_\lambda \triangleq \log \int \exp\left( \sum_{i=1}^d \lambda_i \phi_i(z) - 1 \right) \hat{p}(\mathrm{d}z) = \Lambda_\lambda - 1, \quad \text{(D.7)}$$

where $\Lambda_\lambda$ is defined in Section 2.2. Note that since functions $\phi_i$ are bounded this quantity is finite. Thus, we can just optimize $D_{\bar{\phi}}(\lambda, \Lambda'_\lambda)$ over $\lambda$. By substitution we get

$$D_{\bar{\phi}}(\lambda) \triangleq D_{\bar{\phi}}(\lambda, \Lambda'_\lambda) = \sum_{i=1}^d \lambda_i \bar{\phi}_i - \log \int \exp\left( \sum_{i=1}^d \lambda_i \phi_i(z) \right) \hat{p}(\mathrm{d}z). \quad \text{(D.8)}$$

We observe that $\max_\lambda D_{\bar{\phi}}(\lambda)$ is equivalent to (2.4). From Theorem 3, if $\lambda^*$ optimizes $D_{\bar{\phi}}$, we know that $f_{\lambda^*,\Lambda'_{\lambda^*}}$ optimizes (D.3). Due to equivalence of (D.3) and (D.1), then $q_{\lambda^*}$ defined as

$$q_{\lambda^*}(A) \triangleq \int_A f_{\lambda^*,\Lambda'_{\lambda^*}}(z) \hat{p}(\mathrm{d}z) = \int_{z \in A} \hat{p}(\mathrm{d}z) \cdot \exp\left( \sum_{i=1}^d \lambda_i^* \phi_i(z) - \Lambda_{\lambda^*} \right) \quad \text{(D.9)}$$

for all $A \subseteq \mathcal{Z}$ optimizes (D.1).

## D.2 MAXIMUM ENTROPY DENSITY ESTIMATION WITH $\ell_2^2$ REGULARIZATION

We now study a relaxed form of the maximum entropy density estimation. In this form, instead of imposing strict equality constraints, the mismatch between the expected value $\mathbb{E}_{Z \sim q}[\phi_i(Z)]$ with $\bar{\phi}_i$ is added to the loss. The benefit of this version is that even if $\bar{\phi}_i$ values are not exactly equal to $\mathbb{E}_{Z \sim p}[\phi_i(Z)]$, the problem remains feasible. Moreover, we can adjust the weight of this term in the loss based on the accuracy of $\bar{\phi}_i$ values. Specifically, we define the following problem.

$$\min_{q \in \mathcal{M}(\mathcal{Z})} \quad D_{\mathrm{KL}}(q \parallel \hat{p}) + \frac{1}{\beta^2} \sum_{i=1}^d \left( \mathbb{E}_{Z \sim q}[\phi_i(Z)] - \bar{\phi}_i \right)^2 \quad \text{(D.10)}$$

Similar to the previous section, we can write the above problem in terms of the density $\frac{\mathrm{d}q}{\mathrm{d}\hat{p}}$ and write (Decarreau et al., 1992)

$$\min_{f, \xi} \quad \int f(z) \log f(z) \, \hat{p}(\mathrm{d}z) + \frac{1}{\beta^2} \sum_{i=1}^d \xi_i^2, \quad \text{(D.11)}$$

$$\text{s.t.} \quad \int f(z) \phi_i(z) \, \hat{p}(\mathrm{d}z) - \bar{\phi}_i = \xi_i \quad (1 \le i \le d),$$

$$\int f(z) \, \hat{p}(\mathrm{d}z) = 1,$$

$$f \in L_1(\mathcal{Z}, \hat{p}), \, \xi \in \mathbb{R}^d.$$

The Lagrangian of this problems can be written as

$$L_\beta(f, \xi, \lambda, \Lambda') = L(f, \lambda, \Lambda') + \frac{1}{\beta^2} \sum_{i=1}^{d} \xi_i^2 + \sum_{i=1}^{d} \lambda_i \xi_i \qquad (D.12)$$

The dual objective is then

$$D_{\beta, \bar{\phi}}(\lambda, \Lambda') = \inf_{f, \xi} L_\beta(f, \xi, \lambda, \Lambda')$$

It can be observed that $f$ and $\xi$ can be independently optimized for any fixed $\lambda, \Lambda'$. Due to Lemma 4, the optimal value of $f$ is $f_{\lambda, \Lambda'}$. The optimal value of $\xi_i$ can be calculated as

$$\xi_\lambda = -\frac{1}{2} \beta^2 \lambda \qquad (D.13)$$

We arrive at the following dual objective

$$
\begin{aligned}
D_{\beta, \bar{\phi}}(\lambda, \Lambda') &= L(f_{\lambda, \Lambda'}, \lambda, \Lambda') + \frac{1}{4} \beta^2 \sum_{i=1}^{d} \lambda_i^2 + \frac{1}{\beta^2} \sum_{i=1}^{d} \xi_{\lambda_i}^2 + \sum_{i=1}^{d} \lambda_i \xi_{\lambda_i}, \qquad (D.14) \\
&= L(f_{\lambda, \Lambda'}, \lambda, \Lambda') + \frac{1}{4} \beta^2 \sum_{i=1}^{d} \lambda_i^2 - \frac{1}{2} \beta^2 \sum_{i=1}^{d} \lambda_i^2, \\
&= D_{\bar{\phi}}(\lambda, \Lambda') - \frac{1}{4} \beta^2 \sum_{i=1}^{d} \lambda_i^2,
\end{aligned}
$$

which means we have the dual problem

$$\max_{\lambda \in \mathbb{R}^d, \Lambda' \in \mathbb{R}} D_{\bar{\phi}}(\lambda, \Lambda') - \frac{1}{4} \beta^2 \sum_{i=1}^{d} \lambda_i^2. \qquad (D2)$$

We now show the duality of the problems. Notice how this problem has an extra $\frac{1}{4} \beta^2 \sum_{i=1}^{d} \lambda_i^2$ compared to (2.4). This is the reason this problem is considered the regularized version of (2.4). Notice that the regularization term also makes the dual loss strongly concave. This makes solving the optimization problem easier.

**Theorem 4.** *Assume $\phi_i$ is bounded for $i = 1, \ldots, n$ and $\beta > 0$. The value of (D.11) and (D2) is equal with dual attainment. Furthermore, let $\lambda^*, \Lambda'^*$ be dual optimal. The primal optimal solution is $f_{\lambda^*, \Lambda'^*}$.*

**Proof.** First, we show that the some solution $\lambda^*, \Lambda'^*$ exists for the dual problem. To see this, first note that for any $\lambda \in \mathbb{R}^d$, the optimal value of $\Lambda'$ is $\Lambda'_\lambda$ defined in (D.6). Now we need to show $D_{\beta, \bar{\phi}}(\lambda, \Lambda'_\lambda)$ is maximized by some $\lambda^*$. We have

$$D_{\beta, \bar{\phi}}(\lambda, \Lambda'_\lambda) = D_{\bar{\phi}}(\lambda, \Lambda'_\lambda) - \frac{1}{4} \beta^2 \sum_{i=1}^{d} \lambda_i^2. \qquad (D.15)$$

Since $D$ is concave, $D_{\bar{\phi}}(\lambda, \Lambda'_\lambda)$ and therefore $D_{\beta, \bar{\phi}}(\lambda, \Lambda'_\lambda)$ is also concave. Due to Weierstrass' Theorem (Bertsekas, 2009, Proposition 3.2.1) it suffices to show the set

$$S = \{ \lambda \in \mathbb{R}^d : D_{\beta, \bar{\phi}}(\lambda, \Lambda'_\lambda) \geq D_{\beta, \bar{\phi}}(0, \Lambda'_0) \}$$

is non-empty and bounded. It is trivially non-empty. Assume $|\phi(z)| \leq \phi_{\max}$ for any $z$ and $1 \leq i \leq d$. For any $\lambda \in S$, we have

$$D_{\beta,\bar{\phi}}(0, \Lambda'_0) \leq D_{\beta,\bar{\phi}}(\lambda, \Lambda'_\lambda)$$

$$= D_{\bar{\phi}}(\lambda, \Lambda'_\lambda) - \frac{1}{4}\beta^2 \sum_{i=1}^{d} \lambda_i^2$$

$$= -\int \exp\left(\sum_{i=1}^{d} \lambda_i \phi_i(z) - \Lambda'_\lambda - 1\right)\hat{p}(\mathrm{d}z) + \sum_{i=1}^{d} \lambda_i \bar{\phi}_i - \Lambda'_\lambda - \frac{1}{4}\beta^2 \sum_{i=1}^{d} \lambda_i^2$$

$$= -1 + \sum_{i=1}^{d} \lambda_i \bar{\phi}_i - \log\int \exp\left(\sum_{i=1}^{d} \lambda_i \phi_i(z) - 1\right)\hat{p}(\mathrm{d}z) - \frac{1}{4}\beta^2 \sum_{i=1}^{d} \lambda_i^2$$

$$\leq d\|\lambda\|_\infty \|\bar{\phi}\|_\infty - \log\int \exp\left(-d\|\lambda\|_\infty \phi_{\max} - 1\right)\hat{p}(\mathrm{d}z) - \frac{1}{4}\beta^2 \|\lambda\|_\infty^2$$

$$\leq -\frac{1}{4}\beta^2 \|\lambda\|_\infty^2 + d\|\lambda\|_\infty \|\bar{\phi}\|_\infty + d\|\lambda\|_\infty \phi_{\max} + 1.$$

which enforces an upper bound on $\|\lambda\|_\infty$. This means that some optimal solution $\lambda^*, \Lambda'_{\lambda^*}$ exists.

Now we show that $f_{\lambda^*, \Lambda'_{\lambda^*}}$ is primal optimal with $\xi_{\lambda^*}$. First, note that the derivative (D.6) is zero for $\lambda^*, \Lambda'_{\lambda^*}$ due to the derivation of $\Lambda'_\lambda$. Thus, $f_{\lambda^*, \Lambda'_{\lambda^*}}$ is feasible. Similarly using Lemma 4 we have from (D.15)

$$0 = \frac{\partial D_{\beta,\bar{\phi}}(\lambda^*, \Lambda'_{\lambda^*})}{\partial \lambda_i}$$

$$= \frac{\partial D_{\bar{\phi}}(\lambda^*, \Lambda'_{\lambda^*})}{\partial \lambda_i} - \frac{1}{2}\beta^2 \lambda_i^*$$

$$= \bar{\phi}_i - \int f_{\lambda^*, \Lambda'_{\lambda^*}}(z)\phi_i(z)\hat{p}(\mathrm{d}z) + \xi_{\lambda^* i},$$

which shows $\xi_{\lambda^*}$ is feasible.

Consider another feasible $f, \xi$ for (D.11).

$$\int f(z)\log f(z)\,\hat{p}(\mathrm{d}z) + \frac{1}{\beta^2}\sum_{i=1}^{d}\xi_i^2 = L_\beta(f, \xi, \lambda^*, \Lambda'_{\lambda^*})$$

$$\geq D_{\beta,\bar{\phi}}(\lambda^*, \Lambda'_{\lambda^*})$$

$$= L_\beta(f_{\lambda^*, \Lambda'_{\lambda^*}}, \xi_{\lambda^*}, \lambda^*, \Lambda'_{\lambda^*})$$

$$= \int f_{\lambda^*, \Lambda'_{\lambda^*}}(z)\log f_{\lambda^*, \Lambda'_{\lambda^*}}(z)\,\hat{p}(\mathrm{d}z) + \frac{1}{\beta^2}\sum_{i=1}^{d}\xi_{\lambda^*}{}^2,$$

which proves the claim. $\qquad\qquad\square$

Similar to the exact formulation, we can substitute $\Lambda'$ with $\Lambda'_\lambda$ to obtain a loss function based on $\lambda$. We arrive at the loss function

$$D_{\beta,\bar{\phi}}(\lambda) \triangleq D_{\beta,\bar{\phi}}(\lambda, \Lambda'_\lambda) = D_{\bar{\phi}}(\lambda, \Lambda'_\lambda) - \frac{1}{4}\beta^2 \sum_{i=1}^{d} \lambda_i^2$$

$$= D_{\bar{\phi}}(\lambda) - \frac{1}{4}\beta^2 \sum_{i=1}^{d} \lambda_i^2$$

$$= \sum_{i=1}^{d} \lambda_i \bar{\phi}_i - \log\int \exp\left(\sum_{i=1}^{d} \lambda_i \phi_i(z)\right)\hat{p}(\mathrm{d}z) - \frac{1}{4}\beta^2 \sum_{i=1}^{d} \lambda_i^2. \quad \text{(D.16)}$$

If $\lambda^*$ optimizes $D_{\beta,\bar{\phi}}$, Theorem 4 shows $f_{\lambda^*, \Lambda'_{\lambda^*}}$ optimizes (D.11). Due to equivalence of (D.11) with (D.10), we get that $q_{\lambda^*}$ optimizes (D.10).

### D.3 LEMMAS REGARDING MAXIMUM ENTROPY DENSITY ESTIMATION

**Lemma 5** ([Dudík et al. (2007)](#)). *For $\phi^{(1)}, \phi^{(2)} \in \mathbb{R}^d$, let $\lambda^{(1)}, \lambda^{(2)}$ be the maximizers of $D_{\beta,\bar\phi^{(1)}}$ and $D_{\beta,\bar\phi^{(2)}}$, respectively. We have*

$$\left\| \lambda^{(1)} - \lambda^{(2)} \right\|_2 \le \frac{2}{\beta^2} \cdot \left\| \bar\phi^{(1)} - \bar\phi^{(2)} \right\|_2.$$

**Proof.** Define

$$g(\lambda) \triangleq \log \int \exp\left( \sum_{i=1}^d \lambda_i \phi_i(z) \right) \hat p(\mathrm{d}z).$$

Since $g(\lambda) = \sum_{i=1}^d \lambda_i \bar\phi_i^{(1)} - D_{\bar\phi^{(1)}}(\lambda)$ and $D_{\bar\phi^{(1)}}(\lambda)$ is concave, we know that $g$ is convex. Due to optimality of $\lambda^{(1)}, \lambda^{(2)}$ we have

$$\nabla D_{\beta,\bar\phi^{(1)}}(\lambda^{(1)}) = -\nabla g(\lambda^{(1)}) + \bar\phi^{(1)} - \frac{1}{2}\beta^2\lambda^{(1)} = 0,$$

$$\nabla D_{\beta,\bar\phi^{(2)}}(\lambda^{(2)}) = -\nabla g(\lambda^{(2)}) + \bar\phi^{(2)} - \frac{1}{2}\beta^2\lambda^{(2)} = 0.$$

By taking the difference we get

$$\frac{1}{2}\beta^2(\lambda^{(1)} - \lambda^{(2)}) = -(\nabla g(\lambda^{(1)}) - \nabla g(\lambda^{(2)})) + (\bar\phi^{(1)} - \bar\phi^{(2)}).$$

Multiplying both sides by $(\lambda^{(1)} - \lambda^{(2)})^\top$ we get

$$\frac{1}{2}\beta^2\left\| \lambda^{(1)} - \lambda^{(2)} \right\|_2^2 = -\langle \nabla g(\lambda^{(1)}) - \nabla g(\lambda^{(2)}), \lambda^{(1)} - \lambda^{(2)} \rangle + \langle \bar\phi^{(1)} - \bar\phi^{(2)}, \lambda^{(1)} - \lambda^{(2)} \rangle.$$

Due to the convexity of $g$, we have

$$\langle \nabla g(\lambda^{(1)}) - \nabla g(\lambda^{(2)}), \lambda^{(1)} - \lambda^{(2)} \rangle \ge 0.$$

Thus, we continue

$$\begin{aligned}
\frac{1}{2}\beta^2\left\| \lambda^{(1)} - \lambda^{(2)} \right\|_2^2 &= -\langle \nabla g(\lambda^{(1)}) - \nabla g(\lambda^{(2)}), \lambda^{(1)} - \lambda^{(2)} \rangle + \langle \bar\phi^{(1)} - \bar\phi^{(2)}, \lambda^{(1)} - \lambda^{(2)} \rangle \\
&\le \langle \bar\phi^{(1)} - \bar\phi^{(2)}, \lambda^{(1)} - \lambda^{(2)} \rangle \\
&\le \left\| \bar\phi^{(1)} - \bar\phi^{(2)} \right\|_2 \left\| \lambda^{(1)} - \lambda^{(2)} \right\|_2,
\end{aligned}$$

where we used the Cauchy–Schwarz inequality. Dividing by $\|\lambda^{(1)} - \lambda^{(2)}\|_2$ proves the result. $\qquad\square$

**Lemma 6** ([Dudík et al. (2007)](#)). *Let $\lambda^*$ maximize $D_{\beta,\bar\phi}$ defined in (D.16). Then for any $\lambda$, we have*

$$D_{\mathrm{KL}}(\, p \parallel q_{\lambda^*} \,) \le D_{\mathrm{KL}}(\, p \parallel q_\lambda \,) + \frac{2}{\beta^2}\left\| \mathbb{E}_{Z\sim p}[\phi(Z)] - \bar\phi \right\|_2^2 + \frac{\beta^2}{4}\|\lambda\|_2^2.$$

**Proof.** Define $\bar\phi^* \triangleq \mathbb{E}_{Z\sim p}[\phi(Z)]$. First, we show that for any $\lambda \in \mathbb{R}^d$, we have

$$\begin{aligned}
D_{\mathrm{KL}}(\, p \parallel q_\lambda \,) &= \int p(\mathrm{d}z) \log \frac{p(\mathrm{d}z)}{q_\lambda(\mathrm{d}z)} \\
&= \int p(\mathrm{d}z) \log \frac{p(\mathrm{d}z)}{\hat p(\mathrm{d}z)\exp(\sum \lambda_i \phi_i(z) - \Lambda_\lambda)} \\
&= \int p(\mathrm{d}z) \log \frac{p(\mathrm{d}z)}{\hat p(\mathrm{d}z)} - \int p(\mathrm{d}z)\sum \lambda_i \phi_i(z) + \Lambda_\lambda \\
&= D_{\mathrm{KL}}(\, p \parallel \hat p \,) - \langle \lambda, \bar\phi^* \rangle + \Lambda_\lambda.
\end{aligned} \tag{D.17}$$

Now we can write from (D.16) and (D.17) that

$$D_{\beta,\bar{\phi}}(\lambda) = \langle \lambda, \bar{\phi} \rangle - \Lambda_\lambda - \frac{1}{4}\beta^2 \|\lambda\|_2^2$$

$$= D_{\text{KL}}(\,p \parallel \hat{p}\,) - D_{\text{KL}}(\,p \parallel \hat{p}\,) + \langle \lambda, \bar{\phi}^* \rangle + \langle \lambda, \bar{\phi} - \bar{\phi}^* \rangle - \Lambda_\lambda - \frac{1}{4}\beta^2 \|\lambda\|_2^2$$

$$= D_{\text{KL}}(\,p \parallel \hat{p}\,) - D_{\text{KL}}(\,p \parallel q_\lambda\,) + \langle \lambda, \bar{\phi} - \bar{\phi}^* \rangle - \frac{1}{4}\beta^2 \|\lambda\|_2^2. \qquad (\text{D.18})$$

Define

$$\lambda^{**} \triangleq \underset{\lambda_0}{\operatorname{argmax}}\, D_{\beta,\bar{\phi}^*}(\lambda_0).$$

Due to optimality of $\lambda^*$, we have $D_{\beta,\bar{\phi}}(\lambda^{**}) \le D_{\beta,\bar{\phi}}(\lambda^*)$. Expanding both sides with (D.18), we get from the Cauchy–Schwarz and Lemma 5,

$$D_{\text{KL}}(\,p \parallel q_{\lambda^*}\,) \le \left[ D_{\text{KL}}(\,p \parallel \hat{p}\,) + \langle \lambda^*, \bar{\phi} - \bar{\phi}^* \rangle - \frac{1}{4}\beta^2 \|\lambda^*\|_2^2 \right] - $$

$$\left[ D_{\text{KL}}(\,p \parallel \hat{p}\,) - D_{\text{KL}}(\,p \parallel q_{\lambda^{**}}\,) + \langle \lambda^{**}, \bar{\phi} - \bar{\phi}^* \rangle - \frac{1}{4}\beta^2 \|\lambda^{**}\|_2^2 \right]$$

$$= D_{\text{KL}}(\,p \parallel q_{\lambda^{**}}\,) + \langle \lambda^* - \lambda^{**}, \bar{\phi} - \bar{\phi}^* \rangle - \frac{1}{4}\beta^2(\|\lambda^*\|_2^2 - \|\lambda^{**}\|_2^2)$$

$$\le D_{\text{KL}}(\,p \parallel q_{\lambda^{**}}\,) + \|\lambda^* - \lambda^{**}\|_2 \|\bar{\phi} - \bar{\phi}^*\|_2 + \frac{1}{4}\beta^2 \|\lambda^{**}\|_2^2$$

$$\le D_{\text{KL}}(\,p \parallel q_{\lambda^{**}}\,) + \frac{2}{\beta^2} \|\bar{\phi} - \bar{\phi}^*\|_2^2 + \frac{1}{4}\beta^2 \|\lambda^{**}\|_2^2.$$

On the other hand, due to optimality of $\lambda^{**}$, we have $D_{\beta,\bar{\phi}^*}(\lambda) \le D_{\beta,\bar{\phi}^*}(\lambda^{**})$. Expanding both sides with (D.18), we get

$$D_{\text{KL}}(\,p \parallel q_\lambda\,) + \frac{1}{4}\beta^2 \|\lambda\|_2^2 \ge D_{\text{KL}}(\,p \parallel q_{\lambda^{**}}\,) + \frac{1}{4}\beta^2 \|\lambda^{**}\|_2^2.$$

Combining the last two inequalities, we get

$$D_{\text{KL}}(\,p \parallel q_{\lambda^*}\,) \le D_{\text{KL}}(\,p \parallel q_\lambda\,) + \frac{1}{4}\beta^2 \|\lambda\|_2^2 + \frac{2}{\beta^2} \|\bar{\phi} - \bar{\phi}^*\|_2^2,$$

which proves the claim. $\qquad\square$

## E    PROOFS FOR SECTION 3.1

We first show the following lemma:

**Lemma 7.** *For any policy $\pi$, we have* $\left\| G_{\mathcal{P},\bar{\mathcal{P}}}^\pi \right\|_\infty \le c_1 \|\epsilon_{\text{Model}}\|_\infty$ .

**Proof.** Since the feasibility set of Problem (P1) is convex, and $\mathcal{P}(\cdot|x,a)$ belongs to it, we have from Pythagoras theorem for KL-divergence (see Thm. 11.6.1 of Cover and Thomas 2006) that

$$D_{\text{KL}}(\,\mathcal{P}(\cdot|x,a) \parallel \hat{\mathcal{P}}(\cdot|x,a)\,) \ge D_{\text{KL}}(\,\mathcal{P}(\cdot|x,a) \parallel \bar{\mathcal{P}}(\cdot|x,a)\,) + D_{\text{KL}}(\,\bar{\mathcal{P}}(\cdot|x,a) \parallel \hat{\mathcal{P}}(\cdot|x,a)\,)$$

$$\ge D_{\text{KL}}(\,\mathcal{P}(\cdot|x,a) \parallel \bar{\mathcal{P}}(\cdot|x,a)\,).$$

From Lemma 3 we have

$$\left\| G_{\mathcal{P},\bar{\mathcal{P}}}^\pi \right\|_\infty \le c_1 \sup_{x,a} \sqrt{D_{\text{KL}}(\,\mathcal{P}(\cdot|x,a) \parallel \bar{\mathcal{P}}(\cdot|x,a)\,)}$$

$$\le c_1 \sup_{x,a} \sqrt{D_{\text{KL}}(\,\mathcal{P}(\cdot|x,a) \parallel \hat{\mathcal{P}}(\cdot|x,a)\,)}$$

$$\le c_1 \|\epsilon_{\text{Model}}\|_\infty.$$

$\square$

**Proof of Proposition 1**

**Proof.** Due the constraint in Problem (P1), for any $i$ we have $(\mathcal{P} - \bar{\mathcal{P}})\phi_i = 0$ and therefore $G^{\pi_{\text{PE}}}_{\mathcal{P},\bar{\mathcal{P}}}\phi_i = 0$. Thus, using the proof of Lemma 1, for any $w \in \mathbb{R}^d$

$$
\begin{aligned}
\left\| V^{\pi_{\text{PE}}} - \bar{V}^{\pi_{\text{PE}}} \right\|_\infty &\le \left\| G^{\pi_{\text{PE}}}_{\mathcal{P},\bar{\mathcal{P}}} V^{\pi_{\text{PE}}} \right\|_\infty \\
&= \left\| G^{\pi_{\text{PE}}}_{\mathcal{P},\bar{\mathcal{P}}} \left( V^{\pi_{\text{PE}}} - \sum_i w_i \phi_i \right) \right\|_\infty \\
&= \left\| G^{\pi_{\text{PE}}}_{\mathcal{P},\bar{\mathcal{P}}} \right\|_\infty \left\| V^{\pi_{\text{PE}}} - \sum_i w_i \phi_i \right\|_\infty \\
&= c_1 \|\epsilon_{\text{Model}}\|_\infty \left\| V^{\pi_{\text{PE}}} - \sum_i w_i \phi_i \right\|_\infty.
\end{aligned}
$$

Similarly for control, from (C.3), we have for any $w \in \mathbb{R}^d$

$$
\begin{aligned}
\left\| V^{\pi^*} - V^{\bar{\pi}^*} \right\|_\infty &\le \left\| G^{\pi^*}_{\mathcal{P},\bar{\mathcal{P}}} V^* \right\|_\infty + \left\| G^{\bar{\pi}^*}_{\mathcal{P},\bar{\mathcal{P}}} V^* \right\|_\infty + \left\| G^{\bar{\pi}^*}_{\mathcal{P},\bar{\mathcal{P}}} (V^* - V^{\bar{\pi}^*}) \right\|_\infty \\
&\le \left\| G^{\pi^*}_{\mathcal{P},\bar{\mathcal{P}}} \left( V^* - \sum_i w_i \phi_i \right) \right\|_\infty + \left\| G^{\bar{\pi}^*}_{\mathcal{P},\bar{\mathcal{P}}} \left( V^* - \sum_i w_i \phi_i \right) \right\|_\infty \\
&\quad + \left\| G^{\bar{\pi}^*}_{\mathcal{P},\bar{\mathcal{P}}} (V^* - V^{\bar{\pi}^*}) \right\|_\infty \\
&\le 2 c_1 \left\| \epsilon_{\text{Model}} \right\|_\infty \left\| V^* - \sum_i w_i \phi_i \right\|_\infty + c_1 \left\| \epsilon_{\text{Model}} \right\|_\infty \left\| V^* - V^{\bar{\pi}^*} \right\|_\infty,
\end{aligned}
$$

where we used Lemma 7 in the last inequality. Rearranging the terms yields the result. $\square$

## F  PROOFS FOR SECTION 3.2

In this section, we provide the analysis of MaxEnt MoCo in supremum norm. We will show a sequence of lemmas before providing the result for general $\beta$ and then proof of Theorem 1.

**Lemma 8.** *If $\bar{\mathcal{P}}$ is the solution of the optimization problem (P2), for any $x, a$ we have*

$$
D_{\text{KL}}(\, \mathcal{P}(\cdot|x,a) \, \| \, \bar{\mathcal{P}}(\cdot|x,a) \,) \le D_{\text{KL}}(\, \mathcal{P}(\cdot|x,a) \, \| \, \hat{\mathcal{P}}(\cdot|x,a) \,) + \frac{2}{\beta^2} \epsilon_{\text{Query}}(x,a)^2.
$$

**Proof.** For $\lambda \in \mathbb{R}^d$, define

$$
q_\lambda(A) \triangleq \int_A \hat{\mathcal{P}}(\mathrm{d}y|x,a) \exp\left( \sum_{i=1}^d \lambda_i \phi_i(y) - \Lambda_\lambda \right),
$$

where $\Lambda_\lambda$ is the log-normalizer and $A \subseteq \mathcal{X}$. Due to Lemma 6, for any $\lambda$ we have

$$
D_{\text{KL}}(\, \mathcal{P}(\cdot|x,a) \, \| \, \bar{\mathcal{P}}(\cdot|x,a) \,) \le D_{\text{KL}}(\, \mathcal{P}(\cdot|x,a) \, \| \, q_\lambda(\cdot|x,a) \,) + \frac{2}{\beta^2} \sum_{i=1}^d [(\mathcal{P}\phi_i)(x,a) - \psi_i(x,a)]^2 +
$$

$$
\frac{\beta^2}{4} \|\lambda\|_2^2.
$$

Since $q_0 = \hat{\mathcal{P}}(\cdot|x,a)$, substituting $\lambda = 0$ gives the result. $\square$

**Lemma 9.** *If $\bar{\mathcal{P}}$ is the solution of the optimization problem* (P2), *for any $x, a$ we have*

$$\left\|\mathcal{P}(\cdot|x,a) - \bar{\mathcal{P}}(\cdot|x,a)\right\|_1 \le \sqrt{2}\epsilon_{\text{Model}}(x,a) + \frac{2}{\beta}\epsilon_{\text{Query}}(x,a).$$

**Proof.** Using Lemma 8 and Pinsker's inequality, and the fact that $\sqrt{a+b} \le \sqrt{a} + \sqrt{b}$, we write

$$\left\|\mathcal{P}(\cdot|x,a) - \bar{\mathcal{P}}(\cdot|x,a)\right\|_1 \le \sqrt{2D_{\text{KL}}(\,\mathcal{P}(\cdot|x,a) \,\|\, \bar{\mathcal{P}}(\cdot|x,a)\,)}$$

$$\le \sqrt{2D_{\text{KL}}(\,\mathcal{P}(\cdot|x,a) \,\|\, \bar{\mathcal{P}}(\cdot|x,a)\,) + \frac{4}{\beta^2}\epsilon_{\text{Query}}(x,a)^2}$$

$$\le \sqrt{2}\epsilon_{\text{Model}}(x,a) + \frac{2}{\beta}\epsilon_{\text{Query}}(x,a).$$

$\square$

**Lemma 10.** *For any $x, a$ we have*

$$\sum_{i=1}^{d}\left|(\bar{\mathcal{P}}\phi_i)(x,a) - (\mathcal{P}\phi_i)(x,a)\right| \le \sqrt{d}\Big(2\epsilon_{\text{Query}}(x,a) + \beta\epsilon_{\text{Model}}(x,a)\Big),$$

*also for any policy $\pi$*

$$\sum_{i=1}^{d}\left|(\bar{\mathcal{P}}^\pi\phi_i)(x) - (\mathcal{P}^\pi\phi_i)(x)\right| \le \sqrt{d}\int \pi(\mathrm{d}a|x)\Big(2\epsilon_{\text{Query}}(x,a) + \beta\epsilon_{\text{Model}}(x,a)\Big).$$

**Proof.** For a more compact presentation of the proof, let $p = \mathcal{P}(\cdot|x,a)$, $\hat{p} = \hat{\mathcal{P}}(\cdot|x,a)$, and $\bar{p} = \bar{\mathcal{P}}(\cdot|x,a)$. Let $\phi\colon \mathcal{X} \to \mathbb{R}^d$ and $\psi\colon \mathcal{X} \times \mathcal{A} \to \mathbb{R}^d$ be $d$-dimensional vectors formed by $\phi_i, \psi_i$. For $q \in \mathcal{M}(\mathcal{X})$ and $f\colon \mathcal{X} \to \mathbb{R}^d$, we write

$$q[f] \triangleq \mathbb{E}_{X\sim q}[f(X)].$$

We write

$$\|p[\phi] - \bar{p}[\phi]\|_1 \le \sqrt{d}\|p[\phi] - \bar{p}[\phi]\|_2$$

$$\le \sqrt{d}\Big(\|p[\phi] - \psi(x,a)\|_2 + \|\psi(x,a) - \bar{p}[\phi]\|_2\Big)$$

$$\le \sqrt{d}\Big(\epsilon_{\text{Query}}(x,a) + \|\psi(x,a) - \bar{p}[\phi]\|_2\Big). \tag{F.1}$$

Now note that $\bar{p}$ is the solution of (P2), the value of objective is smaller for $\bar{p}$ than it is for $p$. We obtain

$$D_{\text{KL}}(\,\bar{p} \,\|\, \hat{p}\,) + \frac{1}{\beta^2}\|\bar{p}[\phi] - \psi(x,a)\|_2^2 \le D_{\text{KL}}(\,p \,\|\, \hat{p}\,) + \frac{1}{\beta^2}\|p[\phi] - \psi(x,a)\|_2^2$$

$$\le \epsilon_{\text{Model}}(x,a)^2 + \frac{1}{\beta^2}\epsilon_{\text{Query}}(x,a)^2.$$

Thus,

$$\|\bar{p}[\phi] - \psi(x,a)\|_2 \le \sqrt{\beta^2\epsilon_{\text{Model}}(x,a)^2 + \epsilon_{\text{Query}}(x,a)^2 - \beta^2 D_{\text{KL}}(\,\bar{p} \,\|\, \hat{p}\,)}$$

$$\le \sqrt{\beta^2\epsilon_{\text{Model}}(x,a)^2 + \epsilon_{\text{Query}}(x,a)^2}$$

$$\le \beta\epsilon_{\text{Model}}(x,a) + \epsilon_{\text{Query}}(x,a).$$

Substituting in (F.1) we get

$$\|p[\phi] - \bar{p}[\phi]\|_1 \le \sqrt{d}\|p[\phi] - \bar{p}[\phi]\|_2$$

$$\le \sqrt{d}\Big(\|p[\phi] - \psi(x,a)\|_2 + \|\psi(x,a) - \bar{p}[\phi]\|_2\Big)$$

$$\le \sqrt{d}\Big(2\epsilon_{\text{Query}}(x,a) + \beta\epsilon_{\text{Model}}(x,a)\Big).$$

For the second part we simply write

$$\sum_{i=1}^{d}\left|(\bar{\mathcal{P}}^{\pi}\phi_i)(x) - (\mathcal{P}^{\pi}\phi_i)(x)\right| = \sum_{i=1}^{d}\left|\int \pi(\mathrm{d}a|x)\Big[(\bar{\mathcal{P}}\phi_i)(x,a) - (\mathcal{P}\phi_i)(x,a)\Big]\right|$$

$$\leq \sum_{i=1}^{d}\int \pi(\mathrm{d}a|x)\big|(\bar{\mathcal{P}}\phi_i)(x,a) - (\mathcal{P}\phi_i)(x,a)\big|$$

$$= \int \sum_{i=1}^{d}\pi(\mathrm{d}a|x)\big|(\bar{\mathcal{P}}\phi_i)(x,a) - (\mathcal{P}\phi_i)(x,a)\big|$$

$$\leq \int \sum_{i=1}^{d}\pi(\mathrm{d}a|x)\Big(2\epsilon_{\mathrm{Query}}(x,a) + \beta\epsilon_{\mathrm{Model}}(x,a)\Big),$$

where we used the first part for the second inequality. $\qquad\square$

**Lemma 11.** *If $\bar{\mathcal{P}}$ is the solution of the optimization problem* (P2), *for any policy $\pi$, $w \in \mathbb{R}^d$ and $v: \mathcal{X} \to \mathbb{R}$, we have*

$$\left\|G_{\mathcal{P},\bar{\mathcal{P}}}^{\pi}v\right\|_{\infty} \leq \frac{\gamma}{1-\gamma}\left(\sqrt{2}\|\epsilon_{\mathrm{Model}}\|_{\infty} + \frac{2}{\beta}\|\epsilon_{\mathrm{Query}}\|_{\infty}\right)\left\|v - \sum_{i}w_i\phi_i\right\|_{\infty}$$

$$+ \frac{\gamma\sqrt{d}}{1-\gamma}\Big(\beta\|\epsilon_{\mathrm{Model}}\|_{\infty} + 2\|\epsilon_{\mathrm{Query}}\|_{\infty}\Big)\|w\|_{\infty}$$

**Proof.** We have

$$\left\|G_{\mathcal{P},\bar{\mathcal{P}}}^{\pi}v\right\|_{\infty} = \left\|(\mathbf{I} - \gamma\bar{\mathcal{P}}^{\pi})^{-1}(\gamma\mathcal{P}^{\pi} - \gamma\bar{\mathcal{P}}^{\pi})v\right\|_{\infty} \tag{F.2}$$

$$\leq \left\|(\mathbf{I} - \gamma\bar{\mathcal{P}}^{\pi})^{-1}(\gamma\mathcal{P}^{\pi} - \gamma\bar{\mathcal{P}}^{\pi})(v - \sum_{i}w_i\phi_i)\right\|_{\infty} + \tag{F.3}$$

$$\left\|(\mathbf{I} - \gamma\bar{\mathcal{P}}^{\pi})^{-1}(\gamma\mathcal{P}^{\pi} - \gamma\bar{\mathcal{P}}^{\pi})(\sum_{i}w_i\phi_i)\right\|_{\infty}. \tag{F.4}$$

Using (C.1) in proof of Lemma 2 and Lemma 9 we have

$$\left\|\mathcal{P}^{\pi}(\cdot|x) - \bar{\mathcal{P}}^{\pi}(\cdot|x)\right\|_{1} \leq \int \pi(\mathrm{d}a|x)\left\|\mathcal{P}(\cdot|x,a) - \bar{\mathcal{P}}(\cdot|x,a)\right\|_{1}$$

$$\leq \int \pi(\mathrm{d}a|x)\Big[\sqrt{2}\epsilon_{\mathrm{Model}}(x,a) + \frac{2}{\beta}\epsilon_{\mathrm{Query}}(x,a)\Big]$$

$$\leq \sqrt{2}\|\epsilon_{\mathrm{Model}}\|_{\infty} + \frac{2}{\beta}\|\epsilon_{\mathrm{Query}}\|_{\infty}.$$

Thus, for the first term (F.3), we can write

$$\left\|(\mathbf{I} - \gamma\bar{\mathcal{P}}^{\pi})^{-1}(\gamma\mathcal{P}^{\pi} - \gamma\bar{\mathcal{P}}^{\pi})(v - \sum_{i}w_i\phi_i)\right\|_{\infty}$$

$$= \frac{\gamma}{1-\gamma}\sup_{x}\left\|\mathcal{P}^{\pi}(\cdot|x) - \bar{\mathcal{P}}^{\pi}(\cdot|x)\right\|_{1}\cdot\left\|v - \sum_{i}w_i\phi_i\right\|_{\infty}$$

$$\leq \frac{\gamma}{1-\gamma}\Big[\sqrt{2}\|\epsilon_{\mathrm{Model}}\|_{\infty} + \frac{2}{\beta}\|\epsilon_{\mathrm{Query}}\|_{\infty}\Big]\cdot\left\|v - \sum_{i}w_i\phi_i\right\|_{\infty}.$$

Now, for the second term (F.4), we can write

$$\left\| (\mathbf{I} - \gamma \bar{\mathcal{P}}^\pi)^{-1} (\gamma \mathcal{P}^\pi - \gamma \bar{\mathcal{P}}^\pi)(\sum_i w_i \phi_i) \right\|_\infty$$

$$= \frac{\gamma}{1-\gamma} \left\| (\mathcal{P}^\pi - \bar{\mathcal{P}}^\pi)(\sum_i w_i \phi_i) \right\|_\infty$$

$$= \frac{\gamma}{1-\gamma} \sup_x \left| \sum_i w_i[(\mathcal{P}^\pi \phi_i)(x) - (\bar{\mathcal{P}}^\pi \phi_i)(x))] \right|$$

$$= \frac{\gamma}{1-\gamma} \sup_x \sum_i \left| (\mathcal{P}^\pi \phi_i)(x) - (\bar{\mathcal{P}}^\pi \phi_i)(x) \right| \|w\|_\infty$$

$$= \frac{\gamma \sqrt{d}}{1-\gamma} \left[ 2\|\epsilon_{\text{Query}}\|_\infty + \beta\|\epsilon_{\text{Model}}\|_\infty \right] \|w\|_\infty.$$

Putting the bounds for (F.4) and (F.3) finishes the proof. $\qquad\square$

**Theorem 5.** *Define the mixed error values*

$$e_1 = \frac{\gamma}{1-\gamma} \cdot \left( \sqrt{2}\|\epsilon_{\text{Model}}\|_\infty + \frac{2}{\beta}\|\epsilon_{\text{Query}}\|_\infty \right), \quad e_2 = \frac{\sqrt{d} \cdot \gamma}{1-\gamma} \cdot \left( \beta\|\epsilon_{\text{Model}}\|_\infty + 2\|\epsilon_{\text{Query}}\|_\infty \right).$$

*Then, for any $w_{\max} \geq 0$, we have*

$$\left\| V^{\pi_{PE}} - \bar{V}^{\pi_{PE}} \right\|_\infty \leq e_1 \inf_{\|w\|_\infty \leq w_{\max}} \cdot \left\| V^{\pi_{PE}} - \sum_{i=1}^d w_i \cdot \phi_i \right\|_\infty + e_2 \cdot w_{\max},$$

$$\left\| V^* - V^{\bar{\pi}^*} \right\|_\infty \leq \frac{2e_1}{1-e_1} \inf_{\|w\|_\infty \leq w_{\max}} \cdot \left\| V^* - \sum_{i=1}^d w_i \cdot \phi_i \right\|_\infty + \frac{2e_2}{1-e_1} \cdot w_{\max}.$$

**Proof.** The PE result is a direct consequence of Lemma 11 and (C.2). We have

$$\left\| V^{\pi_{\text{PE}}} - \bar{V}^{\pi_{\text{PE}}} \right\|_\infty = \left\| G^{\pi_{\text{PE}}}_{\mathcal{P},\bar{\mathcal{P}}} V^{\pi_{\text{PE}}} \right\|_\infty$$

$$\leq \inf_{w \in \mathbb{R}^d} \left[ e_1 \cdot \left\| V^{\pi_{\text{PE}}} - \sum_{i=1}^d w_i \cdot \phi_i \right\|_\infty + e_2 \cdot \|w\|_\infty \right]$$

$$\leq e_1 \cdot \inf_{\|w\|_\infty \leq w_{\max}} \cdot \left\| V^{\pi_{\text{PE}}} - \sum_{i=1}^d w_i \cdot \phi_i \right\|_\infty + e_2 \cdot w_{\max}.$$

For control, from (C.3) we have

$$\left\| V^{\pi^*} - V^{\bar{\pi}^*} \right\|_\infty \leq \left\| G^{\pi^*}_{\mathcal{P},\hat{\mathcal{P}}} V^* \right\|_\infty + \left\| G^{\bar{\pi}^*}_{\mathcal{P},\hat{\mathcal{P}}} V^* \right\|_\infty + \left\| G^{\bar{\pi}^*}_{\mathcal{P},\hat{\mathcal{P}}} (V^* - V^{\bar{\pi}^*}) \right\|_\infty.$$

Choosing $w = 0$ in Lemma 11 we get

$$\left\| G^{\bar{\pi}^*}_{\mathcal{P},\hat{\mathcal{P}}} (V^* - V^{\bar{\pi}^*}) \right\|_\infty \leq e_1 \left\| V^* - V^{\bar{\pi}^*} \right\|_\infty$$

Also for any $w$ we get

$$\left\| G^{\pi^*}_{\mathcal{P},\hat{\mathcal{P}}} V^* \right\|_\infty \leq e_1 \cdot \left\| V^* - \sum_{i=1}^d w_i \cdot \phi_i \right\|_\infty + e_2 \cdot \|w\|_\infty$$

$$\left\| G^{\bar{\pi}^*}_{\mathcal{P},\hat{\mathcal{P}}} V^* \right\|_\infty \leq e_1 \cdot \left\| V^* - \sum_{i=1}^d w_i \cdot \phi_i \right\|_\infty + e_2 \cdot \|w\|_\infty.$$

Thus,

$$\left\|V^{\pi^*} - V^{\bar{\pi}^*}\right\|_\infty \le 2e_1 \cdot \left\|V^* - \sum_{i=1}^d w_i \cdot \phi_i\right\|_\infty + 2e_2 \cdot \|w\|_\infty + e_1 \left\|V^* - V^{\bar{\pi}^*}\right\|_\infty.$$

By rearranging, we get

$$\left\|V^{\pi^*} - V^{\bar{\pi}^*}\right\|_\infty \le \inf_{w \in \mathbb{R}^d} \left[ \frac{2e_1}{1-e_1} \cdot \left\|V^* - \sum_{i=1}^d w_i \cdot \phi_i\right\|_\infty + \frac{2e_2}{1-e_1} \cdot \|w\|_\infty \right]$$

$$\le \frac{2e_1}{1-e_1} \inf_{\|w\|_\infty \le w_{\max}} \cdot \left\|V^* - \sum_{i=1}^d w_i \cdot \phi_i\right\|_\infty + \frac{2e_2}{1-e_1} \cdot w_{\max}.$$

$\square$

**Proof of Theorem 1** It is the direct consequence of Theorem 5 with choosing $\beta = \|\epsilon_{\text{Query}}\|_\infty / \|\epsilon_{\text{Model}}\|_\infty$ and observing

$$e_1 = \frac{\gamma(2+\sqrt{2})}{1-\gamma} \cdot \|\epsilon_{\text{Model}}\|_\infty \le 3c_1 \|\epsilon_{\text{Model}}\|_\infty$$

$$e_2 = \frac{3\sqrt{d} \cdot \gamma}{1-\gamma} \cdot \|\epsilon_{\text{Query}}\|_\infty = c_2 \|\epsilon_{\text{Query}}\|_\infty.$$

# G $\quad \ell_p$ ANALYSIS OF MAXENT MOCO

The analysis in the Section 3.2 is based on the supremum norm, which can be overly conservative. First, the error in the model and queries are due to the error in a supervised learning problem. Supervised learning algorithms usually provide guarantees in a weighted $\ell_p$ norm rather than the supremum norm. Second, in the given results, the true value function $V^{\pi_{\text{PE}}}$ and $V^*$ should be approximated with the span of functions $\phi_i$ according to the supremum norm. This is a strong condition. Usually, there are states in the MDP that are irrelevant to the problem or even unreachable. Finding a good approximation of the value function in such states is not realistic.

Hence, in this section we give performance analysis of our method in terms of a weighted $\ell_p$ norm. We first define some necessary quantities before providing the results. For any function $f \colon \mathcal{X} \to \mathbb{R}$ and distribution $\rho \in \mathcal{M}(\mathcal{X})$, the norm $\|f\|_{p,\rho}$ is defined as

$$\|f\|_{p,\rho} \triangleq \left[ \int |f(x)|^p \rho(\mathrm{d}x) \right]^{1/p}.$$

Let $\pi$ be an arbitrary policy, and $\mathcal{P}_m^\pi$ be the $m$-step transition kernel under $\pi$. The discounted future-state distribution $\eta^\pi \colon \mathcal{X} \to \mathcal{M}(\mathcal{X})$ is defined as

$$\eta^\pi(\cdot|x) \triangleq \frac{1}{1-\gamma} \cdot \sum_{m=0}^\infty \gamma^m \mathcal{P}_m^\pi(\cdot|x).$$

Define $\omega^\pi(\cdot|x) \triangleq \int \eta^\pi(\mathrm{d}z|x)\hat{\mathcal{P}}^\pi(\cdot|z)$. This is the distribution of our state when making one transition according to $\hat{\mathcal{P}}^\pi$ from an initial state sampled from the discounted future-state distribution $\eta^\pi(z|x)$. Also let $\epsilon_{\text{Model}}^\pi \colon \mathcal{X} \to \mathbb{R}$ and $\epsilon_{\text{Query}}^\pi \colon \mathcal{X} \to \mathbb{R}$ be defined based on $\epsilon_{\text{Model}}$ and $\epsilon_{\text{Query}}$ similar the way $r^\pi$ is defined based on $r$. Assume for any $i$ and $x, a$ we have $A - B/2 \le \phi_i(x), \psi_i(x,a) \le A + B/2$ for some values $A$ and $B \ge 0$.

Let $\rho \in \mathcal{M}(\mathcal{X})$ be some distribution over states. We define two concentration coefficients for $\rho$. Similar coefficients have appeared in $\ell_p$ error propagation results in the literature (Kakade and

Langford, 2002; Munos, 2003; 2007; Farahmand et al., 2010; Scherrer et al., 2015). Define

$$C_1^\pi(\rho)^4 = \exp\left(\frac{B^2 d}{\beta^2}\right)^2 \int \rho(x) \left\|\frac{\mathrm{d}\eta^\pi(\cdot|x)}{\mathrm{d}\rho}\right\|_\infty^2 \left\|\frac{\mathrm{d}\omega^\pi(\cdot|x)}{\mathrm{d}\rho}\right\|_\infty^2$$

$$C_2^\pi(\rho)^4 = \frac{1}{\gamma} \cdot \int \rho(x) \left\|\frac{\mathrm{d}\eta^\pi(\cdot|x)}{\mathrm{d}\rho}\right\|_\infty^4$$

Here, $\frac{\mathrm{d}\eta^\pi(\cdot|x)}{\mathrm{d}\rho}$ and $\frac{\mathrm{d}\eta^\pi(\cdot|x)}{\mathrm{d}\rho}$ are the Radon-Nikodym derivatives of $\eta^\pi(\cdot|x)$ and $\eta^\pi(\cdot|x)$ with respect to $\rho$. In the $C_1^\pi(\rho)$ defined above, the exponential term forces us to only focus on large values of $\beta$, which is not ideal. This term is appears as an upper bound for $\|\bar{\mathcal{P}}^\pi(\cdot|x)/\hat{\mathcal{P}}^\pi(\cdot|x)\|_\infty$. However, similar to more recent studies on approximate value iteration, it is possible to introduce coefficients that depend on the ratio of the expected values with respect to the two distribution instead of their densities. Due to the more involved nature of those definitions, we only include this simple form of results here and provide further discussion in the supplementary material. The next theorem shows the performance guarantees of our method in terms of weighted $\ell_p$ norms.

**Theorem 6.** *Define*

$$e_1^\pi = \frac{2\gamma}{1-\gamma} \cdot (C_1^\pi(\rho) + C_2^\pi(\rho)) \cdot \sqrt{\sqrt{2} \cdot \|\epsilon_{\mathrm{Model}}^\pi\|_{1,\rho} + \frac{2}{\beta} \cdot \left\|\epsilon_{\mathrm{Query}}^\pi\right\|_{1,\rho}},$$

$$e_2^\pi = \frac{2\gamma\sqrt{d}}{1-\gamma} \cdot C_2(\rho)\left(\beta\|\epsilon_{\mathrm{Model}}^\pi\|_{1,\rho} + 2\|\epsilon_{\mathrm{Query}}^\pi\|_{1,\rho}\right)$$

*Then*

$$\left\|V^{\pi_{PE}} - \bar{V}^{\pi_{PE}}\right\|_\infty \le \frac{2e_1^{\pi_{PE}}}{1-2e_1^{\pi_{PE}}} \cdot \inf_{\|w\|_\infty \le w_{max}} \left\|V^{\pi_{PE}} - \sum_{i=1}^d w_i \cdot \phi_i\right\|_{4,\rho} + \frac{2e_2^{\pi_{PE}}}{1-2e_1^{\pi_{PE}}} \cdot w_{max},$$

*also if $e_1^* = \max_{\pi \in \{\pi^*, \bar{\pi}^*\}} 6e_1^\pi/(1-2e_1^\pi)$ and $e_2^* = \max_{\pi \in \{\pi^*, \bar{\pi}^*\}} 6e_2^\pi/(1-2e_1^\pi)$, we have*

$$\left\|V^* - V^{\bar{\pi}^*}\right\|_{4,\rho} \le \frac{2e_1^*}{1-e_1^*} \inf_{\|w\|_\infty \le w_{max}} \left\|V^* - \sum_{i=1}^d w_i \cdot \phi_i\right\|_{4,\rho} + \frac{2e_2^*}{1-e_1^*} \cdot w_{max}.$$

Notice that the $\beta$ appears in the bound in the same manner as Theorem 5. This will lead to the same dynamics on the choice of $\beta$. We provide the proof of this theorem in Section H.

## H  PROOFS FOR $\ell_p$ ANALYSIS OF MAXENT MOCO

We first show some useful lemmas towards the proof of Theorem 6.

**Lemma 12.** *For $m$ functions $f_1, f_2, \ldots, f_m \colon \mathcal{X} \to \mathbb{R}$, we have*

$$\|f_1 + \cdots + f_m\|_{4,\rho}^4 \le m^3 \sum_{i=1}^m \|f_i\|_{4,\rho}^4.$$

**Proof.** We have

$$\|f_1 + \cdots + f_m\|_{4,\rho}^4 = \int \rho(\mathrm{d}x)\left(\sum_i f_i(x)\right)^4$$

$$\le \int \rho(\mathrm{d}x)\left[\left(\sum_i 1^{4/3}\right)^{3/4}\left(\sum_i f_i(x)^4\right)^{1/4}\right]^4$$

$$= m^3 \int \rho(\mathrm{d}x) \sum_i f_i(x)^4$$

$$= m^3 \sum_{i=1}^m \|f_i\|_{4,\rho}^4$$

$\square$

**Lemma 13.** *For any policy $\pi$, we have $G_{\bar{\mathcal{P}},\bar{\mathcal{P}}}^{\pi} = G_{\bar{\mathcal{P}},\mathcal{P}}^{\pi} G_{\mathcal{P},\bar{\mathcal{P}}}^{\pi} - G_{\bar{\mathcal{P}},\mathcal{P}}^{\pi}.$*

**Proof.** We write

$$
\begin{aligned}
G_{\bar{\mathcal{P}},\mathcal{P}}^{\pi} & G_{\mathcal{P},\bar{\mathcal{P}}}^{\pi} - G_{\bar{\mathcal{P}},\mathcal{P}}^{\pi} \\
&= (\mathbf{I} - \gamma\mathcal{P}^{\pi})^{-1}\Big((\gamma\bar{\mathcal{P}}^{\pi} - \gamma\mathcal{P}^{\pi})(\mathbf{I} - \gamma\bar{\mathcal{P}}^{\pi})^{-1} + \mathbf{I}\Big)(\gamma\mathcal{P}^{\pi} - \gamma\bar{\mathcal{P}}^{\pi}) \\
&= (\mathbf{I} - \gamma\mathcal{P}^{\pi})^{-1}\Big((\gamma\bar{\mathcal{P}}^{\pi} - \gamma\mathcal{P}^{\pi})(\mathbf{I} - \gamma\bar{\mathcal{P}}^{\pi})^{-1} + \mathbf{I}\Big)(\gamma\mathcal{P}^{\pi} - \gamma\bar{\mathcal{P}}^{\pi}) \\
&= (\mathbf{I} - \gamma\mathcal{P}^{\pi})^{-1}\Big((\gamma\bar{\mathcal{P}}^{\pi} - \gamma\mathcal{P}^{\pi})(\mathbf{I} - \gamma\bar{\mathcal{P}}^{\pi})^{-1} + (\mathbf{I} - \gamma\bar{\mathcal{P}}^{\pi})(\mathbf{I} - \gamma\bar{\mathcal{P}}^{\pi})^{-1}\Big)(\gamma\mathcal{P}^{\pi} - \gamma\bar{\mathcal{P}}^{\pi}) \\
&= (\mathbf{I} - \gamma\mathcal{P}^{\pi})^{-1}(\mathbf{I} - \gamma\mathcal{P}^{\pi})(\mathbf{I} - \gamma\bar{\mathcal{P}}^{\pi})^{-1}(\gamma\mathcal{P}^{\pi} - \gamma\bar{\mathcal{P}}^{\pi}) \\
&= (\mathbf{I} - \gamma\bar{\mathcal{P}}^{\pi})^{-1}(\gamma\mathcal{P}^{\pi} - \gamma\bar{\mathcal{P}}^{\pi}) \\
&= G_{\mathcal{P},\bar{\mathcal{P}}}^{\pi}.
\end{aligned}
$$

$\square$

**Lemma 14.** *For any $w \in \mathbb{R}^d$ we have*

$$
\Big\|(\mathcal{P}^{\pi} - \bar{\mathcal{P}}^{\pi})(\sum w_i\phi_i)\Big\|_{1,\rho} \leq \sqrt{d}\Big(2\big\|\epsilon_{\text{Query}}^{\pi}\big\|_{1,\rho} + \beta\big\|\epsilon_{\text{Model}}^{\pi}\big\|_{1,\rho}\Big) \cdot \|w\|_{\infty}.
$$

**Proof.** We write

$$
\begin{aligned}
\Big\|(\mathcal{P}^{\pi} - \bar{\mathcal{P}}^{\pi})(\sum w_i\phi_i)\Big\|_{1,\rho} &= \int \rho(\mathrm{d}x)\Big|\sum_i w_i\Big((\mathcal{P}^{\pi}\phi_i)(x) - (\bar{\mathcal{P}}^{\pi}\phi_i)(x)\Big)\Big| \\
&\leq \|w\|_{\infty}\int \rho(\mathrm{d}x)\sum_i\Big|\Big((\mathcal{P}^{\pi}\phi_i)(x) - (\bar{\mathcal{P}}^{\pi}\phi_i)(x)\Big)\Big| \\
&\leq \|w\|_{\infty}\int \rho(\mathrm{d}x)\Big[\sqrt{d}\int \pi(\mathrm{d}a|x)\Big(2\epsilon_{\text{Query}}(x,a) + \beta\epsilon_{\text{Model}}(x,a)\Big)\Big] \\
&= \|w\|_{\infty}\int \rho(\mathrm{d}x)\Big[\sqrt{d}\Big(2\epsilon_{\text{Query}}^{\pi}(x) + \beta\epsilon_{\text{Model}}^{\pi}(x)\Big)\Big] \\
&= \sqrt{d}\Big(2\big\|\epsilon_{\text{Query}}^{\pi}\big\|_{1,\rho} + \beta\big\|\epsilon_{\text{Model}}^{\pi}\big\|_{1,\rho}\Big) \cdot \|w\|_{\infty},
\end{aligned}
$$

where we used Lemma 10.

$\square$

**Lemma 15.** *Define*

$$
\mathrm{TV}_{\rho}^{\pi}(\mathcal{P}, \bar{\mathcal{P}}) \triangleq \int \rho(\mathrm{d}x)\big\|\mathcal{P}^{\pi}(\cdot|x) - \bar{\mathcal{P}}^{\pi}(\cdot|x)\big\|_1.
$$

*We have*

$$
\mathrm{TV}_{\rho}^{\pi}(\mathcal{P}, \bar{\mathcal{P}}) \leq \sqrt{2}\|\epsilon_{\text{Model}}^{\pi}\|_{1,\rho} + \frac{2}{\beta}\big\|\epsilon_{\text{Query}}^{\pi}\big\|_{1,\rho}.
$$

**Proof.** Using (C.1) in proof of Lemma 2 and Lemma 9 we have

$$
\begin{aligned}
\int \rho(\mathrm{d}x)\big\|\mathcal{P}^{\pi}(\cdot|x) - \bar{\mathcal{P}}^{\pi}(\cdot|x)\big\|_1 &\leq \int \rho(\mathrm{d}x)\int \pi(\mathrm{d}a|x)\big\|\mathcal{P}(\cdot|x,a) - \bar{\mathcal{P}}(\cdot|x,a)\big\|_1 \\
&\leq \int \rho(\mathrm{d}x)\int \pi(\mathrm{d}a|x)\Big[\sqrt{2}\epsilon_{\text{Model}}(x,a) + \frac{2}{\beta}\epsilon_{\text{Query}}(x,a)\Big] \\
&= \sqrt{2}\|\epsilon_{\text{Model}}^{\pi}\|_{1,\rho} + \frac{2}{\beta}\big\|\epsilon_{\text{Query}}^{\pi}\big\|_{1,\rho}.
\end{aligned}
$$

$\square$

**Lemma 16.** *Assume for any $i$ and $x, a$ we have $A - B/2 \leq U_i(x), Y_i(x, a) \leq A + B/2$ for some values $A$ and $B \geq 0$. Then we have*

$$\left\| \frac{\mathrm{d}\bar{\mathcal{P}}(\cdot|x,a)}{\mathrm{d}\hat{\mathcal{P}}(\cdot|x,a)} \right\|_\infty \leq \exp\left( \frac{2B^2 d}{\beta^2} \right).$$

**Proof.** Assume $\lambda$ is the dual problem of MaxEnt density estimation resulted in $\bar{\mathcal{P}}(\cdot|x, a)$. We have

$$\frac{\mathrm{d}\bar{\mathcal{P}}(\cdot|x,a)}{\mathrm{d}\hat{\mathcal{P}}(\cdot|x,a)}(y) = \exp\left( \sum_i \lambda_i \phi_i(y) - \Lambda_\lambda \right).$$

We have by Jensen's inequality

$$\begin{aligned}
\Lambda_\lambda &= \log \int \bar{\mathcal{P}}(\mathrm{d}y|x,a) \exp\left( \sum_i \lambda_i \phi_i(y) \right) \\
&\geq \int \bar{\mathcal{P}}(\mathrm{d}y|x,a) \log\left( \exp\left( \sum_i \lambda_i \phi_i(y) \right) \right) \\
&= \int \bar{\mathcal{P}}(\mathrm{d}y|x,a) \left( \sum_i \lambda_i \phi_i(y) \right) \\
&= \sum_i \lambda_i \cdot \mathbb{E}_{Y \sim \hat{\mathcal{P}}(\cdot|x,a)}[\phi_i(Y)].
\end{aligned}$$

Thus

$$\begin{aligned}
\frac{\mathrm{d}\bar{\mathcal{P}}(\cdot|x,a)}{\mathrm{d}\hat{\mathcal{P}}(\cdot|x,a)}(y) &= \exp\left( \sum_i \lambda_i \phi_i(y) - \Lambda_\lambda \right) \\
&\leq \exp\left( \sum_i \lambda_i (\phi_i(y) - \mathbb{E}_{Y \sim \hat{\mathcal{P}}(\cdot|x,a)}[\phi_i(Y)]) \right) \\
&\leq \exp(B\|\lambda\|_1).
\end{aligned}$$

Now to bound $\|\lambda\|_1$, note that for $\psi'(x,a) = \mathbb{E}_{Y \sim \hat{\mathcal{P}}(\cdot|x,a)}[\phi_i(Y)]$ the solution of (P2) is $\hat{\mathcal{P}}(\cdot|x,a)$ that corresponds to dual parameters $\lambda' = 0$. Using to Lemma 5,

$$\|\lambda\|_2 = \|\lambda - \lambda'\|_2 \leq \frac{2}{\beta^2} \|\psi(x,a) - \psi'(x,a)\|_2 \leq \frac{2}{\beta^2} \sqrt{d}B.$$

We get

$$\frac{\mathrm{d}\bar{\mathcal{P}}(\cdot|x,a)}{\mathrm{d}\hat{\mathcal{P}}(\cdot|x,a)}(y) \leq \exp(B\|\lambda\|_1) \leq \exp\left( B\sqrt{d}\|\lambda\|_2 \right) \leq \exp\left( \frac{2B^2 d}{\beta^2} \right).$$

$\square$

**Lemma 17.** *Let $e_1^\pi, e_2^\pi$ be defined as in Theorem 6. For any policy $\pi$, $v \colon \mathcal{X} \to \mathbb{R}$ and $w \in \mathbb{R}^d$ we have*

$$\left\| G_{\bar{\mathcal{P}}, \mathcal{P}}^\pi v \right\|_{4,\rho}^4 \leq (e_1^\pi)^4 \cdot \left\| v - \sum_i w_i \phi_i \right\|_{4,\rho}^4 + (e_2^\pi)^4 \cdot \|w\|_\infty^4.$$

**Proof.** Define

$$u \triangleq v - \sum_i w_i \phi_i$$

$$\bar{\omega}^\pi(\cdot|x) \triangleq \int \eta^\pi(\mathrm{d}z|x)\bar{\mathcal{P}}^\pi(\cdot|z)$$

$$\Delta\mathcal{P}(\mathrm{d}z|y) \triangleq \left|\mathcal{P}^\pi(\mathrm{d}z|y) - \bar{\mathcal{P}}^\pi(\mathrm{d}z|y)\right|$$

$$\mathrm{TV}_\rho \triangleq \int \rho(\mathrm{d}x)\left\|\mathcal{P}^\pi(\cdot|x) - \bar{\mathcal{P}}^\pi(\cdot|x)\right\|_1$$

$$D \triangleq (\mathbf{I} - \gamma\mathcal{P}^\pi)^{-1}(\gamma\bar{\mathcal{P}}^\pi - \gamma\mathcal{P}^\pi)(v - \sum_i w_i\phi_i)$$

$$E \triangleq (\mathbf{I} - \gamma\mathcal{P}^\pi)^{-1}(\gamma\bar{\mathcal{P}}^\pi - \gamma\mathcal{P}^\pi)(\sum_i w_i\phi_i).$$

We have

$$
\begin{aligned}
G^\pi_{\bar{\mathcal{P}},\mathcal{P}}v &= (\mathbf{I} - \gamma\mathcal{P}^\pi)^{-1}(\gamma\bar{\mathcal{P}}^\pi - \gamma\mathcal{P}^\pi)v \\
&= (\mathbf{I} - \gamma\mathcal{P}^\pi)^{-1}(\gamma\bar{\mathcal{P}}^\pi - \gamma\mathcal{P}^\pi)(v - \sum_i w_i\phi_i) + (\mathbf{I} - \gamma\mathcal{P}^\pi)^{-1}(\gamma\bar{\mathcal{P}}^\pi - \gamma\mathcal{P}^\pi)(\sum_i w_i\phi_i) \\
&= D + E.
\end{aligned}
$$

We bound norm of each term separately. For $A$, we write using the Cauchy–Schwarz inequality

$$
\begin{aligned}
\|D\|^4_{4,\rho} &= \int_x \rho(\mathrm{d}x)\left[\iint_{y,z} \frac{1}{1-\gamma}\eta^\pi(\mathrm{d}y|x)\cdot\gamma(\bar{\mathcal{P}}^\pi(\mathrm{d}z|y) - \mathcal{P}^\pi(\mathrm{d}z|y))\cdot u(z)\right]^4 \\
&\leq \int_x \rho(\mathrm{d}x)\left[\iint_{y,z} \frac{1}{1-\gamma}\eta^\pi(\mathrm{d}y|x)\cdot\gamma\Delta\mathcal{P}(\mathrm{d}z|y)\cdot|u(z)|\right]^4 \\
&= \frac{\gamma^4}{(1-\gamma)^4}\int_x \rho(\mathrm{d}x)\left[\iint_{y,z}\left(\sqrt{\rho(\mathrm{d}y)}\cdot\sqrt{\Delta\mathcal{P}(\mathrm{d}z|y)}\right)\left(\frac{\eta^\pi(\mathrm{d}y|x)\cdot|u(z)|\cdot\sqrt{\Delta\mathcal{P}(\mathrm{d}z|y)}}{\sqrt{\rho(\mathrm{d}y)}}\right)\right]^4 \\
&\leq \frac{\gamma^4}{(1-\gamma)^4}\int_x \rho(\mathrm{d}x)\left(\iint_{y,z}\rho(\mathrm{d}y)\cdot\Delta\mathcal{P}(\mathrm{d}z|y)\right)^2\cdot\left(\iint_{y,z}\frac{\eta^\pi(\mathrm{d}y|x)^2\cdot u(z)^2\cdot\Delta\mathcal{P}(\mathrm{d}z|y)}{\rho(\mathrm{d}y)}\right)^2 \\
&= \frac{\gamma^4}{(1-\gamma)^4}\cdot\mathrm{TV}^2_\rho\cdot\int_x \rho(\mathrm{d}x)\left(\iint_{y,z}\frac{\eta^\pi(\mathrm{d}y|x)^2\cdot u(z)^2\cdot\Delta\mathcal{P}(\mathrm{d}z|y)}{\rho(\mathrm{d}y)}\right)^2 \\
&= \frac{\gamma^4}{(1-\gamma)^4}\cdot\mathrm{TV}^2_\rho\cdot\int_x \rho(\mathrm{d}x)\left[\int_z\left(\sqrt{\rho(\mathrm{d}z)}\cdot u(z)^2\right)\cdot\left(\int_y\frac{\eta^\pi(\mathrm{d}y|x)^2\cdot\Delta\mathcal{P}(\mathrm{d}z|y)}{\sqrt{\rho(\mathrm{d}z)}\cdot\rho(\mathrm{d}y)}\right)\right]^2 \\
&\leq \frac{\gamma^4}{(1-\gamma)^4}\cdot\mathrm{TV}^2_\rho\cdot\int_x \rho(\mathrm{d}x)\left[\int_z \rho(\mathrm{d}z)u(z)^4\right]\left[\int_z\left(\int_y\frac{\eta^\pi(\mathrm{d}y|x)^2\cdot\Delta\mathcal{P}(\mathrm{d}z|y)}{\sqrt{\rho(\mathrm{d}z)}\cdot\rho(\mathrm{d}y)}\right)^2\right] \\
&= \frac{\gamma^4}{(1-\gamma)^4}\cdot\mathrm{TV}^2_\rho\cdot\|u\|^4_{4,\rho}\cdot\iint_{x,z}\rho(\mathrm{d}x)\left(\int_y\frac{\eta^\pi(\mathrm{d}y|x)^2\cdot\Delta\mathcal{P}(\mathrm{d}z|y)}{\sqrt{\rho(\mathrm{d}z)}\cdot\rho(\mathrm{d}y)}\right)^2 \\
&= \frac{\gamma^4}{(1-\gamma)^4}\cdot\mathrm{TV}^2_\rho\cdot\|u\|^4_{4,\rho}\cdot C.
\end{aligned}
$$

For $C$ we write

$$
\begin{aligned}
C &= \iint_{x,z} \rho(\mathrm{d}x)\left(\int_y \frac{\eta^\pi(\mathrm{d}y|x)^2 \cdot \Delta\mathcal{P}(\mathrm{d}z|y)}{\sqrt{\rho(\mathrm{d}z)\cdot\rho(\mathrm{d}y)}}\right)^2 \\
&= \int_x \rho(\mathrm{d}x)\int_z \rho(\mathrm{d}z)\left(\int_y \frac{\eta^\pi(\mathrm{d}y|x)^2 \Delta\mathcal{P}(\mathrm{d}z|y)}{\rho(\mathrm{d}y)\rho(\mathrm{d}z)}\right)^2 \\
&= \int_x \rho(\mathrm{d}x)\left\|\frac{\mathrm{d}\eta^\pi(\cdot|x)}{\mathrm{d}\rho}\right\|_\infty^2 \int_z \rho(\mathrm{d}z)\left(\int_y \frac{\eta^\pi(\mathrm{d}y|x)\Delta\mathcal{P}(\mathrm{d}z|y)}{\rho(\mathrm{d}z)}\right)^2 \\
&= \int_x \rho(\mathrm{d}x)\left\|\frac{\mathrm{d}\eta^\pi(\cdot|x)}{\mathrm{d}\rho}\right\|_\infty^2 \int_z \rho(\mathrm{d}z)\left(\frac{\int_y \eta^\pi(\mathrm{d}y|x)\mathcal{P}^\pi(\mathrm{d}z|y) + \int_y \eta^\pi(\mathrm{d}y|x)\bar{\mathcal{P}}^\pi(\mathrm{d}z|y)}{\rho(\mathrm{d}z)}\right)^2 \\
&= \int_x \rho(\mathrm{d}x)\left\|\frac{\mathrm{d}\eta^\pi(\cdot|x)}{\mathrm{d}\rho}\right\|_\infty^2 \int_z \rho(\mathrm{d}z)\left(\frac{\gamma^{-1}\eta^\pi(\mathrm{d}z|x)}{\rho(\mathrm{d}z)} + \frac{\bar{\omega}(\mathrm{d}z|y)}{\rho(\mathrm{d}z)}\right)^2 \\
&= 2\int_x \rho(\mathrm{d}x)\left\|\frac{\mathrm{d}\eta^\pi(\cdot|x)}{\mathrm{d}\rho}\right\|_\infty^2 \int_z \rho(\mathrm{d}z)\left[\left(\frac{\gamma^{-1}\eta^\pi(\mathrm{d}z|x)}{\rho(\mathrm{d}z)}\right)^2 + \left(\frac{\bar{\omega}(\mathrm{d}z|y)}{\rho(\mathrm{d}z)}\right)^2\right] \\
&= 2\int_x \rho(\mathrm{d}x)\left\|\frac{\mathrm{d}\eta^\pi(\cdot|x)}{\mathrm{d}\rho}\right\|_\infty^2 \int_z \rho(\mathrm{d}z)\left[\gamma^{-1}\left\|\frac{\mathrm{d}\eta^\pi(\cdot|x)}{\mathrm{d}\rho}\right\|_\infty^2 + \left\|\frac{\mathrm{d}\bar{\omega}(\cdot|y)}{\mathrm{d}\rho}\right\|_\infty^2\right] \\
&= 2\gamma^{-1}\int_x \rho(\mathrm{d}x)\left\|\frac{\mathrm{d}\eta^\pi(\cdot|x)}{\mathrm{d}\rho}\right\|_\infty^4 + 2\int_x \rho(\mathrm{d}x)\left\|\frac{\mathrm{d}\eta^\pi(\cdot|x)}{\mathrm{d}\rho}\right\|_\infty^2\left\|\frac{\mathrm{d}\bar{\omega}(\cdot|y)}{\mathrm{d}\rho}\right\|_\infty^2 \\
&\leq 2\gamma^{-1}\int_x \rho(\mathrm{d}x)\left\|\frac{\mathrm{d}\eta^\pi(\cdot|x)}{\mathrm{d}\rho}\right\|_\infty^4 + 2\exp\left(\frac{4B^2 d}{\beta^2}\right)\int_x \rho(\mathrm{d}x)\left\|\frac{\mathrm{d}\eta^\pi(\cdot|x)}{\mathrm{d}\rho}\right\|_\infty^2\left\|\frac{\mathrm{d}\omega(\cdot|y)}{\mathrm{d}\rho}\right\|_\infty^2 \\
&= 2C_2^\pi(\rho)^4 + 2C_1^\pi(\rho)^4,
\end{aligned}
$$

where we used Lemma 16. Also from Lemma 15 we have

$$
\mathrm{TV}_\rho \leq \sqrt{2}\|\epsilon_{\mathrm{Model}}^\pi\|_{1,\rho} + \frac{2}{\beta}\|\epsilon_{\mathrm{Query}}^\pi\|_{1,\rho}
$$

This means

$$
\begin{aligned}
\|D\|_{4,\rho}^4 &\leq \frac{2\gamma^4}{(1-\gamma)^4}\cdot(C_2^\pi(\rho)^4 + C_1^\pi(\rho)^4)\cdot\left(\sqrt{2}\|\epsilon_{\mathrm{Model}}^\pi\|_{1,\rho} + \frac{2}{\beta}\|\epsilon_{\mathrm{Query}}^\pi\|_{1,\rho}\right)^2\cdot\|u\|_{4,\rho}^4 \\
&\leq \frac{2\gamma^4}{(1-\gamma)^4}\cdot(C_2^\pi(\rho) + C_1^\pi(\rho))^4\cdot\left(\sqrt{2}\|\epsilon_{\mathrm{Model}}^\pi\|_{1,\rho} + \frac{2}{\beta}\|\epsilon_{\mathrm{Query}}^\pi\|_{1,\rho}\right)^2\cdot\|u\|_{4,\rho}^4.
\end{aligned}
$$

Now we bound the $E$ term. Define

$$
f(x) \triangleq \left|\mathbb{E}_{Y\sim\mathcal{P}^\pi(\cdot|x)}\left[\sum w_i\phi_i(Y)\right] - \mathbb{E}_{Y\sim\bar{\mathcal{P}}^\pi(\cdot|x)}\left[\sum w_i\phi_i(Y)\right]\right|.
$$

Using Lemma 14, we have

$$
\|f\|_{1,\rho} \leq \sqrt{d}\left(2\|\epsilon_{\mathrm{Query}}^\pi\|_{1,\rho} + \beta\|\epsilon_{\mathrm{Model}}^\pi\|_{1,\rho}\right)\cdot\|w\|_\infty. \tag{H.1}
$$

We have

$$
\begin{aligned}
\|E\|_{4,\rho}^4 &\leq \frac{\gamma^4}{(1-\gamma)^4}\int_x \rho(\mathrm{d}x)\left(\int_y \eta^\pi(\mathrm{d}y|x)f(y)\right)^4 \\
&\leq \frac{\gamma^4}{(1-\gamma)^4}\int_x \rho(\mathrm{d}x)\left\|\frac{\mathrm{d}\eta^\pi(\cdot|x)}{\mathrm{d}\rho}\right\|_\infty^4\left(\int_y \rho(\mathrm{d}y|x)f(y)\right)^4 \\
&\leq \frac{\gamma^4}{(1-\gamma)^4}\|f\|_{1,\rho}^4\int_x \rho(\mathrm{d}x)\left\|\frac{\mathrm{d}\eta^\pi(\cdot|x)}{\mathrm{d}\rho}\right\|_\infty^4 \\
&= \frac{\gamma^4}{(1-\gamma)^4}\cdot\gamma C_2^\pi(\rho)^4\cdot d^2\left(2\|\epsilon_{\mathrm{Query}}^\pi\|_{1,\rho} + \beta\|\epsilon_{\mathrm{Model}}^\pi\|_{1,\rho}\right)^4\cdot\|w\|_\infty^4.
\end{aligned}
$$

Putting things together using Lemma 12:

$$
\begin{aligned}
\left\|G^{\pi}_{\bar{\mathcal{P}},\mathcal{P}}v\right\|^4_{4,\rho} &= \|D+E\|^4_{4,\rho} \\
&\le 8\|D\|^4_{4,\rho} + 8\|E\|^4_{4,\rho} \\
&\le \frac{16\gamma^4}{(1-\gamma)^4}\cdot(C^{\pi}_2(\rho)+C^{\pi}_1(\rho))^4\cdot\left(\sqrt{2}\|\epsilon^{\pi}_{\text{Model}}\|_{1,\rho}+\frac{2}{\beta}\|\epsilon^{\pi}_{\text{Query}}\|_{1,\rho}\right)^2\cdot\|u\|^4_{4,\rho} \\
&\quad +\frac{8\gamma^4}{(1-\gamma)^4}\cdot\gamma C^{\pi}_2(\rho)^4\cdot d^2\left(2\|\epsilon^{\pi}_{\text{Query}}\|_{1,\rho}+\beta\|\epsilon^{\pi}_{\text{Model}}\|_{1,\rho}\right)^4\cdot\|w\|^4_{\infty} \\
&\le (e^{\pi}_1)^4\cdot\|u\|^4_{4,\rho}+(e^{\pi}_2)^4\cdot\|w\|^4_{\infty}.
\end{aligned}
$$

$\square$

**Proof of Theorem 6 for PE**

**Proof.** We have

$$
\begin{aligned}
\left\|V^{\pi_{\text{PE}}}-\bar{V}^{\pi_{\text{PE}}}\right\|_{4,\rho} &= \left\|G^{\pi_{\text{PE}}}_{\mathcal{P},\bar{\mathcal{P}}}V^{\pi_{\text{PE}}}\right\|_{4,\rho} \\
&= \left\|G^{\pi_{\text{PE}}}_{\bar{\mathcal{P}},\mathcal{P}}G^{\pi_{\text{PE}}}_{\mathcal{P},\bar{\mathcal{P}}}V^{\pi_{\text{PE}}}-G^{\pi_{\text{PE}}}_{\bar{\mathcal{P}},\mathcal{P}}V^{\pi_{\text{PE}}}\right\|_{4,\rho} \\
&\le 2^{3/4}\left\|G^{\pi_{\text{PE}}}_{\bar{\mathcal{P}},\mathcal{P}}G^{\pi_{\text{PE}}}_{\mathcal{P},\bar{\mathcal{P}}}V^{\pi_{\text{PE}}}\right\|_{4,\rho}+2^{3/4}\left\|G^{\pi_{\text{PE}}}_{\bar{\mathcal{P}},\mathcal{P}}V^{\pi_{\text{PE}}}\right\|_{4,\rho} \\
&\le 2\left\|G^{\pi_{\text{PE}}}_{\bar{\mathcal{P}},\mathcal{P}}G^{\pi_{\text{PE}}}_{\mathcal{P},\bar{\mathcal{P}}}V^{\pi_{\text{PE}}}\right\|_{4,\rho}+2\left\|G^{\pi_{\text{PE}}}_{\bar{\mathcal{P}},\mathcal{P}}V^{\pi_{\text{PE}}}\right\|_{4,\rho}.
\end{aligned}
$$

Using Lemma 17 with $w=0$ we have

$$
\begin{aligned}
\left\|G^{\pi_{\text{PE}}}_{\bar{\mathcal{P}},\mathcal{P}}G^{\pi_{\text{PE}}}_{\mathcal{P},\bar{\mathcal{P}}}V^{\pi_{\text{PE}}}\right\|_{4,\rho} &\le e^{\pi_{\text{PE}}}_1\left\|G^{\pi_{\text{PE}}}_{\mathcal{P},\bar{\mathcal{P}}}V^{\pi_{\text{PE}}}\right\|_{4,\rho} \\
&\le e^{\pi_{\text{PE}}}_1\left\|V^{\pi_{\text{PE}}}-\bar{V}^{\pi_{\text{PE}}}\right\|_{4,\rho}.
\end{aligned}
$$

Also from Lemma 17 we have

$$
\begin{aligned}
\left\|G^{\pi_{\text{PE}}}_{\bar{\mathcal{P}},\mathcal{P}}V^{\pi_{\text{PE}}}\right\|_{4,\rho} &\le \left((e^{\pi_{\text{PE}}}_1)^4\left\|V^{\pi_{\text{PE}}}-\sum w_i\phi_i\right\|^4_{4,\rho}+(e^{\pi_{\text{PE}}}_2)^4\|w\|^4_{\infty}\right)^{1/4} \\
&\le e^{\pi_{\text{PE}}}_1\left\|V^{\pi_{\text{PE}}}-\sum w_i\phi_i\right\|_{4,\rho}+e^{\pi_{\text{PE}}}_2\|w\|_{\infty}
\end{aligned}
$$

with substitution we get

$$
\left\|V^{\pi_{\text{PE}}}-\bar{V}^{\pi_{\text{PE}}}\right\|_{4,\rho}\le 2e^{\pi_{\text{PE}}}_1\left\|V^{\pi_{\text{PE}}}-\bar{V}^{\pi_{\text{PE}}}\right\|_{4,\rho}+2e^{\pi_{\text{PE}}}_1\left\|V^{\pi_{\text{PE}}}-\sum w_i\phi_i\right\|_{4,\rho}+2e^{\pi_{\text{PE}}}_2\|w\|_{\infty}.
$$

Rearranging the terms give the result.

$\square$

**Proof of Theorem 6 for Control**

**Proof.** From proof of (C.3), we get

$$
\begin{aligned}
\left\|V^{\pi^*}-V^{\bar{\pi}^*}\right\|_{4,\rho} &\le \left\|\left|G^{\pi^*}_{\mathcal{P},\hat{\mathcal{P}}}V^*\right|+\left|G^{\bar{\pi}^*}_{\mathcal{P},\hat{\mathcal{P}}}V^*\right|+\left|G^{\bar{\pi}^*}_{\mathcal{P},\hat{\mathcal{P}}}(V^*-V^{\bar{\pi}^*})\right|\right\|_{4,\rho} \\
&\le 3\left\|G^{\pi^*}_{\mathcal{P},\hat{\mathcal{P}}}V^*\right\|_{4,\rho}+3\left\|G^{\bar{\pi}^*}_{\mathcal{P},\hat{\mathcal{P}}}V^*\right\|_{4,\rho}+3\left\|G^{\bar{\pi}^*}_{\mathcal{P},\hat{\mathcal{P}}}(V^*-V^{\bar{\pi}^*})\right\|_{4,\rho}.
\end{aligned}
$$

From Lemma 17 with $w=0$

$$
3\left\|G^{\bar{\pi}^*}_{\mathcal{P},\hat{\mathcal{P}}}(V^*-V^{\bar{\pi}^*})\right\|_{4,\rho}\le e^*_1\left\|V^*-V^{\bar{\pi}^*}\right\|_{4,\rho}.
$$

Also for any $w$

$$3\left\|G_{\mathcal{P},\hat{\mathcal{P}}}^{\pi^*}V^*\right\|_{4,\rho} \le e_1^*\left\|V^* - \sum w_i\phi_i\right\|_{4,\rho} + e_2^*\|w\|_\infty$$

$$3\left\|G_{\mathcal{P},\hat{\mathcal{P}}}^{\bar{\pi}^*}V^*\right\|_{4,\rho} \le e_1^*\left\|V^* - \sum w_i\phi_i\right\|_{4,\rho} + e_2^*\|w\|_\infty.$$

Thus,

$$\left\|V^{\pi^*} - V^{\bar{\pi}^*}\right\|_{4,\rho} \le 2e_1^*\left\|V^* - \sum w_i\phi_i\right\|_{4,\rho} + 2e_2^*\|w\|_\infty + e_1^*\left\|V^* - V^{\bar{\pi}^*}\right\|_{4,\rho}.$$

Rearranging proves the result. $\qquad\square$

# I  PROOFS FOR SECTION 4

Here, we give the proof of Theorem 2 after the following lemma.

**Lemma 18.** *If $V_k$ is the value function at iteration $k$ of MoCoVI for control. Let $\beta, \epsilon_{\text{Query}}^\infty$ be defined as in Theorem 2. We have*

$$\|V_k - V^*\|_\infty \le 3c_1\|\epsilon_{\text{Model}}\|_\infty \inf_{\|w\|_\infty \le w_{\max}}\left\|V^* - \sum w_i\phi_i\right\|_\infty + c_2\left\|\epsilon_{\text{Query}}^\infty\right\|_\infty w_{\max}.$$

**Proof.** Let $\bar{\mathcal{P}}_k$ be the corrected transition dynamics used to obtain $V_k$. Let $r_k = r + (\gamma\mathcal{P} - \gamma\bar{\mathcal{P}}_k)V^*$. According to Rakhsha et al. (2022), $V^* = V^*(r_k, \bar{\mathcal{P}}_k) = V^{\pi^*}(r_k, \bar{\mathcal{P}}_k)$. Now we have

$$\begin{aligned}
V^* - V_k &= V^*(r_k, \bar{\mathcal{P}}_k) - V^{\pi_k}(r, \bar{\mathcal{P}}_k)\\
&\succcurlyeq V^{\pi_k}(r_k, \bar{\mathcal{P}}_k) - V^{\pi_k}(r, \bar{\mathcal{P}}_k)\\
&= (\mathbf{I} - \gamma\bar{\mathcal{P}}^{\pi_k})^{-1}(r_k^{\pi_k} - r^{\pi_k})\\
&= (\mathbf{I} - \gamma\bar{\mathcal{P}}^{\pi_k})^{-1}(\gamma\mathcal{P}^{\pi_k} - \gamma\bar{\mathcal{P}}^{\pi_k})V^*\\
&= G_{\mathcal{P},\bar{\mathcal{P}}_k}^{\pi_k}V^*.
\end{aligned}$$

On the other hand

$$\begin{aligned}
V^* - V_k &= V^{\pi^*}(r_k, \bar{\mathcal{P}}_k) - V^*(r, \bar{\mathcal{P}}_k)\\
&\preccurlyeq V^{\pi^*}(r_k, \bar{\mathcal{P}}_k) - V^{\pi^*}(r, \bar{\mathcal{P}}_k)\\
&= (\mathbf{I} - \gamma\bar{\mathcal{P}}^{\pi^*})^{-1}(r_k^{\pi^*} - r^{\pi^*})\\
&= (\mathbf{I} - \gamma\bar{\mathcal{P}}^{\pi^*})^{-1}(\gamma\mathcal{P}^{\pi^*} - \gamma\bar{\mathcal{P}}^{\pi^*})V^*\\
&= G_{\mathcal{P},\bar{\mathcal{P}}_k}^{\pi^*}V^*.
\end{aligned}$$

Thus,

$$\begin{aligned}
\|V^* - V_k\|_\infty &\le \max\left(\left\|G_{\mathcal{P},\bar{\mathcal{P}}_k}^{\pi_k}V^*\right\|_\infty, \left\|G_{\mathcal{P},\bar{\mathcal{P}}_k}^{\pi^*}V^*\right\|_\infty\right)\\
&\le 3c_1\|\epsilon_{\text{Model}}\|_\infty \inf_{\|w\|_\infty \le w_{\max}}\left\|V^* - \sum w_i\phi_{k+i}\right\|_\infty + c_2\left\|\epsilon_{\text{Query}}^\infty\right\|_\infty w_{\max},
\end{aligned}$$

where the last inequality is from Theorem 1. $\qquad\square$

**Proof of Theorem 2**

**Proof.**

For PE, we note that from Theorem 1 we have for any $K \le k \ge 1$

$$\begin{aligned}
\|V^{\pi_{\text{PE}}} - V_k\|_\infty &\le 3c_1\|\epsilon_{\text{Model}}\|_\infty \inf_{\|w\|_\infty \le w_{\max}}\left\|V^{\pi_{\text{PE}}} - \sum w_i\phi_{k+i}\right\|_\infty + c_2\left\|\epsilon_{\text{Query}}^\infty\right\|_\infty w_{\max}\\
&\le \gamma'\|V^{\pi_{\text{PE}}} - V_{k-1}\|_\infty + c_2\left\|\epsilon_{\text{Query}}^\infty\right\|_\infty w_{\max}.
\end{aligned}$$

By induction, we get

$$\|V^{\pi_{\text{PE}}} - V_K\|_\infty \le \gamma'^K \|V^{\pi_{\text{PE}}} - V_0\|_\infty + \frac{1 - \gamma'^K}{1 - \gamma'} c_2 \|\epsilon_{\text{Query}}^\infty\|_\infty w_{\text{max}}.$$

For control, note that according to Lemma 18, for $1 \le k \le K$

$$\|V^* - V_k\|_\infty \le 3c_1 \|\epsilon_{\text{Model}}\|_\infty \inf_{\|w\|_\infty \le w_{\text{max}}} \left\|V^* - \sum w_i \phi_{k+i}\right\|_\infty + c_2 \|\epsilon_{\text{Query}}^\infty\|_\infty w_{\text{max}}$$

$$\le \gamma' \|V^* - V_{k-1}\|_\infty + c_2 \|\epsilon_{\text{Query}}^\infty\|_\infty w_{\text{max}}.$$

Consequently

$$\|V^* - V_{K-1}\|_\infty \le \gamma'^{K-1} \|V^* - V_0\|_\infty + \frac{1 - \gamma'^{K-1}}{1 - \gamma'} c_2 \|\epsilon_{\text{Query}}^\infty\|_\infty w_{\text{max}}.$$

Finally, based on Theorem 1,

$$\|V^* - V^{\pi_K}\|_\infty \le \frac{6c_1 \|\epsilon_{\text{Model}}\|_\infty}{1 - 3c_1 \|\epsilon_{\text{Model}}\|_\infty} \inf_{\|w\|_\infty \le w_{\text{max}}} \left\|V^* - \sum_{i=1}^d w_i \phi_{i+K}\right\|_\infty$$

$$+ \frac{2c_2 \|\epsilon_{\text{Query}}\|_\infty}{1 - 3c_1 \|\epsilon_{\text{Model}}\|_\infty} \cdot w_{\text{max}}$$

$$\le \frac{2}{1 - 3c_1 \|\epsilon_{\text{Model}}\|_\infty} \gamma' \left\|V^* - V_{K-1}\right\|_\infty + \frac{2c_2 \|\epsilon_{\text{Query}}\|_\infty}{1 - 3c_1 \|\epsilon_{\text{Model}}\|_\infty} \cdot w_{\text{max}}$$

$$\le \frac{2\gamma'^K}{1 - 3c_1 \|\epsilon_{\text{Model}}\|_\infty} \left\|V^* - V_0\right\|_\infty + \frac{1 - \gamma'^K}{1 - \gamma'} \frac{2c_2 \|\epsilon_{\text{Query}}\|_\infty}{1 - 3c_1 \|\epsilon_{\text{Model}}\|_\infty} w_{\text{max}}.$$

$\square$

## J  ADDITIONAL EMPIRICAL DETAILS

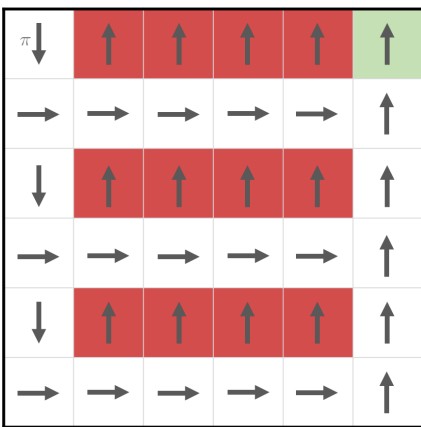

Figure 2: Modified Cliffwalk environment (Rakhsha et al., 2022).

We perform our experiments on a $6 \times 6$ gridworld environment introduced by Rakhsha et al. (2022). The environment is shown in Figure 2. There are 4 actions in the environment: (UP, RIGHT, DOWN, LEFT). When an action is taken, the agent moves towards that direction with probability $0.9$. With probability of $0.1$ it moves towards another direction at random. If the agent attempts to exit the environment, it stays in place. The middle 4 states of the first, third, and fifth row are *cliffs*. If the agent falls into a cliff, it stays there permanently and receives reward of $-32, -16, -8$ every iterations for the first, third, and fifth row cliffs, respectively. The top-right corner is the goal state, which awards reward of 20 once reached. We consider this environment with $\gamma = 0.9$.

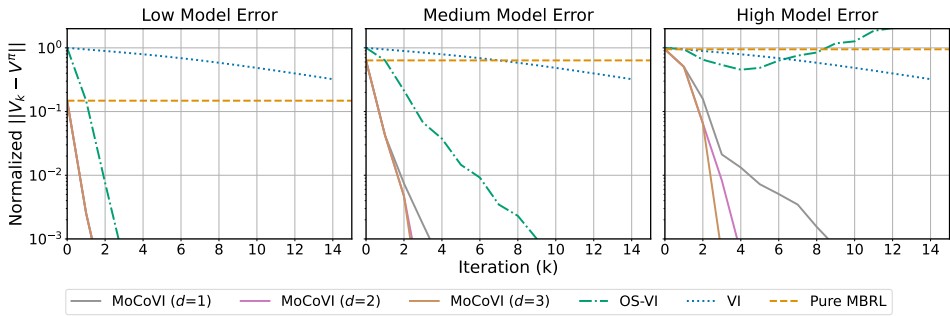

Figure 3: Policy evaluation results comparing MoCoVI with VI, pure MBRL and OS-VI. *(Left)* low ($\lambda = 0.1$), *(Middle)* medium ($\lambda = 0.5$), and *(Right)* high ($\lambda = 1$) model errors. Each curve is average of 20 runs. Shaded areas show the standard error.

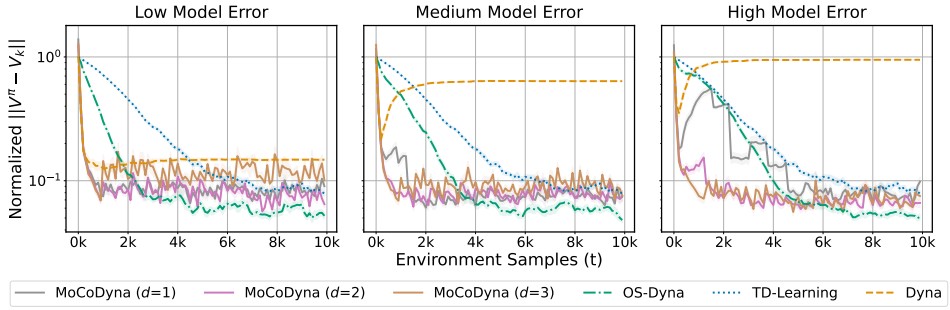

Figure 4: Policy evaluation results comparing MoCoDyna with Dyna, OS-Dyna and TD-learning. *(Left)* low ($\lambda = 0.1$), *(Middle)* medium ($\lambda = 0.5$), and *(Right)* high ($\lambda = 1$) model errors. Each curve is average of 20 runs. Shaded areas show the standard error.

For MoCoVI, we set the initial measurement functions $\phi_i$ for $i = 1, \cdots, d$ constant zero functions. We can set $\psi_i = 0$ for $i = 1, \cdots, d$ without querying $\mathcal{P}$. This makes the comparison of algorithms fair as MoCoVI is not given extra queries before the first iteration. The convergence of MoCoVI with exact queries and $\beta = 0$ is shown in Figures 1 and 3 for the control and PE problems.

Figures 1 and 4 show the performance of MoCoDyna compared to other algorithms in the PE and control problems. As discussed after Theorem 1, it is beneficial to choose measurement functions such that the true value function can be approximated with $\sum_i w_i \phi_i$ for some small weights $w_i$. To achieve this in our implementation, we initialize $\phi_{1:d+c}$ with an orthonormal set of functions. Also, in line 9 of Algorithm 1, we maintain this property of measurement functions by subtracting the projection of the new value function $V_t$ onto the span of the previous $d - 1$ functions before adding it to the measurement functions. We have

$$\phi_{d+c} \leftarrow V_t - \sum_{i=c+1}^{d+c-1} \langle \phi_i, V_t \rangle \cdot \phi_i, \tag{J.1}$$

and then we normalize $\phi_{d+c}$ to have a fixed euclidean norm. The hyperparameters of MoCoDyna for PE and control problems are given in Tables 3 and 4.

**Model Error Reduction.**  To show that the model correction procedure in MoCoDyna improves the accuracy of the model, we plot the error of original and corrected dynamics in the control problem in Figure 5. The model error is measured by taking the average of $\|\mathcal{P}(\cdot|x, a) - \hat{\mathcal{P}}(\cdot|x, a)\|_1$ or $\|\mathcal{P}(\cdot|x, a) - \bar{\mathcal{P}}(\cdot|x, a)\|_1$ over all $x, a$. We observe that higher order correction better reduces the error.

**Computation Cost.**  In Table 1 we provide the average time the calculation of $\bar{\mathcal{P}}$ has taken in MoCoDyna in the control problem. This is total time to calculate $\bar{\mathcal{P}}(\cdot|x, a)$ for all 144 state-action

pairs in the environment. In our implementation, the dual variables of the optimization problem for all state-action pairs are optimized with a single instance of the BFGS algorithm in SciPy library. Note that in general, different instances of the optimization problem (P2) for a batch of state-action pairs can be solved in parallel to reduce the computation time. Table 2 shows the full run time of the algorithms. It is important to note that in Algorithm 1, apart from reporting the current policy for the purpose of evaluation in line 6, MoCoDyna only needs to plan with $\bar{\mathcal{P}}$ every $K$ steps to have $V_t$ in line 9. In our implementation, planning is done every 2000 steps to evaluate the algorithm. Performing the planning only when needed in line 9 would make the algorithm computationally faster.

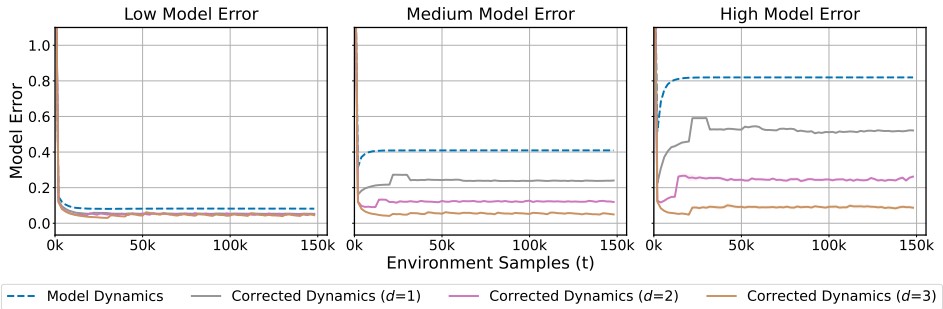

Figure 5: Comparison of the error of the original uncorrected model compared to error of corrected dynamics in the PE problem. *(Left)* low ($\lambda = 0.1$), *(Middle)* medium ($\lambda = 0.5$), and *(Right)* high ($\lambda = 1$) model errors. Each curve is average of 10 runs. Shaded areas show the standard error.

Table 1: Average computation time (seconds) of $\bar{\mathcal{P}}$ during a run of algorithms in the control problem for low ($\lambda = 0.1$), medium ($\lambda = 0.5$), and high ($\lambda = 1$) model errors.

|  | MoCoDyna1 | MoCoDyna2 | MoCoDyna3 |
|---|---|---|---|
| $\lambda = 0.1$ | 0.24 | 0.58 | 1.51 |
| $\lambda = 0.5$ | 0.29 | 0.52 | 1.39 |
| $\lambda = 1$ | 0.2 | 0.5 | 1.44 |

Table 2: Run time (seconds) for a single run of algorithms in the control problem for low ($\lambda = 0.1$), medium ($\lambda = 0.5$), and high ($\lambda = 1$) model errors.

|  | TD Learning | Dyna | OS-Dyna | MoCoDyna1 | MoCoDyna2 | MoCoDyna3 |
|---|---|---|---|---|---|---|
| $\lambda = 0.1$ | 44 | 50 | 555 | 119 | 134 | 200 |
| $\lambda = 0.5$ | 44 | 34 | 565 | 113 | 114 | 169 |
| $\lambda = 1$ | 44 | 33 | 600 | 91 | 110 | 172 |

Table 3: Hyperparamters for the PE problem. Cells with multiple values provide the value of the hyperparameter for different model errors with $\lambda = 0.1$, $\lambda = 0.5$, and $\lambda = 1$, respectively.

|  | TD Learning | OS-Dyna | MoCoDyna1 | MoCoDyna2 | MoCoDyna3 |
|---|---|---|---|---|---|
| learning rate | 0.2 | 0.05, 0.05, 0.05 | - | - | - |
| $c$ | - | - | 2, 2, 2 | 2, 2, 2 | 2, 2, 2 |
| $\beta$ | - | - | 0.02, 0.02, 0.02 | 0.16, 0.16, 0.16 | 0.14, 0.14, 0.14 |
| $K$ | - | - | 250, 400, 750 | 300, 300, 400 | 300, 300, 400 |

Table 4: Hyperparamters for the control problem. Cells with multiple values provide the value of the hyperparameter for different model errors with $\lambda = 0.1$, $\lambda = 0.5$, and $\lambda = 1$, respectively.

|  | TD Learning | OS-Dyna | MoCoDyna1 | MoCoDyna2 | MoCoDyna3 |
|---|---|---|---|---|---|
| learning rate | 0.2 | $0.02, 0.02, 0.02$ | - | - | - |
| $c$ | - | - | $2, 2, 2$ | $2, 2, 2$ | $2, 2, 2$ |
| $\beta$ | - | - | $0.02, 0.02, 0.02$ | $0.16, 0.16, 0.16$ | $0.14, 0.14, 0.14$ |
| $K$ | - | - | $10k, 10k, 10k$ | $6k, 6k, 6k$ | $10k, 10k, 10k$ |

