# OpenReview forum: "Maximum Entropy Model Correction in Reinforcement Learning"
_ICLR.cc/2024/Conference — ICLR 2024 poster_

### Official Review · Reviewer_gS8c · 2023-10-31

**Soundness:** 3 good
**Presentation:** 3 good
**Contribution:** 4 excellent
**Rating:** 8
**Confidence:** 4

**Summary:**

This paper proposes Value Iteration (VI) and Dyna-style sample-based version utilizing an approximate environmental model. Under the assumption that the approximate model $\hat{\mathcal{P}}$ is given, a corrected model $\bar{\mathcal{P}}$ is derived by minimizing the KL divergence $D_{KL} ( \cdot \parallel \hat{\mathcal{P}})$ with feature matching constraints. The proposed method, MoCoVI and MoCoDyna, employ the corrected dynamics to learn a state value function. Then, the value function trained with the corrected dynamics is used as a new basis function used in the feature matching constraints. MoCoVI and MoCoDyna are compared with OS-VI and OS-Dyna, respectively, on the modified Cliffwalk environment. The experimental results show that the proposed methods converge to the true value function faster than the baselines.

**Strengths:**

The motivation is well explained, and the paper is overall well-written and well-organized. The proofs given in the supplementary materials are complicated, but the authors provide a detailed explanation.

**Weaknesses:**

The proposed method is evaluated on the modified Cliffwalk environment that is relatively small. I am unsure if the proposed method is scalable.

**Questions:**

1. At the $k$-th iteration, the proposed method uses the basis functions $\phi_{k+1:k+d}$ and the query results $\psi_{k+1:k+d}$ to obtain $V^k$. I would like to know why $\psi_{1:k}$ and $\psi_{1:k}$ are discarded. Would you discuss this point?
2. The function approximator for the value function is unclear. Is $V$ approximated by $\sum_{i}^d w_i \phi_i$? If so, adding $V$ as a new basis function at the next iteration is problematic because it is linearly dependent on the basis functions.
3. On page 4, the authors mentioned that the number of basis functions, $d$, is usually small, but I am unsure whether it is true in general. I do not know the details of the modified Cliffwalk environment, but I think it is a toy problem. Is $d$ still small if the proposed method is applied to tasks with huge state space?

The following are minor comments.
- The end of the second paragraph in Section 2.2: $\phi(Z)$ should be $\phi_i(Z)$.
- The paragraph below equation (3.2): I do not understand the following sentence: $\bar{\mathcal{P}}$ is not constructed by the agent.
- In the first paragraph of Section 3.2, the authors define the function $\phi: \mathcal{X} \to \mathbb{R}^d$, but $\phi (x, a)$ is used to compute $\epsilon_{\mathrm{Model}}(x, a)$. Is it a typo?

---

> ### Author Response · Authors · 2023-11-22
> **Response to Reviewer gS8c**
>
> We thank the reviewer for their positive review of the paper.
>
> > I am unsure if the proposed method is scalable.
>
> In terms of scalability and the cost of optimization procedure in MaxEnt MoCo, we point out three properties that we believe makes the procedure scalable:
>
> 1. **Scale of the dual problem**: The dual problem of optimization problems (P1) and (P2) has only $d$ parameters. This means that it does not scale with the size of MDP. The value of $d$ will be kept rather small, since the agent needs to store $d$ functions for $\phi_i$ and $\psi_i$. Moreover, the dual problem of (P2), which is the version implemented in practice, is $\beta^2/4$-strongly convex (See equation D.15). The convexity allows efficiently solving the problem.
>
> 2. **Parallelization:** Sampling from the model is usually done in batches. Correcting each of these samples is an independent optimization problem that can be parallelized with careful implementation.
>
> 3. **Adjusting the computation cost:** It is possible to adjust the amount of correction to meet the computational budget. Dual parameters equal to zero correspond to no correction and using the model itself. We expect that partially optimizing the dual parameters and making few updates on them still partially corrects the model. Another technique is to use stronger regularization of dual parameters by choosing a larger value for the hyper parameter $\beta$, which makes the loss more strongly convex and easier to optimize. We provide an analysis of MaxEnt MoCo for the general choice of $\beta$ in Theorem 5 and 6.
>
> We have included the runtimes of our experiments in the new revision of the paper. We have also provided the computation time for the correction procedure, which is 0.2 - 1.5 seconds for all 144 state-actions depending on the setting.
>
> ### **Question 1:**
>
> The main reason for discarding $\phi_{1:k}$ and $\psi_{1:k}$ is solely to conserve memory. Otherwise, the required memory of the algorithm would grow by each iteration as $O(k)$. One other minor reason is that when the number of basis functions grows, the second term in the optimization problem (P2) will dominate the first term. This can be fixed by adjusting the parameter $\beta$ accordingly, but it would make the algorithm more complicated.
>
> ### **Question 2:**
>
> No, we never construct any value function as a linear combination of the basis functions. The performance of the algorithm does depend on how well the true value function is approximated with a linear combination of the basis functions. This is only for the purpose of theoretical analysis. The distance between the true value function and the span of basis functions is not necessarily zero and appears in our bounds. More importantly, we add the new value function obtained from planning to the basis functions which is different from the true value function. Yet, such issues can be avoided with a properly designed BasisCreation function in line 9 of the MoCoDyna algorithm, for example by subtracting the part of the new value function that belongs to the span of previous basis functions.
>
> ### **Question 3:**
>
> If the queries are accurate, the correction of MaxEnt MoCo and the convergence rate of MoCoVI improve with a larger value of $d$.  However, the main benefits of our methods are achievable even with $d=1$. For example, our theory shows that if the model is accurate enough, MoCoVI can converge to the true value function and do so faster than VI. This holds even for $d=1$ and large/continuous MDPs. Also, a single constant basis function for $d=1$ might be effective in MaxEnt MoCo (See proposition 1 and lemma 1).
>
> In large continuous MDPs used in deep RL, the query result $\psi_i$ will probably be obtained by training a neural network to estimate $E[\phi(X’_i)]$ at next state $X_i’$ from the input $(X_i, A_i)$ as a regression problem. This means that $d$ will determine the number of heads of a neural network (if a shared network is used for all $\phi_i$ functions) or the number of networks (if a separate network is trained for each $\psi_i$). The choice of $d$ is a design decision. It might be beneficial to use the computation resources to train a better/larger model instead of training a huge network for queries and doing a high amount of computation for the correction procedure. For these reasons, we expect the optimal choice of $d$ to be small even in large environments.
>
> > The paragraph below equation (3.2): I do not understand the following sentence: $\bar P$ is not constructed by the agent.
>
> It means that the agent does not store $\bar P$ as a function of state-actions to next-state distributions the same way it does for $\hat P$. For example if the model $\hat P$ is a neural network, we do not create a new network for $\bar P$. Instead, we compute the distribution $\bar P(\cdot | x,a)$ when needed by solving the optimization problem (P1) or (P2).

---

### Official Review · Reviewer_sy6v · 2023-11-01

**Soundness:** 3 good
**Presentation:** 3 good
**Contribution:** 3 good
**Rating:** 8
**Confidence:** 3

**Summary:**

The paper proposes a new method for MBRL, where a correction of the model is learnt. The correction improves the distribution of the next state prediction. The work consists of a model correcting approach, and its application to value iteration and to a dyna-based approach. A benefit of the approach is the combination of a model with a faster convergence thanks to the use of a model. The model-correcting prediction helps to keep the model close to the true dynamics of the environment.

**Strengths:**

The work is well motivated and the introduction / background are concise yet provide enough detail to set the scene. The paper is well, logically written. At times, it leads to a question (eg isn't computing this for all (x,a) expensive?, in the intro), to later answer the question. The approach is interesting and improves over conventional approaches. Good theoretical contributions.

The appendices are useful, equally logically structured, and contain additional analysis and proofs.

**Weaknesses:**

The main weakness in my view is the limited number of experiments, moreover the experiments are relatively simple.  I would have liked to see how the approach is used and how its performs (also in terms of time) for large or continuous environments (which seems to be planned for future work).

I'm not quite sure about the related work and comparison to approaches that use multiple models to improve predictions, which would be a more appropriate comparison (apart from OS-VI, eg residual models or even ensemble models), along with a comparison of computational costs (during training or inference). Even the additional empirical results in the appendix are not particularly large.

**Questions:**

- In the introduction, you say MaxEnt MoCo first obtains $E[\phi_i(X')]$ for all $(x,a)$. That sounds expensive and seems to be computed on demand; but what is the cost of solving the lazy approximation (P1).
- Can you comment about computational cost and scalablity of the approach given that the experiments were limited?
- I may have missed it but would it make sense to compare how close the dynamics of the real environment with the predicted and corrected one? If that's the case, evaluations of how well the environment can be predicted might be helpful even without additional results from RL experiments.

---

> ### Author Response · Authors · 2023-11-22
> **Response to Reviewer sy6v**
>
> We thank the reviewer for their positive review of the paper.
>
> > The main weakness in my view is the limited number of experiments, moreover the experiments are relatively simple. I would have liked to see how the approach is used and how its performs (also in terms of time) for large or continuous environments (which seems to be planned for future work).
>
> We agree with the reviewer that more extensive experiments can be very insightful for our proposed approach.
>
> > I'm not quite sure about the related work and comparison to approaches that use multiple models to improve predictions, which would be a more appropriate comparison (apart from OS-VI, eg residual models or even ensemble models), along with a comparison of computational costs (during training or inference). Even the additional empirical results in the appendix are not particularly large.
>
> Surely there’s a large body of research on dealing with model error in MBRL. Many of these efforts can be considered ways to improve the approximate dynamics $\hat P$. Consequently our method can be implemented along with them. The empirical comparison of the gains achieved with these techniques to our method is more reasonable in deep RL experiments and is to be studied in future work.
>
> > In the introduction, you say MaxEnt MoCo first obtains $E[\phi(X’)]$ for all $(x,a)$. That sounds expensive and seems to be computed on demand; but what is the cost of solving the lazy approximation (P1).
>
> In large or continuous MDPs, we can use function approximation to estimate these expectations. It is equivalent of fitting a regression model $\psi$ to a dataset $(X_i, A_i, \phi(X_i’))$ where $(X_i, A_i)$ is the input and $\phi(X_i’)$ is the output. Then an estimate of that expectation can be obtained at any $(x,a)$ by simply evaluating $\psi$ on $(x,a)$.
>
> >Can you comment about computational cost and scalablity of the approach given that the experiments were limited?
>
> In terms of scalability and the cost of optimization procedure in MaxEnt MoCo, we point out three properties that we believe makes the procedure scalable:
>
> 1. **Scale of the dual problem**: The dual problem of optimization problems (P1) and (P2) has only $d$ parameters. This means that it does not scale with the size of MDP. The value of $d$ will be kept rather small, since the agent needs to store $d$ functions for $\phi_i$ and $\psi_i$. Moreover, the dual problem of (P2), which is the version implemented in practice, is $\beta^2/4$-strongly convex (See equation D.15). The convexity allows efficiently solving the problem.
>
> 2. **Parallelization:** Sampling from the model is usually done in batches. Correcting each of these samples is an independent optimization problem that can be parallelized with careful implementation.
>
> 3. **Adjusting the computation cost:** It is possible to adjust the amount of correction to meet the computational budget. Dual parameters equal to zero correspond to no correction and using the model itself. We expect that partially optimizing the dual parameters and making few updates on them still partially corrects the model. Another technique is to use stronger regularization of dual parameters by choosing a larger value for the hyper parameter $\beta$, which makes the loss more strongly convex and easier to optimize. We provide an analysis of MaxEnt MoCo for the general choice of $\beta$ in Theorem 5 and 6.
>
> We have included the runtimes of our experiments in the new revision of the paper. We have also provided the computation time for the correction procedure, which is 0.2 - 1.5 seconds for all 144 state-actions depending on the setting.
>
> > I may have missed it but would it make sense to compare how close the dynamics of the real environment with the predicted and corrected one? If that's the case, evaluations of how well the environment can be predicted might be helpful even without additional results from RL experiments.
>
> We thank the reviewer for the suggestion. We have now included Figure 5 in the appendix to show the error of original and corrected dynamics in our experiments. It can be observed that the correction procedure successfully reduces the model error. The effect is stronger with larger values of $d$.

---

### Official Review · Reviewer_aRvY · 2023-11-06

**Soundness:** 3 good
**Presentation:** 2 fair
**Contribution:** 2 fair
**Rating:** 6
**Confidence:** 3

**Summary:**

This paper proposes a novel method for planning with an imperfect model, called MaxEnt Model Correction (MaxEnt MoCo). It *corrects* the next-state distribution of the model such that its expected value aligns with the true environment. This is achieved through Maximum Entropy density estimation.

Building on top of MaxEnt MoCo, they propose Model Correcting Value Iteration (MoCoVI) and the sample-based variant Model Correcting Dyna (MoCoDyna). Both methods iteratively update the basis function, using the value functions derived from MaxEnt MoCo.
Theoretical analysis suggests that the MoCoVI may converge to the true value function at a faster rate than approximate VI, under specific conditions. The efficacy of the proposed methods is empirically validated in a 6x6 grid world environment.

**Strengths:**

1. The paper delivers rigorous theoretical results.

2. The approach is novel to the best of my knowledge.

3. The paper is overall well organized, making it relatively easy to follow.

**Weaknesses:**

1. I am not fully convinced by the potential faster convergence of MoCoVI than VI.

    Let’s say that the model is perfect, why should MoCoVI enjoy a faster convergence? Also, in Theorem 2, the comparison between the order of $\gamma'$  and the order of $\gamma$ could be oversimplified. For instance, the constant could matter: the big O hides the constant $(3\sqrt{2})^K$ which could become significant.  Besides that, the $\gamma'^K$ also hides $(\frac{1}{1-\gamma})^K$ but is not addressed in the paper.

    Another concern is regarding the maximum of the ratio over K steps in $\gamma'$. Can the authors comment on the implications of it on the robustness of the algorithm?

2. The gap between theory and experiments: the theoretical analysis suggests applicability to both finite and continuous MDPs, yet experimental validation is confined to a small-scale tabular MDP. Broadening the experimental scope to include continuous MDPs would substantiate the theoretical findings more comprehensively.

3. The requirement in MaxEnt MoCo to sample the dynamics `d` times for each state-action visited, seems to restrict its practical applicability.

4. The paper could be improved with additional clarifications in certain sections. I would ask the authors to help address the following:

    a. Is there any assumption on the action space? It does not appear to have been stated in the paper, except for MoCoDyna where Algorithms 1 assumes a finite MDP. It appears that there's an implicit assumption that MaxEnt MoCo and MoCoVI are applicable to both finite and infinite action spaces. However, the approximate VI method that MoCoVI is compared with, traditionally assumes a finite action space [Munos 2007]. Could the authors clarify why the finite action space assumption is not necessary for MoCoVI?

    b. The introduction claims that the theoretical analysis is applicable to “both finite and continuous MDPs”. Does this applicability refer to both the exact and approximate versions of the proposed methods? Considering the complexity often associated with analyzing continuous spaces in RL, a further explanation on how the proposed analysis overcomes these challenges would be beneficial.

    c. The algorithm for MoCoDyna, as presented in Section 5, is specific to finite MDPs, yet prior analysis encompasses both finite and continuous MDPs. The paper suggests the possibility of extending MoCoDyna to incorporate function approximation without elaborating on the approach. Could the authors clarify if the focus on finite MDPs is solely for the sake of a simpler presentation, or are there inherent difficulties when adapting the algorithm to function approximations?

    d. The significance of the additional `c` features in the `d+c` features in MoCoDyna is not clear. Could the authors provide clarifications, and guidance on how to select `c`?

    e. The paper assumes that we can make exact queries to P in MoCoVI. Yet, the algorithm description only has approximation $\psi$ and lacks details on how $\psi$ is updated.

Minor:

It would be beneficial if the algorithmic procedures for MaxEnt MoCo and MoCoVI were presented (perhaps in the appendix) in a structured algorithm format, similar to MoCoDyna.

Typos:

1. Is the $|| V^{\pi_{PE}} ||_\infty$ before Sec. 3.2 missing anything?

2. Sec. 3.2: domain of $\phi$ should be $\mathcal{X}$ $\times \mathcal{A}$ instead of $\mathcal{X}$.

3. Sec. 3.1: in the equation at the end of paragraph 1, the expectation should be w.r.t. $\bar{P}$.

**Questions:**

Please see the weakness section for my questions.

---

> ### Author Response · Authors · 2023-11-22
> **Response to Reviewer aRvY (Part 1)**
>
> We want to thank the reviewer for their thorough review and feedback. Here are some clarifications about the points raised in the review.
>
> ### **1)**
> The reviewer has pointed out an insightful special case that shows why MoCoVI converges faster than VI. Assume that the model is perfect ($\epsilon_\text{Model} = 0$). Also for simplicity assume that both VI and MoCoVI obtain $PV$ for any value function $V$ perfectly ($\epsilon_\text{Query} = 0$). In this case, MoCoVI converges immediately and $V_0$ will be the solution ($V^{\pi_\text{PE}}$ for PE and $V^*$ for control). This is reflected in our bound in Theorem 2 as $\gamma’ = 0$.
>
> To see this, observe that the correction procedure will not change $\hat P$ at all because distribution $\hat P(\cdot | x,a)$ satisfies the constraints in optimization problem (P1) and achieves the optimal objective value of $0$. Consequently, the corrected dynamics $\bar P$ used to calculate $V_0$ will be the model $\hat P$ itself, which due to our assumption of perfect model is equal to $P$. Then as described in the second paragraph of Section 4, we have $V_0 = V^*(R, \bar P) = V^*(R, \hat P) = V^*(R, P) = V^*$.
>
> In general and regarding Theorem 2, we want to clarify that the big O notation used in the discussion of Theorem does not hide the constants $3 \sqrt{2}$ and $1/(1-\gamma)$. In fact, they do impact the asymptotic rate $O(\gamma’^k)$. This notation hides any constant coefficient for the terms in the bound (e.g. $1 - 3c_1 \lVert \epsilon_\text{Model} \rVert$ in the second inequality) but constants inside $\gamma’$ change the asymptotic rate similar to how $O(2^x)$ and $O(3^x)$ are different. The key property of this rate regardless of these constants is that as $\epsilon_\text{Model}$ goes to zero, the rate $\gamma’$ goes to zero as well. Thus, if the model error is smaller than a threshold, which is at most $(1-\gamma)/(3\sqrt{2})$, we will have $\gamma’ < \gamma$, and MoCoVI converges faster than VI. The aforementioned constants affect the value of this threshold, but not the existence of it.
>
> The maximum term in $\gamma’$ captures how well the linear combination of the past calculated value functions in MoCoVI can approximate the true value function. This term is upper bounded by 1, so the theorem would still hold without it. However, it could be very small or potentially zero, if for example $V^*$ is well approximate with the initial basis functions. A tighter bound would be to use the geometric mean instead of the maximum, but we decided on this version for the sake of simplicity.
>
>
> ### **2)**
>  We agree with the reviewer on this point. Though, the experimental setup for a model-based RL algorithm on a continuous MDP requires the use of function approximators such as DNN. Implementations that work competitively in practice usually involve a lot of implementation-level fine-tuning. In this paper we focus on the fundamental understanding of MaxEnt MoCo through theory and do not get involved with these implementation nuances. Yet, we believe the study of our algorithms in continuous MDPs with function approximation is a very interesting future work.
>
> ### **3)**
>   This might be a misunderstanding as MaxEnt MoCo does not require sampling the dynamics $d$ times. Having $d$ constraints in (P1) or terms in the summation in (P2) may incorrectly suggest that we need the number of samples to be proportional to $d$ to obtain $\psi_i$ functions (which estimate $P\phi_i$). But this is not needed as the same set of samples can be used to find all $\psi_i$ for $i = 1, \dotsc, d$. For example, if we have a single sample from $X’ \sim \mathcal{P}(\cdot|x,a)$, we can form $\psi_i(x,a) = \phi_i(X’)$ for all $i = 1, \dotsc, d$.
>
> As we mentioned in the beginning of Section 5 and 3.2, finding functions $\psi_i \colon \mathcal{X} \times \mathcal{A} \to \mathbb{R}$ is a regression problem. When using function approximation in continuous MDPs, this can be done by fitting a regression model to any dataset of real samples even though it doesn’t cover all state-action pairs. In finite MDPs and without the generalization of function approximation, any RL algorithm requires knowledge of all state-action pairs relevant to the problem. Still, the requirement of samples does not depend on the order $d$ of the algorithm as the same samples can be used for finding all query result functions $\psi_i$.

---

> > ### Author Response · Authors · 2023-11-22
> > **Response to Reviewer aRvY (Part 2)**
> >
> > ### **4a)**
> > In short, we don’t assume that action space is finite. This has become possible with our assumption that we can solve the approximate MDP $\hat M$ given the model $\hat P$ accurately. In the new revision, we have stated this assumption explicitly in Section 2. Practically, this is a reasonable assumption considering there are MBRL algorithms that work well in environments of our interest that have continuous actions (e.g. [Janner et al. 2019] in paper). However, providing algorithms with theoretical guarantees or analysis in continuous action spaces is challenging. Value iteration is one example, and many studies on VI assume finite actions. In the following, we discuss the challenges of continuous actions in more details.
> >
> > We first need to explain why most Approximate (or Fitted) Value Iteration algorithms consider finite action spaces. We mention two reasons:
> >
> > 1) **Computational.**
> > The computation of $\arg\max_{a \in \mathcal{A} } Q(x,a)$, needed to compute the greedy policy, or $\max_{a \in \mathcal{A}} Q(x,a)$, appearing in the Bellman optimality operator, becomes non-trivial. As the function $Q(x,\cdot): \mathcal{A} \rightarrow \mathbb{R}$ is a nonlinear and non-concave function in general, computing the maximizer is NP-Hard, unless we make a further assumption on the action-value functions.
> >
> > 	In practice, we can use an approximate optimizer, for example by performing gradient ascent on the action. This may empirically work well, but we cannot provide a guarantee on the sub-optimality of the obtained action, that is, if the optimizer finds $a^+$, the value of $Q(x,a^+)$ can be much worse than $\max_a Q(x,a)$.
> > 	As we cannot provide a guarantee on the quality of the maximizer, without further assumptions, many theoretical work on FVI avoided this issue altogether by focusing on finite action spaces, for which the maximizer can be found exactly.
> >
> > 2) **Statistical.**
> > If we want to provide a statistical theoretical guarantee on FVI, which is going beyond the error propagation result, we need to consider the capacity of the involved functions spaces. Specifically the complexity of the space
> > $\\{ \max_{a \in \mathcal{A}} Q(·,a) : Q \in \mathcal{F} \\}$ appears in such results. When the action space is continuous (a subset of $R^p$), the complexity of this latter space might be infinity even if the complexity of $\mathcal{F}$ is finite. This is shown in Proposition 2.1 of
> > Antos, Munos, Szepesvari, “Fitted Q-iteration in continuous action-space MDPs," Advances in Neural Information Processing Systems, 2007.
> >
> > 	There are ways to control this complexity. For example, we can assume that $\mathcal{F}$ consists of functions that are Lipschitz w.r.t. the state. This is a requirement that is not needed if the action space is finite, so many theoretical papers on FVI avoided this extra assumption and worked with a finite action space.
> >
> > After this background, we are ready to discuss why we did not explicitly have any assumption on the action space.
> >
> > **The computational challenge:** We assume that for a given model $\bar{\mathcal{P}}$, we can solve the MDP $\bar{M} = (\mathcal{X}, \mathcal{A}, \mathcal{R}, \mathcal{\bar{P}})$, and the solution is $\bar{V}^*$.
> > The computational challenge of solving this augmented MDP is ignored in our theoretical analysis. However, note that solving an MDP given a model of it is standard in MBRL algorithms and is not a new practical challenge.
> >
> > **The statistical challenge:** We are not concerned with the possible statistical challenges of using a continuous action space because we do not provide any statistical learning theoretical guarantee after all. Our results are expressed in terms of the norms of model error ($\lVert\epsilon_\text{Model} \rVert$) and query error ($\lVert\epsilon_\text{Query} \rVert$). For our purposes, the model might even be a given simulator that is not learned from samples. If the model or queries are estimated using samples, it is possible to provide statistical analysis of the errors. We have not gone into these details. In other words, we have provided "error propagation" results.
> >
> > One may ask why [Munos 2007] considered finite action space, even though their work is only about error propagation and not statistical guarantee. We note that Munos' work also considers that the state space is finite (except in a short discussion in Section 7). We believe the reason is mainly to simplify the exposition.
> > We also suspect that the reason Munos did not talk about continuous action space was because he was aware of both statistical and computational challenges when he wrote that paper. When that work was published, Munos together with Szepesvari and Antos have already published some papers for which the statistical challenge of infinite action space would show itself ([1] and [34] in the paper), so he probably have decided to have avoid discussing them and have a cleaner presentation. Of course, this is merely our guess about his intention.

---

> > > ### Author Response · Authors · 2023-11-22
> > > **Response to Reviewer aRvY (Part 3)**
> > >
> > > ### **4b)**
> > >  Both of our exact and approximate results apply to continuous MDPs. Similar form of analysis for the comparable RL algorithms (e.g. value iteration) in continuous MDPs is well-understood (See last paragraph of Section 3 for some related work). A major part of the difficulty of theoretical RL in continuous MDPs is due to the difficulty of exploration. We do not deal with exploration in this paper and the existing techniques in the literature are sufficient for our results.
> > >
> > > ### **4c)**
> > >  We presented MoCoDyna in finite MDPs due to the simplicity of its implementation while capturing the most critical challenges that arise when using samples from the environment instead of distributions, i.e. asynchronous updates and approximation in queries. Since both the model correction procedure and queries are extendable to continuous MDPs, we believe the algorithm can be extended to continuous MDPs with almost the same structure.
> > >
> > > In terms of scalability and the cost of the correction procedure in continuous MDPs, as discussed in the paper, we do not need to iterate over all state-action pairs. We only correct samples taken from the model. The dual problems of optimization problems (P1) and (P2) have only $d$ parameters. This means that it does not scale with the size of MDP even if the MDP is continuous. A small modification we expect is that the first integral in the dual loss (See eq. 2.4) needs to be estimated with samples from $\hat p$. To see this, note that the integral is actually an expected value under distribution $\hat p$
> > >
> > > The other part that will change in continuous MDPs is the way we obtain the query results $\psi_i$ (line 5 in the algorithm box). In continuous MDPs, the agent usually stores the past samples in its replay buffer. Using the replay buffer, the query result can be obtained by fitting a regression model. For each sample $X_n, A_n, X’_n$ in the replay buffer, the input of the regression is $(X_n, A_n)$ and the target is $\phi_i(X’_n)$. Finally, the planning part of the algorithm (line 6), which solves an MDP given a model of it, can be changed to any RL algorithm that works in continuous MDPs.
> > >
> > > ### **4d)**
> > >  The additional $c$ basis functions is only needed when using stochastic approximation (SA) to estimate $\psi$ functions as we do in this paper. In stochastic approximation, we do not store samples collected in the past. To form an estimate of $P \phi$ for a new basis function $\phi$ we need to wait a number of steps to collect samples and make SA updates. We do not use the $c$ most newly added basis functions because the corresponding $\psi$ functions have not received enough updates to approximate $P\phi$. This technique ensures all the basis functions used for MoCo have received at least $cK$ updates. $c$ should be chosen such that $cK$ is large enough. We set $c=2$ throughout our experiments.
> > >
> > > ### **4e)**
> > >  We are not making that assumption for MoCoVI. We did discuss the special case of exact queries and the PE problem to provide intuition of the algorithm, but the algorithm definition is for general approximate queries. We have a brief discussion of how the estimates can be formed in the beginning of Section 3 and give a detailed version for finite MDPs in Section 5. Since any method such as stochastic approximation or regression is applicable in MoCoVI, we keep the algorithm definition general and don’t go into details of this procedure.

---

### Author Response · Authors · 2023-11-23
**General Response to Reviewers and Rebuttal Revision**

We thank the reviewers for their reviews and valuable feedback. We are glad to see the reviewers have found our work novel and interesting (reviewers aRvY and sy6v), with rigorous and good theoretical contributions (reviewers aRvY and sy6v) that improves conventional approaches (reviewer sy6v), and well-written (all reviewers).

We have made a revision of the paper according to the reviewers feedback. The significant changes are highlighted with blue color. The changes include:
1. Some clarifications in the main text
2. New figure showing how the correction procedure reduces model error
3. Details on the computational cost and runtime of algorithms.
4. More details regarding the experiments and hyperparameters
5. Several minor fixes

---

### Meta-Review · Area_Chair_iQNj · 2023-12-11

**Metareview:**

This study introduces a pioneering approach for planning using an imperfect model, termed 'MaxEnt Model Correction' (MaxEnt MoCo). This method adjusts the predicted next-state distribution of the model to ensure its expected value aligns closely with the actual environment, utilizing Maximum Entropy density estimation for this purpose

This paper meets the standards of ICLR and has received recognition from the reviewers.

**Justification For Why Not Higher Score:**

N/A

**Justification For Why Not Lower Score:**

N/A

---

### Decision · Program_Chairs · 2024-01-16

Accept (poster)